A proposed terminology for the dentition of gomphodont cynodonts and dental morphology in Diademodontidae and Trirachodontidae

Hendrickx Christophe 1 christophendrickx@gmail.com
Abdala Fernando 1 2
http://orcid.org/0000-0002-1008-0687 Choiniere Jonah N. 1
1 Evolutionary Studies Institute and School of Geosciences, University of the Witwatersrand , Johannesburg, Gauteng , South Africa
2 CONICET-Fundación Miguel Lillo, Unidad Ejecutora Lillo , San Miguel de Tucumán , Argentina
De Baets Kenneth
Electronic publication date: 2019 Jun 13
Publication date: 2019
Volume: 7
Electronic Location ID: e6752
Received 2018 Mar 9; Accepted 2019 Mar 10
Copyright: © 2019 Hendrickx et al.
Copyright year: 2019
Copyright holder: Hendrickx et al.
License: This is an open access article distributed under the terms of the Creative Commons Attribution License, which permits unrestricted use, distribution, reproduction and adaptation in any medium and for any purpose provided that it is properly attributed. For attribution, the original author(s), title, publication source (PeerJ) and either DOI or URL of the article must be cited.
License URL: https://creativecommons.org/licenses/by/4.0/

Keywords: Teeth, Gomphodont, Trirachodontids, Diademodontids, Dentition, Dental evolution, Cynodontia

Funding: DST-NRF Centre of Excellence, Palaeosciences PPD2015/17CH and CoE2017-038 Postdoctoral Fellowship from the University Research Committee (URC) of the University of the Witwatersrand to Christophe Hendrickx; the Palaeontological Scientific Trust (PAST), the DST-NRF Centre of Excellence, Palaeosciences, an NRF African Origins Platform (AOP) #98800 NRF Competitive Support for Rated Researchers (CPRR) #98906 NRF, CONICET and Agencia Nacional de Promoción Científica y Tecnológica, Argentina (PICT) 2015–2389 This research was financially supported by the DST-NRF Centre of Excellence, Palaeosciences grant PPD2015/17CH and CoE2017-038 as well as the Postdoctoral Fellowship from the University Research Committee (URC) of the University of the Witwatersrand to Christophe Hendrickx; the Palaeontological Scientific Trust (PAST), the DST-NRF Centre of Excellence, Palaeosciences, an NRF African Origins Platform (AOP) grant (#98800), and an NRF Competitive Support for Rated Researchers (CPRR) grant to Jonah Choiniere (#98906); and NRF, CONICET and Agencia Nacional de Promoción Científica y Tecnológica, Argentina (PICT 2015–2389 to N. Giannini) to Fernando Abdala. The funders had no role in study design, data collection and analysis, decision to publish, or preparation of the manuscript.

==============================
Gomphodont cynodonts were close relatives of mammals and one of the Mesozoic lineages of cynodont therapsids that became extinct at the end of the Triassic. Gomphodonts were omnivorous to herbivorous animals characterized by labiolingually expanded postcanines, which allowed tooth-to-tooth occlusion. The morphology of the upper and lower postcanines presents important means of distinguishing among major lineages within Gomphodontia, that is, Diademodontidae, Trirachodontidae, and Traversodontidae, but the dentition of most Diademodontidae and Trirachodontidae remain poorly documented. Here, we present a comprehensive description of the dentition of each diademodontid and trirachodontid species, as well as detailed illustrations of each dental unit, after firsthand examination of material and 3D reconstructions of postcanine teeth. Based on dental morphology, Trirachodon berryi and “Trirachodon kannemeyeri,” considered as separate taxa by some authors are here interpreted as representing different ontogenetic stages of the same species. Likewise, Sinognathus and Beishanodon, thought to belong to non-cynognathian cynodonts and traversodontids by some authors, are referred to Trirachodontidae and Gomphodontia based on dental characters, respectively. Finally, we propose a standardized list of terms and abbreviations for incisors, canines, and postcanines anatomical entities, with the goal of facilitating future descriptions and communication between researchers studying the gomphodont dentition.

Introduction

Gomphodont cynodonts form a radiation of Triassic therapsids known from the late Olenekian to the Norian on all continents but Australia (Battail, 1983; Abdala & Ribeiro, 2003; Hopson, 2005, 2014; Abdala & Gaetano, 2018; Fig. 1A). Members of this clade were small to medium-sized (from approximately 30 cm to two m in body length), quadrupedal animals characterized by labiolingual expansion of the upper and sometimes lower postcanines (gomphodont morphology) allowing crown-to-crown occlusion (Seeley, 1895, 1908; Crompton, 1972; Reisz & Sues, 2000; Abdala, Neveling & Welman, 2006; Hopson, 2014). Such morphology of the postcanines suggests that gomphodonts were omnivorous or possibly exclusively herbivorous animals, feeding on hard plant material (Reisz & Sues, 2000; Abdala, Neveling & Welman, 2006; Liu & Abdala, 2014). Three clades, mainly differentiated by their postcanine morphology, are currently recognized among Gomphodontia, the Diademodontidae, Trirachodontidae, and Traversodontidae (Hopson, 2005; Liu & Abdala, 2014; Hendrickx, Abdala & Choiniere, 2016).

Figure 1 Phylogeny and geographic and stratigraphic distribution of cynodont clades.

(A) Tree topology based on the results of the cladistic analyses of Liu & Abdala (2014) for cynognathians and Martínez, Fernández & Alcober (2013) and Martinelli, Soares & Schwanke (2016) for probainognathians. Titanogomphodon, considered to be a close relative of Diademodon by Keyser (1973) and Martinelli, De La Fuente & Abdala (2009), is placed among Diademodontidae. Node 1, Eucynodontia; Node 2, Probainognathia; Node 3, Cynognathia; Node 4, Gomphodontia; Node 5, Diademodontidae; and Node 6, Trirachodontidae. Arrows indicate that the clades extend beyond the Lower Jurassic. (B) Stratigraphic distribution of Diademodontidae (in yellow) and Trirachodontidae (in blue-purple), with Diademodon in pale yellow, Titanogomphodon in dark yellow, Langbergia in pale blue, Cricodon in blue sky, Trirachodon in dark blue, Sinognathus in light purple and Beishanodon in dark purple. Stratigraphic extension of geological units based on Liu, Ramezani & Li (2018) for the Ermaying Formation, Gao et al. (2010) and Niu et al. (2018) for the Hongyanjing Formation, Rubidge (2005) for the Cynognathus Assemblage Zone, Wynd et al. (2018) for the Manda Beds and the Upper Omingonde, Ntawere and Fremouw formations, and Ottone et al. (2014) for the Río Seco de la Quebrada Formation. Stars denote the U-PB geochronologic ages of 243.53 Ma from Liu, Ramezani & Li (2018) for the upper Ermaying Formation in China, and 235.82 Ma from Ottone et al. (2014) for the Río Seco de la Quebrada Formation in Argentina. Wavy lines indicate unconformities and diagonal lines formational interfingering, whereas dashed lines indicate an unclear relationship to the geologic time scale. Age of the Triassic stage limits based on Cohen et al. (2018). Abbreviations: CAZ, Cynognathus Assemblage Zone; E, Early Triassic; Fm, Formation; M, Middle Triassic; Ma, million years; Pr., Province. Ottone et al.’s (2014) hypothesis refers to the Carnian dating for levels of the Río Seco de la Quebrada Formation. Black silhouettes credit: Sergey Meleshin (Sinognathus), José Eduardo Camargo Martínez (all others), used with permission.

Diademodontidae is an early diverging lineage of gomphodonts with low taxonomic diversity at the generic and species level (Fig. 1A). Two valid taxa (i.e., Diademodon and Titanogomphodon) known from the Olenekian and Anisian of southern and East Africa, Argentina and possibly Antarctica currently compose this group (Keyser, 1973; Hammer, 1995; Martinelli, De La Fuente & Abdala, 2009; Fig. 1B). A possible lazarus diademodontid from the Lower Jurassic of South Africa (Bordy et al., 2017; Abdala & Gaetano, 2018) was described by Abdala et al. (2007) but most likely represents a non-gomphodont tetrapod (F. Abdala, personal observation). The postcanine dentition of diademodontids is heterogeneous and separated into conical, gomphodont, and sectorial teeth, which is considered as the most primitive condition of dental morphology among Gomphodontia (Crompton, 1972; Gow, 1978; Hopson, 2005). The upper and lower gomphodont postcanines of diademodontids display several ridges and accessory cusps whose position varies on the crown and along the tooth row. The distalmost part of the diademodontid tooth row is also characterized by a minimum of three sectorial teeth, and one or several transitional postcanines gradually changing from a gomphodont morphology to a more labiolingually compressed sectorial type of crown (Crompton, 1972; Hopson, 2005).

Trirachodontidae are represented by five valid species restricted to the Olenekian and Anisian of Southern Africa and Asia (Abdala, Neveling & Welman, 2006; Gao et al., 2010; Sidor & Hopson, 2018; Figs. 1A and 1B). Trirachodontids were relatively small animals (i.e., <1 m), and some had a burrowing/fossorial lifestyle (Groenewald, Welman & Maceachern, 2001; Smith & Swart, 2002). The postcanine tooth row of trirachodontids encompasses a lower number of sectorial teeth than that of diademodontids, which occur immediately distal to the gomphodont postcanines (Hopson, 2005). The gomphodont postcanine of trirachodontids shows a clearly defined centrally positioned transverse crest made of labial, central and lingual cusps and ringed by mesial and distal cingula on the rim of the tooth (Abdala, Neveling & Welman, 2006).

The most derived and most taxonomically diverse gomphodont radiation are the Traversodontidae, which form a major element of tetrapod communities in Gondwana during the Middle and Late Triassic (Liu & Sues, 2010). This clade currently encompasses more than 20 taxa known from the Anisian to the Norian and possibly Rhaetian of Africa, South America, North America, and Europe (Hopson, 2014; Liu & Abdala, 2014; Fig. 1A). Traversodontids increased in body size throughout the Triassic, with younger members reaching up to two m in length (Reisz & Sues, 2000; Liu & Abdala, 2014). The upper and lower postcanines of traversodontids typically encompass gomphodont teeth, and only a few taxa retained one or several sectorial postcanines at the distal rear of the tooth row in small, possibly young individuals (Liu & Sues, 2010).

Similar to other cynodonts, the dentition of gomphodonts is the most diagnostic element of the skeleton (Liu & Sues, 2010). Because of their high resistance compared to other parts of the skeleton due to greater density and lower permeability (Martin, 1999), teeth are also the most commonly preserved material in Gomphodontia, with dental elements known in every gomphodont taxa hitherto described. Given the postcanine diversity and complexity among gomphodonts, the dentition of these cynodonts typically receives particular attention and is often relatively well-described. Nonetheless, thorough description of the dental material is often provided for gomphodont postcanines, whereas information as well as detailed figures on the incisors, canines and conical and sectorial postcanines are omitted in the descriptions of many gomphodont taxa. In addition, we have noticed inconsistencies in the terminology and abbreviations used in discussions of the dental gomphodont anatomy, with several authors providing different terms for the same dental structure. The labiomesial accessory cusp seen in the gomphodont postcanines of some traversodontids has, for instance, received no less than 11 different terms, and more than eight terms describe the labial and labiodistal accessory cusps. Such a large number of terms and abbreviations for the same dental structure leads to confusion, and a clear and detailed terminology, which will greatly facilitate the description and communication of the dentition of gomphodont cynodonts, remain to be provided.

This paper: (i) proposes a standardized list of terms and abbreviations for incisors, canines, and postcanines anatomical entities, with the goal of facilitating future descriptions and illustrations of the gomphodont dentition; and (ii) provides a comprehensive description and detailed illustrations of the dentition of all known diademodontid and trirachodontid taxa. Tooth replacement pattern and postcanine occlusion, which were treated in detail by several authors for Diademodontidae and Trirachodontidae (Crompton, 1955, 1972; Fourie, 1963; Ziegler, 1969; Hopson, 1971; Osborn, 1974; Sidor & Hopson, 2018), is beyond the scope of this paper. Likewise, a comprehensive description of the dentition of Traversodontidae, which will form the base of another contribution, falls outside the scope of this study. Finally, the evolution of the gomphodont dentition will be thoroughly explored in a third contribution based on a cladistic analysis performed on a dentition-based data matrix encompassing all gomphodont taxa.

Material and Methods

Dental features were investigated on incisors, canines and gomphodont and sectorial postcanines preserved within the upper and lower jaws as well as isolated teeth of non-traversodontid gomphodonts. The dentition of 89 specimens belonging to six diademodontid and trirachodontid genus-level taxa deposited in 11 scientific collections from South Africa, Namibia, Germany, the UK, and China were examined first-hand (Table 1). The specimens were referred to a diademodontid/trirachodontid species based on a dental diagnosis that will be provided for each gomphodont taxon in a forthcoming contribution. Denticles, crown ornamentations and enamel texture were observed with a digital microscope AM411T-Dino-Lite Pro. Only Beishanodon youngi could not be examined first hand, and we relied on Gao et al.’s (2010) publication in which the dentition was comprehensively described and illustrated.

Table 1 List of specimens of non-traversodontid gomphodonts examined in this study.

Taxa	Specimens	
Diademodon tetragonus	SAM-PK-571, 4002, 5877, 6216, 6218, 6219, 11265, K175, K177, K180, K183, ?K4660, ?K4661, K5223, K5266, K8971, K9968, K9969; AM 458, 3753; BP/1/2522, 3639, 3756, 4529, 4669, 4677; BSP 1934 VIII 14, 15, 16, 19, 20, 505; MB R1004; NHMUK PV R3303, R3588, R3765; GSN R321, RK3	
Titanogomphodon crassus	GSN R323	
Langbergia modisei	NMQR 3255, 3251, 3256, 3268, 3280, 3281; BP/1/5362, 5363; SAM-PK-11481	
Cricodon metabolus	UMZC T905; BP/1/5540, 5835, 6102, 6159; NHMUK PV R3722, K36800; SAM-PK-6212, K5881a, b	
Trirachodon berryi	NHMUK PV R3579, R2807, R3306, R3307, R3350, R3721; AM 434, 461; BP/1/4258, 4658, 4661; BSP 1934 VIII 21, 22, 23; CGP INN 2000-7-2A, CGP unnumbered; NMQR 1399; SAM-PK-987, 5880, K142, K170, K171, K4801, K4803, K5821 (=12168?), K7888, K10157, K10161, K10176, K10207, K10411	
Sinognathus gracilis	IVPP V2339	
Note:

Specimens with the best-preserved dentition are underlined, and holotypic specimens are in bold.

3D-models of teeth were generated for lower and upper postcanines in all taxa but Sinognathus and Beishanodon (Table 1) through photogrammetric techniques using Agisoft Photoscan Standard (Version 1.3.4, 2017, retrieved from http://www.agisoft.com/downloads/installer/) and the photos taken with the digital microscope AM411T-Dino-Lite Pro. The batch process followed in Agisoft Photoscan to reconstruct the postcanines in 3D consists of four steps: (i) more than 150 photos taken in all views were aligned with the highest accuracy using standard options (i.e., with generic pre-selection and 40,000 and 4,000 key point limit and tie point limit, respectively); (ii) a dense cloud was then built in ultra-high quality with an aggressive depth filtering and no reuse depth map; (iii) the mesh was then built with a high face count and default options (i.e., a custom face count of 200,000 faces, arbitrary surface type, interpolation enabled, and vertex color calculated); (iv) the texture was finally built using the default options (i.e., generic mapping mode, texture from all cameras, mosaic blending mode, texture size, and count of 4,096 and 1, respectively, no color correction and using the hole filling option). A total of 20 3D-models are deposited and freely downloadable in the MorphoBrowser database (Appendix 1; http://pantodon.science.helsinki.fi/morphobrowser/).

The dental morphology of Trirachodon berryi was also investigated based on CT-scan data from the specimen AM 461, a fully preserved skull and the holotype of Trirachodon kannemeyeri. AM 461 was CT-scanned at the Evolutionary Studies Institute (ESI) of the University of the Witwatersrand (Johannesburg) using a Nikon Metrology computed tomography XTH 225/320 LC, with a voxel size of 0.0668 mm, and generating 1,778 images of 1,109*779 pixels resolution. The postcanines were visualized and reconstructed using the software VGStudio Max 3.0 available at the ESI.

Neotrirachodon expectatus (Tatarinov, 2002) and Redondagnathus hunti (Lucas et al., 1999; Spielmann & Lucas, 2012), classified as trirachodontids by their authors, were not included in this study as they probably do not represent gomphodont cynodonts (Sidor & Hopson, 2018). Neotrirachodon, synonymized with Antecosuchus by Ivakhnenko (2011), likely belongs to a bauriid therocephalian (Battail & Surkov, 2000; Abdala, Neveling & Welman, 2006; Abdala & Smith, 2009; Gao et al., 2010; Sues & Hopson, 2010; Ivakhnenko, 2011) whereas Redondagnathus’ dental material displays several features absent in Trirachodontidae, namely: a central cusp strongly mesially/distally deflected from the labial and lingual cusps and much higher than the two latter cusps, no valley-like concavities separating the central cusp from the labial and lingual cusps, a cingulum significantly apically higher than the other one, apically pointed cingular cuspules varying dramatically in size along the cingulum, presence of a basally inclined spalling surface extending below the cingulum as well as an important protuberance on the basal part of the root (Sidor & Hopson, 2018; C. Hendrickx, 2018, personal observation). We, therefore, agree with Sidor & Hopson (2018) and consider that the teeth of Redondagnathus do not share enough dental features with trirachodontids and gomphodonts to be confidently referred to these clades. Likewise, CGP JSM 100, interpreted as a possible juvenile Trirachodon by Hopson (2005), was not considered in this study as the specimen likely represents a new taxon of basal traversodontid (C. Hendrickx, 2018, personal observation; F. Abdala, 2018, personal observation), a possibility also discussed by Hopson (2005). The dentition of this specimen, indeed, shows several dental features absent in trirachodontids and seen in basal traversodontids. They include unserrated incisors, a central cusp of transverse crest close to the lingual cusp in the upper postcanines, quadrangular or subrectangular lower postcanines in occlusal view, a long axis of lower gomphodont postcanine parallel to the long axis of the mandibular tooth row, and a labial cusp lower than the lingual cusp in lower postcanines. The dentition and phylogenetic position of CGP JSM 100 will be thoroughly discussed elsewhere.

Terminology

Quantitative parameters

The dental positional and morphometric nomenclature follows Smith & Dodson (2003) and Hendrickx, Mateus & Araújo (2015), with the following abbreviations being used in this study (Table 2).

Table 2 Measurement variables and associated terms and abbreviations used in this study.

Crown base length (CBL)	Maximum mesiodistal extent of the crown base at the level of the cervix (i.e., the transition between the crown and the root and forming the basal extension of the enamel layer; Smith, Vann & Dodson, 2005; Hendrickx, Mateus & Araújo, 2015)	
Crown base width (CBW)	Labiolingual extent of the crown base at mid-length, perpendicular to the CBL, and at the level of the cervix (Smith, Vann & Dodson, 2005)	
Crown height (CH)	Maximum apicobasal extent of the distal margin of the crown (Hendrickx, Mateus & Araújo, 2015)	
Crown base ratio (CBR)	Ratio expressing the labiolingual elongation of the base crown and corresponding to the quotient of CBW by CBL (CBR = CBW ÷ CBL; Smith, Vann & Dodson, 2005)	
Denticle size density index (DSDI)	Ratio expressing the size difference between mesial and distal denticles (Rauhut & Werner, 1995) and corresponding to the quotient of the number of denticles per five mm on the mesial carina at mid-crown (MC) by the number of denticles per five mm on the distal carina (DC) at mid-crown (DSDI = MC ÷ DC)	

Proposed dental terminology

Crown microstructure nomenclature uses the terminology proposed by Sander (1997a, 1999). The anatomical nomenclature used to describe and annotate the external tooth morphology (Fig. 2) follows the terminology and abbreviations proposed below (Table 3). The notation presented by Crompton & Jenkins (1968) to describe the series of cusps (i.e., a, b, c, d, e, f for the lower postcanines, and A, B, C, D, E, and F, for the upper postcanines) in Thrinaxodon and Triassic mammals, and used by some authors to describe the sectorial teeth of gomphodonts (e.g., Sues & Hopson, 2010; Sidor & Hopson, 2018) and non-mammaliaform probainognathians (Oliveira et al., 2011; Soares, Martinelli & Oliveira, 2014; Martinelli, Soares & Schwanke, 2016; Martinelli et al., 2017; Pacheco et al., 2018), was incorporated into our proposed terminology. All abbreviations proposed by Crompton & Jenkins (1968) must, however, be italicized to not be confused with similar abbreviations referred to other dental sub-units (e.g., “c” is for the canine whereas “c” is for the distal accessory cusp). Our terminology and the abbreviations we used to annotate the figures do not take into consideration the position of the tooth within the upper or lower jaw and left or right side of the skull, as done by Crompton & Jenkins (1968) and other authors (Hopson, 2005; Sidor & Hopson, 2018). Abbreviations in capital and lower-case letters can nonetheless be used by authors to annotate dental features from the upper and lower jaw, respectively. In the same way, the letters “l” and “r” followed by the corresponding abbreviation can be used to annotate dental features from the left and right jaw, respectively. For instance, “lGPC” refers to the left upper gomphodont postcanine whereas “rmmc” refers to the mesial main cusp of the right lower postcanine.

Figure 2 Dental terminology used in this study.

(A and B) Diademodon rostrum in (A) palatal and (B) labial views; (C and F) idealized right upper gomphodont postcanine in (C) apical and (F) distal views (based on the upper gomphodont postcanine of Scalenodon angustifrons by Hopson, 2005; modified); (D and G) idealized left lower gomphodont postcanine in (D) apical and (G) distal views (based on the lower gomphodont postcanine of Scalenodon angustifrons by Hopson, 2005; modified); (E) upper right gomphodont postcanine of Menadon besairiei (UA 10601) in apical view; (H) upper right gomphodont postcanine of Diademodon tetragonus (SAM-PK-571) in apical view; (I) idealized sectorial postcanine of Diademodon tetragonus in lingual view (based on the second right upper sectorial postcanine of Diademodon tetragonus BP/1/4529); (J) apex of main cusp of an upper sectorial postcanine of Diademodon tetragonus with close up on the denticles, in labial view; (K) close up on the enamel surface texture of the second right upper sectorial postcanine of Diademodon tetragonus (BP/1/4529). Abbreviations: c, canine; ca, carina; cac, central accessory cusp; cap, crown apex; cc, central cusp (in green); cev, central valley; ci, cingulum; cic, cingular cuspules; clar, centrolabial ridge; clir, centrolingual ridge; co, crown; cos, concave surface; cpc, conical postcanine; cri, central ridge; cu, cusp; db, distal basin (in light grey); dac, distal accessory cusp; dar, distal accessory ridge; dca, distal carina; dcc, distal cingular cuspule; dci, distal cingulum (in violet); de, denticle; dia, diastema; dmc, distal main cusp; dri, distal ridge; dv, distal valley; ec, ectopterygoid; ent, enamel texture; gpc, gomphodont postcanine; i, incisor; j, jugal; lac, labial cusp (in red); lacc, labial cingular cusp; laci, labial cingulum (in orange); lacr, labiocentral ridge; ladc, labiodistal accessory cusp; ladr, labiodistal ridge; lamr, labiomesial ridge; lar, labial ridge; lav, labial valley; lic, lingual cusp (in blue); licc, lingual cingular cuspule; lici, lingual cingulum (in turquoise); licr, linguocentral ridge; lidc, linguodistal accessory cusp; lidr, linguodistal ridge; limc, linguomesial accessory cusp; limr, linguomesial ridge; lir, lingual ridge; liv, lingual valley; lri, longitudinal ridge; mar, mesial accessory ridge; mb, mesial basin (in light grey); mc, main cusp; mca, mesial carina; mcc, mesial cingular cuspule; mci, mesial cingulum (in beige); mmc, mesial main cusp; mri, mesial ridge; mv, mesial valley; mx, maxilla; pal, palatine; pc, postcanine; pcf, paracanine fossa; pmx, premaxilla; ro, root; se, serration; spc, sectorial postcanine; tc, transverse crest (in yellow); tpc, transitional postcanine; tun, transverse undulation.

For a consistent terminology, the positional terms mesial, distal, labial, and lingual, proposed as standard terms by Smith & Dodson (2003), were favored, with length, width, and height referring to the dimension of an anatomical entity in the mesiodistal, labiolingual, and apicobasal directions, respectively. We also favor the following hierarchy for the combination of positional terms: (1) apico- and baso- (i.e., apicocentral, apicodistal, apicomesial, apicolingual, basocentral, basodistal, basomesial, basolingual, basolabial); (2) centro- (centrolabial, centrolingual, centromesial, centrodistal); (3) labio- and linguo- (labiomesial, labiodistal, linguomesial, linguodistal); and (4) -mesial and -distal. An exception to this relates to the transverse crest, in which the centrolabial, centrolingual, labiocentral, and linguocentral ridges describe different entities of the crest. As for the dental formula, we propose the following notation: i4/3 : c1/1 : pc7-16/7-13 (cpc1-6 : gpc2-7 : tpc0-2 : spc1-4), with i for incisors, c for canines, pc for postcanines, cpc for conical postcanines, gpc for gomphodont postcanines, tpc for transitional postcanines, and spc for sectorial postcanines. Each dental sub-unit is separated by a colon, and the numbers, or ranges (noted x–x), before and after the slash correspond to the number of teeth from the upper and lower jaws, respectively, for each dental sub-unit (Table 3).

Table 3 Anatomical terms and abbreviations of the gomphodont dentition used in this study.

Accessory cusp (ac)	Minor pointed or rounded projection of dentine covered with enamel on the mesial/distal ridge (i.e., the labiomesial (lamc), labiodistal (ladc), linguomesial (limc) and linguodistal (lidc) accessory cusps; Figs. 2C–2G) and/or transverse crest (i.e., the central accessory cusp (cac); Fig. 2C) of a gomphodont postcanine as well as on the carina (i.e., the mesial (mac) and distal (dac) accessory cusps; Fig. 2I) of a sectorial postcanine	
Basin (ba)	Deep concavity located on the occlusal surface of a gomphodont postcanine, on the mesial and/or distal surfaces of the crown (i.e., the mesial (mb) and distal (db) basins, respectively; Figs. 2C–2E and 2G), and receiving the crown’s cusps and crests of postcanines of the opposite jaw during occlusion. The basin seen on gomphodont postcanines is also referred as “valley” (Crompton, 1972; Godefroit & Battail, 1997; Hopson, 2005)	
Canine (c)	Maxillary and dentary tooth located between the distalmost incisor and the mesialmost postcanine and specialized for cutting and/or piercing (Figs. 2A, 2B and 3F). It is usually the largest tooth of the series	
Carina (ca)	A sharp, narrow, and well-delimited ridge or keel-shaped structure running apicobasally on the crown and, in some case, on the root base, and typically making the cutting edge of the tooth (McGraw-Hill, 2003; Brink & Reisz, 2014; Fig. 2A). Incisors and canines of gomphodont cynodonts often have denticulated mesial and distal carinae, whereas the carinae of sectorial teeth bear accessory cusps and/or minute denticles. The carinae are also referred as “cutting edges” (Crompton, 1955; Sues & Hopson, 2010), “cutting ridges,” “enamel ridges” (Kemp, 1980), “serrated margins,” “marginal ridges” (Hopson, 1984), “keeled edges” and “sectorial edges” (Kammerer et al., 2012)	
Central accessory cusp (cac)	Minor pointed or rounded projection labial and/or lingual to the central cusp on the transverse crest of upper gomphodont postcanine (Fig. 2C). Central accessory cusps correspond to the “buccal accessory cusp” of Sues, Hopson & Shubin (1992) and the “accessory cusps on transverse ridge” of Hopson (2005)	
Central cusp (cc)	Main centrally positioned projection of dentine covered with enamel on the transverse crest of the gomphodont postcanine (Figs. 2C and 3). The central cusp is also known as the “middle cusp” (Kemp, 1980), “main central cusp” (Crompton, 1955; Hopson, 2014; Sidor & Hopson, 2018) and “upper central cusp”/“lower central cusp” of Hopson (2005)	
Central ridge (cri)	Labiolingually oriented crest centrally positioned on the mesial basin of a gomphodont postcanine (Fig. 2E)	
Central valley (cev)	Depression delimited by the lingual margin of the labial cusp and the labial margin of the lingual cusps on the transverse crest of the lower postcanine (Fig. 2G). The central valley corresponds to the “saddle” of Crompton (1972)	
Centrolabial ridge (clar)	Labiolingually oriented crest-like structure running on the labial surface of the central cusp, following the edge of the transverse crest, and connected to the labiocentral ridge of the labial cusp (Fig. 2F)	
Centrolingual ridge (clir)	Labiolingually oriented crest-like structure running on the lingual surface of the central cusp, following the edge of the transverse crest, and connected to the linguocentral ridge of the lingual cusp (Fig. 2F)	
Cingular cuspule (cic)	Small accessory cusp on a cingulum of a gomphodont or sectorial postcanine (Fig. 2D). Cingular cuspules are also referred as “cingular cusps” (Crompton, 1955; Abdala & Ribeiro, 2003; Abdala, Neveling & Welman, 2006)	
Cingulum (ci)	Bulge or shelf made of a succession of cingular cuspules on the rim of the occlusal surface of the gomphodont postcanine and on the basolingual or basolabial side of a sectorial tooth (Modified from Illiger, 1811; Owen, 1840; Sander, 1997c; Langer & Ferigolo, 2013; Fig. 2G). Cingula are also referred as “crenulations” by Seeley (1894)	
Concave surface (cos)	Apicobasally elongated concavity adjacent to the mesial and/or distal carinae on the labial and/or lingual surfaces of the crown in incisors and canines (Fig. 2B). The presence of concave surfaces on the lingual surface of the incisors and canines results in the salinon-shaped cross-sectional outlines of the crown, that is, an outline with both mesial and distal carinae facing linguomesially and linguodistally, respectively, subsymmetrical mesial and distal crown sides, a convex labial margin, and a biconcave lingual margin (Hendrickx, Mateus & Araújo, 2015, figure 5R)	
Conical postcanine (cpc)	Conidont tooth located in the mesialmost part of the postcanine tooth row (Figs. 2A, 2B and 3B). While Diademodon bears three or more conical postcanines (Fig. 3B), trirachodontids (Fig. 3) and some traversodontids (e.g., Boreogomphodon, possibly Andescynodon and Massetognathus) have one or two conical teeth, and the most derived traversodontids do not possess conical postcanine at all. Conical postcanine have also been known as “premolars” by 19th and 20th century authors (Broom, 1905, 1913; Brink, 1955; Grine, 1977; Gow, 1978)	
Crown (co)	Portion of the tooth covered with enamel, typically situated above the gum and protruding into the mouth (“couronne” of Fauchard, 1728; Cuvier, 1805; Schwenk, 2000; McGraw-Hill, 2003; Fig. 2I)	
Cusp (cu)	Pointed or rounded projection of dentine covered with enamel on the occlusal surface of a gomphodont postcanine or on the carina of sectorial teeth (Fig. 2G)	
Denticle (de)	An elaborate type of serration being formed by a projection of dentine covered with enamel along the carina of incisors, canines, and sectorial postcanines, as well as the crests of gomphodont postcanines (“dentelure” of Cuvier, 1805; Owen, 1840; McGraw-Hill, 2003; Brink & Reisz, 2014; Fig. 2J). The denticles are also referred as “denticulations” (Kammerer et al., 2012), and the large denticles in low number (<15) on the carinae are known as “mega-serrations”/“megaserrations” (Hopson, 1984), “cuspules” (Abdala & Ribeiro, 2003; Sidor & Hopson, 2018), “marginal cuspules” (Hopson, 1984; Sues, Hopson & Shubin, 1992; Abdala & Ribeiro, 2003; Abdala, Neveling & Welman, 2006; Liu & Abdala, 2014), “cusps” (Kammerer et al., 2012), “accessory cusp” and “posterior accessory cusp” (Ranivoharimanana et al., 2011; Kammerer et al., 2012)	
Diastema (dia)	Space separating the last upper incisor from the canine and the upper and/or lower canine from the first postcanine (Fig. 2B)	
Distal accessory cusp (dac, D or d)	Minor pointed or rounded projection of a sectorial postcanine, distal to the distal main cusp (Figs. 2I, 3Q and 3R). The distal accessory cusp is also referred as the “posterior cingular cusp” (Crompton, 1963; Abdala, Jasinoski & Fernandez, 2013), “posterior accessory cusp” (Abdala & Ribeiro, 2003; Abdala, Neveling & Welman, 2006; Sues & Hopson, 2010; Sidor & Hopson, 2018), “heel cusp” (Sues & Hopson, 2010), and “cuspule” (Liu & Sues, 2010). The distal accessory cusp of gomphodont is homologous to the “posterior cingular cusp” (sensu Crompton, 1963) D and d of the upper and lower postcanines, respectively, in Crompton & Jenkins’ (1968) notation	
Distal accessory ridge (dar)	Crest-like structure on the distal surface of the transverse crest, perpendicular, diagonally oriented or even parallel from the latter (Figs. 2H, 3A and 3N)	
Distal basin (db)	Main concavity distal to the transverse crest on the occlusal surface of a postcanine (Figs. 2C, 2D, 2G and 2H). The distal basin, which is typically delimited by the distal ridge/cingulum distally, is also known as the “posterior basin” (Crompton, 1972; Hopson, 1984, 2005, 2014; Godefroit & Battail, 1997; Sues, Olsen & Carter, 1999; Hopson & Sues, 2006; Liu & Sues, 2010; Sues & Hopson, 2010; Sidor & Hopson, 2018) and the “posterior valley” (Crompton, 1972; Godefroit & Battail, 1997; Hopson, 2005)	
Distal cingular cuspule (dcc)	Minor pointed or rounded projection on the distal cingulum (dci) of a gomphodont postcanine (Figs. 2G and 3G). Distal cingular cusps are also known as “posterior cingular cusps” (Hopson, 2005; Abdala, Hancox & Neveling, 2005; Abdala, Jasinoski & Fernandez, 2013) and “heel cusp” (Crompton, 1955)	
Distal cingulum (dci)	Labiolingually oriented row of distal cingular cuspules (dcc) distal to the transverse crest and typically delimiting the distal rim of the occlusal surface of a gomphodont postcanine (Figs. 2C, 2D and 2F). The distal cingulum is also known as the “posterior cingulum” (Crompton, 1972; Kemp, 1980; Hopson, 2005; Abdala, Hancox & Neveling, 2005; Liu & Sues, 2010; Sues & Hopson, 2010; Martinelli, 2010; Hendrickx, Abdala & Choiniere, 2016), “posterior marginal cingulum” (Hopson, 2005; Sidor & Hopson, 2018), “posterior cingular crest” (Abdala & Ribeiro, 2003; Kammerer et al., 2012), and “crenulated posterior ridge” (Crompton, 1955)	
Distal main cusp (dmc, C or c)	Largest pointed or rounded projection of a sectorial postcanine distal to the main cusp (Figs. 2I, 3P and 3S). The distal main cusp is also known as the “posterior cusp” (Hopson, 2005; Liu & Sues, 2010; Sidor & Hopson, 2018), and the “posterior accessory cusp” (Crompton, 1963; Abdala & Ribeiro, 2003; Abdala, Neveling & Welman, 2006; Sues & Hopson, 2010; Sidor & Hopson, 2018). The distal main cusp is homologous to the “posterior accessory cusp” (sensu Crompton, 1963) C and c of the upper and lower postcanines, respectively, in the notation proposed by Crompton & Jenkins (1968)	
Distal ridge (dri)	Labiolingually oriented crest-like structure distal to the transverse crest and delimiting the distal rim of the occlusal surface of a gomphodont postcanine (Fig. 2H). Also known as the “posterior wall” (Godefroit & Battail, 1997; Hopson & Sues, 2006; Hopson, 2014) and “posterior ridge” (Crompton, 1955; Godefroit & Battail, 1997)	
Distal valley (dv)	Depression delimited by the distal margin of the main cusp and the mesial margin of the distal main cusp on a sectorial postcanine (Fig. 2I)	
Enamel texture (ent)	Pattern of sculpturing on the crown surface at a sub-millimeter scale (Hendrickx, Mateus & Araújo, 2015; Fig. 2K). The enamel surface texture of incisors, canines, and postcanines of gomphodont cynodonts shows different patterns (Hendrickx, Mateus & Araújo, 2015, figure 6). A smooth enamel texture is here defined as the absence of enamel texture so that the crown surface does not show any irregularity. A non-oriented enamel texture with no pattern is referred as irregular. Finally, the enamel surface texture is called braided if the texture is oriented and made of alternating and interweaving grooves and short, moderately elongated or long sinuous ridges that are typically apicobasally oriented on the crown (Hendrickx, Mateus & Araújo, 2015; Fig. 2K). The term “crenulation” was used by Sidor & Hopson (2018) to describe the braided enamel texture seen on the canines of Cricodon	
Gomphodont postcanine (gpc)	Oval, quadrangular, subrectangular or subtriangular tooth in apical view allowing tooth-to-tooth occlusion (Figs. 2A, 2B and 3B). Upper gomphodont postcanines are typically labiolingually expanded whereas lower gomphodont postcanines can be labiolingually expanded, quadrangular or mesiodistally expanded	
Incisor (i)	Premaxillary or dentary tooth mesial to the canine and specialized for cutting (Figs. 2B and 3F)	
Labial cingular cuspule (lacc)	Minor pointed or rounded projection on the labial cingulum of a gomphodont and/or sectorial postcanine (Fig. 2C)	
Labial cingulum (laci)	Mesiodistally oriented row of accessory cuspules on the basolabial surface of a sectorial postcanine and/or delimiting the labial rim of the occlusal surface of a gomphodont postcanine (Fig. 2C). The labial cingulum is also known as the “external cingulum” (Crompton, 1972; Flynn et al., 2000; Abdala & Ribeiro, 2003; Abdala & Sa-Teixeira, 2004; Hopson, 2005, 2014; Abdala, Neveling & Welman, 2006; Melo, Martinelli & Soares, 2017), “buccal cingulum” (Sues, Olsen & Carter, 1999; Sues & Hopson, 2010; Melo, Martinelli & Soares, 2017), “crenulated ridge” (Crompton, 1955) and “cingular labial crest” (Martinelli, 2010)	
Labial cusp (lac)	Main labially positioned projection on the transverse crest of a gomphodont postcanine (Figs. 2C–2H and 3). The labial cusp is also know as the “external cusp” (Seeley, 1894; Crompton, 1972; Kemp, 1980; Flynn et al., 2000; Hopson, 2005, 2014; Hendrickx, Abdala & Choiniere, 2016; Sidor & Hopson, 2018), “upper external cusp”/“lower external cusp” (Hopson, 2005), “main external cusp” (Hopson, 2005, 2014; Sidor & Hopson, 2018), “main upper external cusp”/“main lower external cusp” (Hopson, 2014), “buccal cusp” (Grine, 1977; Sues, Hopson & Shubin, 1992; Sues, Olsen & Carter, 1999; Hopson & Sues, 2006), “buccal main cusp” (Melo, Abdala & Soares, 2015; Melo, Martinelli & Soares, 2017), “posterior buccal main cusp” (Melo, Abdala & Soares, 2015), “labial main cusp” (Godefroit & Battail, 1997; Abdala, Barberena & Dornelles, 2002) and “main labial cusp” (Chatterjee, 1982; Hopson, 1984, 2005, 2014; Godefroit & Battail, 1997; Abdala, Barberena & Dornelles, 2002; Abdala & Ribeiro, 2003; Abdala, Hancox & Neveling, 2005; Abdala & Smith, 2009; Martinelli, 2010; Ranivoharimanana et al., 2011; Kammerer et al., 2012; Liu & Abdala, 2014)	
Labial ridge (lar)	Mesiodistally oriented and labially positioned crest-like structure delimiting the labial rim of the occlusal surface of a lower gomphodont postcanine in Traversodontidae (Fig. 2D). The labial ridge is also referred as the “external ridge” (Seeley, 1895; Kemp, 1980; Hopson, 1984), “shearing ridge” (Hopson, 1984), “buccal ridge” (Sues, Olsen & Carter, 1999; Hopson & Sues, 2006; Liu & Sues, 2010), “posterior ridge” (Crompton, 1972), and “buccal longitudinal crest” (Hopson & Sues, 2006)	
Labial valley (lav)	Depression delimited by the lingual margin of the labial cusp and the labial margin of the central cusp on the transverse crest (Fig. 2F). The labial valley corresponds to the “embayment” of Crompton (1972) and the “V-shaped notch” of many authors (Crompton, 1955; Sues, Olsen & Carter, 1999; Sues & Hopson, 2010; Martinelli, 2010; Hopson, 2014)	
Labiocentral ridge (lacr)	Labiolingually oriented crest-like structure running on the lingual surface of the labial cusp, following the edge of the transverse crest, and connected to the centrolabial ridge of the central cusp (Figs. 2F and 2G)	
Labiodistal accessory cusp (ladc)	Minor pointed or rounded projection distal to the labial cusp and located on the labiodistal margin of the occlusal surface of a gomphodont postcanine (Figs. 2C, 2D and 2F–2H). The labiodistal accessory cusp is also known as the “posterior labial cusp” (Crompton, 1955; Liu, 2007; Martinelli, 2010; Liu & Abdala, 2014; Melo, Abdala & Soares, 2015), “posterolabial cusp” (Abdala & Ribeiro, 2003; Hopson, 2014), “posterior labial accessory cusp” (Kammerer et al., 2012), “posterior external accessory cusp” (Hopson, 2014), “posterior accessory labial cusp” (Abdala, Barberena & Dornelles, 2002; Abdala & Ribeiro, 2003; Abdala & Sa-Teixeira, 2004; Battail, 2005; Abdala, Hancox & Neveling, 2005; Kammerer et al., 2012), “posterior buccal cusp” (Liu & Sues, 2010; Melo, Abdala & Soares, 2015), “posterobuccal cusp” (Sues, Olsen & Carter, 1999; Hopson & Sues, 2006; Liu & Sues, 2010; Sues & Hopson, 2010), “posterior buccal accessory cusp” (Melo, Abdala & Soares, 2015), “posterolateral accessory cusp” (Hopson, 1985) and “posterior accessory cusp” (Crompton, 1972; Hopson, 1985)	
Labiodistal ridge (ladr)	Mesiodistally oriented crest-like structure running on the distal surface of the labial cusp and typically connected to the distal ridge/cingulum (Figs. 2C and 2F). The labiodistal ridge corresponds to the “posteroexternal ridge” of Hopson (2005) and Hendrickx, Abdala & Choiniere (2016)	
Labiomesial accessory cusp (lamc)	Main pointed or rounded projection mesial to the labial cusp and located on the labiomesial margin of the occlusal surface of a gomphodont postcanine (Fig. 2C). The labiomesial accessory cusp is also known as the “anterior accessory cusp of upper postcanine”/“anterior accessory cusp of lower postcanine” (Crompton, 1972), “anterior labial cusp” (Crompton, 1955; Liu, 2007; Martinelli, 2010), “anterolabial cusp” (Flynn et al., 2000; Liu, 2007; Hopson, 2014; Liu & Abdala, 2014), “anterolabial accessory cusp” (Hopson, 2014; Liu & Abdala, 2014; Melo, Martinelli & Soares, 2017), “anteroexternal accessory cusp” (Hopson, 2014), “anterior buccal cusp” (Melo, Abdala & Soares, 2015), “anterobuccal cusp” (Sues, Olsen & Carter, 1999; Hopson & Sues, 2006; Liu & Sues, 2010; Sues & Hopson, 2010), “anterior buccal accessory cusp” (Melo, 2014; Melo, Abdala & Soares, 2015), “mesiobuccal accessory cusp” (Melo, Martinelli & Soares, 2017), and “anterior accessory labial cusp” (Abdala, Barberena & Dornelles, 2002; Abdala & Ribeiro, 2003; Abdala & Sa-Teixeira, 2004; Battail, 2005; Abdala, Hancox & Neveling, 2005; Abdala & Smith, 2009; Ranivoharimanana et al., 2011; Kammerer et al., 2012)	
Labiomesial ridge (lamr)	Mesiodistally oriented crest-like structure running on the mesial surface of the labial cusp and typically connected to the mesial ridge/cingulum (Fig. 2C). The labiomesial ridge corresponds to the “anterior ridge” (Crompton, 1972; Hopson, 1985) and “anteroexternal ridge” (Hopson, 2005; Hendrickx, Abdala & Choiniere, 2016)	
Lingual cingular cuspule (licc)	Minor pointed or rounded projection on the lingual cingulum of a conical or sectorial postcanine (Figs. 2I and 3O). The lingual cingular cuspules are also known as the “lingual cingular cusps” (Sidor & Hopson, 2018)	
Lingual cingulum (lici)	Mesiodistally oriented row of cuspules on the basolingual surface of a sectorial postcanine and/or delimiting the lingual rim of the occlusal surface of a gomphodont postcanine (Figs. 2I and 3O)	
Lingual cusp (lic)	Main lingually positioned projection on the transverse crest of the postcanine (Figs. 2C–2H and 3). The lingual cusp is also known as the “internal cusp” (Romer, 1967; Kemp, 1980; Hopson, 1985), “main internal cusp” (Hopson, 2005, 2014; Sidor & Hopson, 2018), “upper internal cusp”/“lower internal cusp” (Hopson, 2005), “lingual main cusp” (Melo, Martinelli & Soares, 2017) and “main lingual cusp” (Crompton, 1955; Godefroit & Battail, 1997; Godefroit, 1999; Sues & Hopson, 2010; Melo, Martinelli & Soares, 2017)	
Lingual ridge (lir)	Mesiodistally oriented and lingually positioned crest-like structure delimiting the lingual rim of the occlusal surface of a gomphodont postcanine (Fig. 2E)	
Lingual valley (liv)	Depression delimited by the labial margin of the lingual cusp and the lingual margin of the central cusp on the transverse crest (Fig. 2F). The labial valley corresponds to the “embayment” of Godefroit & Battail (1997) and the “V-shaped notch” of Hopson & Sues (2006)	
Linguocentral ridge (licr)	Labiolingually oriented crest-like structure running on the labial surface of the lingual cusp, following the edge of the transverse crest, and connected to the centrolingual ridge of the central cusp (Fig. 2F)	
Linguodistal accessory cusp (lidc)	Minor pointed or rounded projection distal to the lingual cusp and located on the linguodistal margin of the occlusal surface of a gomphodont postcanine (Figs. 2C and 2F). The linguodistal accessory cusp is also known as the “posterior lingual cusp” (Crompton, 1955; Abdala, Barberena & Dornelles, 2002; Kammerer et al., 2012; Melo, Abdala & Soares, 2015), “posterolingual cusp” (Hopson & Sues, 2006; Liu, 2007), “posteromesial accessory cusp” (Hopson, 1985) and “posterior accessory lingual cusp” (Abdala & Ribeiro, 2003; Kammerer et al., 2012)	
Linguodistal ridge (lidr)	Mesiodistally oriented crest-like structure running on the distal surface of the lingual cusp and typically connected to the distal ridge/cingulum (Figs. 2C and 2F). The linguodistal ridge is equivalent to the “posterointernal ridge” of Hopson (2005)	
Linguomesial accessory cusp (limc)	Main pointed or rounded projection mesial to the lingual cusp and located on the linguomesial margin of the occlusal surface of a gomphodont postcanine (Figs. 2C and 2H). The linguomesial accessory cusp is also known as the “anterior accessory cusp of upper postcanine”/“anterior accessory cusp of lower postcanine” (Crompton, 1972), “anterior lingual cusp” (Crompton, 1955; Abdala & Smith, 2009; Martinelli, 2010; Melo, Abdala & Soares, 2015), “anterior lingual accessory cusp” (Melo, 2014), “anterolingual cusp” (Hopson, 1984; Sues, Hopson & Shubin, 1992; Abdala & Ribeiro, 2003; Abdala, Neveling & Welman, 2006; Gao et al., 2010; Sues & Hopson, 2010; Ranivoharimanana et al., 2011; Liu & Abdala, 2014), “anterolingual accessory cusp” (Sues & Hopson, 2010; Hopson, 2014; Liu & Abdala, 2014; Melo, Abdala & Soares, 2015; Sidor & Hopson, 2018) and “anterointernal accessory cusp” (Hopson, 2014)	
Linguomesial ridge (limr)	Mesiodistally oriented crest-like structure running on the mesial surface of the lingual cusp and typically connected to the mesial ridge/cingulum (Fig. 2C). The linguomesial ridge is equivalent to the “anterointernal ridge” of Hopson (2005)	
Longitudinal ridge (lri)	Apicobasally high and mesiodistally short convexity on the labial and/or lingual surface of incisors and/or canines (modified from Hendrickx, Mateus & Araújo, 2015; Fig. 2B). Longitudinal ridges are also known as “flutes”/“fluting” (Seeley, 1894; Crompton, 1955), “longitudinal striations” (Hopson, 1984), “vertical striations” (Liu & Sues, 2010), and “vertical ridges” (Sues & Hopson, 2010)	
Main cusp (mc, A or a)	Major projection of dentine covered with enamel on the sectorial postcanine (Figs. 2I, 3Q and 3S). The main cusp can be denticulated on both its mesial and distal carinae. The main cusp is also known as the “central cusp” (Liu & Sues, 2010; Sues & Hopson, 2010; Sidor & Hopson, 2018), and the “central main cusp” (Sidor & Hopson, 2018). The main cusp is homologous to the cusps A and a of the upper and lower postcanines, respectively, in Crompton & Jenkins’ (1968) notation	
Mesial accessory cusp (mac, E or e)	Minor pointed or rounded projection on the mesial carina of a sectorial postcanine, mesial, or linguomesial to the mesial main cusp. The mesial accessory cusp is homologous to the “anterior cingular cusp” (sensu Crompton, 1963) E and e of the upper and lower postcanines, respectively, in Crompton & Jenkins’ (1968) notation. Visible in some sectorial postcanines of Thrinaxodon and possibly Cynognathus, one or several mesial accessory cusps are assumed to be present in the multicuspid/crenulated lower sectorial teeth of juveniles Andescynodon (PVL 4390) and Massetognathus (MCZ 4267)	
Mesial accessory ridge (mar)	Crest-like structure on the mesial surface of the transverse crest, perpendicular, diagonally-oriented or parallel to the latter (Figs. 2H, 3A, 3C and 3N)	
Mesial basin (mb)	Main concavity mesial to the transverse crest on the occlusal surface of a gomphodont postcanine (Figs. 2C, 2E and 2H). The mesial basin is also known as the “anterior excavation in the crown” (Crompton, 1955), “anterior valley” (Crompton, 1972), “occlusal basin” (Melo, Martinelli & Soares, 2017) and “anterior basin” (Sues, Hopson & Shubin, 1992; Godefroit & Battail, 1997; Sues, Olsen & Carter, 1999; Hopson & Sues, 2006; Liu & Sues, 2010; Hopson, 2014)	
Mesial cingular cuspule (mcc)	Minor pointed or rounded projection on the mesial cingulum of a gomphodont postcanine (Figs. 2C, 2H and 3E)	
Mesial cingulum (mci)	Labiolingually oriented row of accessory cuspules mesial to the transverse crest and typically delimiting the mesial rim of the occlusal surface of a gomphodont postcanine (Fig. 2C). The mesial cingulum is also known as the “anterior cingulum” (Sues, Hopson & Shubin, 1992; Abdala & Sa-Teixeira, 2004; Battail, 2005; Hopson, 2005; Hopson & Sues, 2006; Abdala, Neveling & Welman, 2006; Kammerer et al., 2008; Gao et al., 2010; Sues & Hopson, 2010; Liu & Abdala, 2014; Hendrickx, Abdala & Choiniere, 2016), “crenulated ridge” (Kemp, 1980; Abdala, Neveling & Welman, 2006), “anterior crenulated ridge” (Crompton, 1955), “anterior marginal cingulum” (Hopson, 2005; Sidor & Hopson, 2018), “anterior external cingulum” (Hopson, 2005) and “anterior cingular crest” (Abdala & Ribeiro, 2003; Kammerer et al., 2012)	
Mesial main cusp (mmc, B or b)	Largest pointed or rounded projection on the mesial carina of a sectorial postcanine directly mesial to the main cusp (Figs. 2I, 3Q and 3S). The mesial main cusp, as used by Sidor & Hopson (2018), is also known as the “anterior cusp” (Liu & Sues, 2010; Sues & Hopson, 2010; Sidor & Hopson, 2018), the “anterior main cusp” (Abdala, Jasinoski & Fernandez, 2013), the “anterior accessory cusp” (Crompton, 1963; Abdala, Neveling & Welman, 2006; Sues & Hopson, 2010; Sidor & Hopson, 2018) and the “mesial accessory cusp” (Martinelli, Soares & Schwanke, 2016). The mesial main cusp is homologous to the “anterior accessory cusp” (sensu Crompton, 1963) B and b of the upper and lower postcanines, respectively, in Crompton & Jenkins’ (1968) notation	
Mesial ridge (mri)	Labiolingually oriented crest-like structure mesial to the transverse crest and typically delimiting the mesial rim of the occlusal surface of a gomphodont postcanine (Figs. 2C and 2E). Also known as “transverse ridge” (Seeley, 1894), “anterior wall” (Hopson, 1985; Sues, Olsen & Carter, 1999; Flynn et al., 2000; Battail, 2005; Martinelli, 2010; Melo, Abdala & Soares, 2015; Melo, Martinelli & Soares, 2017) and “anterior crest” (Melo, Abdala & Soares, 2015)	
Mesial valley (mv)	Depression delimited by the mesial margin of the main cusp and the distal margin of the mesial main cusp on a sectorial postcanine (Fig. 2I)	
Postcanine (pc)	Maxillary or dentary tooth positioned distal to the canine. Postcanines include conical, gomphodont and sectorial teeth, which can change (e.g., Diademodon) from one morphology to the other (Fig. 2B)	
Root (ro)	Portion of the tooth beneath the gum and embedded in an alveolus (“racine” of Fauchard, 1728; Cuvier, 1805; “radix dentis” of Illiger, 1811; Owen, 1840; Hillson, 2005; Fig. 2I)	
Sectorial postcanine (spc)	Labiolingually compressed tooth typically located in the distalmost part of the postcanine tooth row, distal to the gomphodont teeth, and more rarely in the anteriormost part of the postcanine tooth row, and adapted for cutting in a shearing manner. Sectorial postcanines typically include a main cusp and often one or several accessory cusps mesial and/or distal to the main cusp (Figs. 2A, 2B, 3F and 3H)	
Serration (se)	A projection along a ridge or keel-like structure of a tooth, whether composed of enamel or by both enamel and dentine (modified from Brink & Reisz, 2014; Fig. 2J). Unlike the carinae of incisors, canines, and sectorial teeth which bear large and elaborate serrations (i.e., denticles), the serrated transverse crest and mesial and distal ridges of gomphodont postcanines only have simple and minute serrations visible with a microscope. Besides the well-delimited denticles of non-gomphodont postcanines, these serrations should not be confused with the triangular, hemi-spherical or sub-pyramidal cusps and cuspules present on the transverse crest and/or cingula of gomphodont and sectorial postcanines of some taxa	
Shouldering (sho)	Extension of the labiomesial margin of the upper postcanine forward, producing a “shoulder-like” process over the preceding postcanine (modified from Romer, 1967; Abdala & Ribeiro, 2003: figure 10C, D)	
Transitional postcanine (tpc)	Labiolingually expanded sectorial postcanine sharing an intermediate morphology between a gomphodont tooth and a sectorial postcanine (Figs. 2A, 3B, 3D and 3O). Transitional postcanines are typically formed by a recurved blade shape labial portion and a relatively flat lingual projection (Goñi & Goin, 1988). They are also referred as “intermediate gomphodont” (Fourie, 1963; Osborn, 1974; Grine, 1977; Goñi & Goin, 1988), “intermediate sectorial” (Osborn, 1974; Goñi & Goin, 1988), “semi-gomphodont” (Hopson, 1971; Crompton, 1972; Brink, 1977), “sub-gomphodont” (Hopson, 1964) and “sub-sectorial” (Martinelli, 2010) postcanines	
Transverse crest (tc)	Main labiolingually oriented ridge on the occlusal surface of the gomphodont postcanine and bearing the labial, lingual and often the central cusps (Figs. 2C–2E and 2H). The transverse crest is also known as the “transverse ridge” (Crompton, 1972; Hopson, 1984, 2005, 2014; Godefroit, 1999; Abdala, Neveling & Welman, 2006; Sues & Hopson, 2010; Sidor & Hopson, 2018), “anterior ridge” (Martinelli, 2010), “transverse anterior ridge” (Hopson & Sues, 2006), “anterior crest” (Liu & Abdala, 2014), “central crest” (Hendrickx, Abdala & Choiniere, 2016; Sidor & Hopson, 2018) and “posterior transverse crest” (Abdala, Barberena & Dornelles, 2002; Abdala & Ribeiro, 2003; Melo, Abdala & Soares, 2015)	
Transverse undulation (tun)	Band like enamel wrinkle extending along most of the incisor or canine length, typically from the mesial to distal carinae, perpendicular from the long axis of the crown (modified from Hendrickx, Mateus & Araújo, 2015; Fig. 2B)	

Figure 3 Dentition of non-traversodontid Gomphodontia.

(A) Upper gomphodont postcanine (SAM-PK-571a); and (B) cranial dentition (mainly based on BSP 1934 VIII 14) of Diademodon tetragonus in apical and palatal views, respectively; (C) last right upper gomphodont postcanine; and (D) cranial dentition of Titanogomphodon crassus (GSN R322) in apical and palatal views, respectively; (E) upper gomphodont postcanine (reconstruction based on the fourth and third left upper gomphodont teeth of NMQR 3251 and 3255, respectively); and (F) cranial dentition (NMQR 3255) of Langbergia modisei in apical and palatal views, respectively; (G) last right upper gomphodont postcanine (UMCZ T905); and (H) cranial dentition (reconstruction based on BP/1/6102 and UMCZ T905 for the anterior and posterior portions of the cranium, respectively, and NHCC LB28 for the anterior postcanine dentition) of Cricodon metabolus in apical and palatal views, respectively (the dashed-line represents the lateral margin of the cranium of NHCC LB28); (I and J) antepenultimate right upper gomphodont postcanines (NHM PV R3307, Trirachodon “kannemeyeri” morphotype, and BSP 1934 VIII 21, Trirachodon berryi morphotype, for (I) and (J), respectively); and (K) cranial dentition (BP/1/4658) of Trirachodon berryi in apical and palatal views, respectively; (L) fourth right upper gomphodont postcanine; and (M) cranial dentition of Beishanodon youngi (PKUP V3007) in apical and palatal views, respectively; (N) lower gomphodont postcanine (left anterior tooth of SAM-PK-571b); and (O) mandibular dentition (reconstruction based on MBR 1004 for the mandible, incisor and canine morphology, SAM-PK-571b and AM 3753 for the anterior postcanine dentition, and SAM-PK-K177 for the posterior postcanine dentition) of Diademodon tetragonus in apical and dorsal views, respectively; (P–S) upper sectorial postcanines of (P) Diademodon tetragonus (MB R1004); (Q) Langbergia modisei (NMQR 3251); (R) Trirachodon berryi (SAM-PK-4801); and (S) Cricodon metabolus (UMCZ T905) in labial view; (T) lower gomphodont postcanine (third right tooth of NMQR 5251); and (U) mandibular dentition (NMQR 3251) of Langbergia modisei in apical and dorsal views, respectively; (V) lower gomphodont postcanine (last right tooth of UMCZ T905); and (W) mandibular dentition (based on SAM-PK-5881a and UMCZ T905 for the anterior and posterior dentitions, respectively) of Cricodon metabolus in apical and dorsal views, respectively; (X) lower gomphodont postcanine (antepenultimate right gomphodont tooth of SAM-PK-K4801); and (Y) mandibular dentition (BP/1/4658) of Trirachodon berryi in apical and dorsal views, respectively; (Z) lower gomphodont postcanine (sixth and antepenultimate right gomphodont tooth of SAM-PK-K171); and (AA) mandibular dentition (SAM-PK-K171) of Trirachodon berryi (T. “kannemeyeri” morphotype) in apical and dorsal views, respectively. Abbreviations: c, canine (in orange); cpc, conical postcanine (in yellow); dac, distal accessory cusp; dar, distal accessory ridge; dcc, distal cingular cuspule; gpc, gomphodont postcanine (in violet); i, incisor (in blue); licc, lingual cingular cuspule; lici, lingual cingulum; mar, mesial accessory ridge; mc, main cusp; mcc, mesial cingular cuspule; mmc, mesial main cusp; pcf, postcanine fossa; spc, sectorial postcanine (in green); tpc, transitional postcanine (in red).

Results

Diademodontidae Haughton, 1924

Diademodon tetragonusSeeley, 1894

Holotype: SAM-PK-571a (Wonderboom, subzone B, Cynognathus AZ, South Africa), two isolated upper postcanines (holotype of Diademodon brachytiara; Seeley, 1894); SAM-PK-571b (Burghersdorp, subzone B, Cynognathus AZ, South Africa), two isolated canines, an incomplete upper jaw and a small portion of mandible (holotype of D. tetragonus; Seeley, 1894).

Referred dental material: from Burghersdorp, subzone B, Cynognathus AZ, South Africa: AM 438, 458 (holotype of Gomphognathus kannemeyeri), BP/1/2522 (Luiperdskop locality), 3769, 3771–3773, 3776 (holotype of Cragievarus kitchingi; Cragievar locality for the four last specimens), NHMUK PV R2574, R2575, R3581 (holotype of Microgomphodon eumerus), SAM-PK-3426, K175, K177, K180, K183, K9968, UMCZ T.433 (Luiper Kop locality); from Steynsburg, subzone B, Cynognathus AZ, South Africa, AM 3753 (holotype of Octagomphus woodi); from Winnaars Baken, subzone B, Cynognathus AZ, South Africa: AMNH FR 5518 and 5519 (holotype of Cyclogomphodon platyrhinus), BP/1/3511, NHMUK PV R3587, R3588 (referred to D. browni), R3765 (holotype of D. entomophonus), R4092, R9216, SAM-PK-8015, 11265; from Aliwal North, subzone B, Cynognathus AZ, South Africa: NHMUK PV R3303 (holotype of D. mastacus), R3304 (holotype of D. browni), R3305 (holotype of M. oligocynus), R3308, R3724, SAM-PK-4002, 5877 (referred to Cyclogomphodon platyrhinus), 6216, 6218, 6219; from Queenstown, subzone B or C, Cynognathus AZ, South Africa: SAM-PK-?K4660, ?K4661; from Hofmeyr, subzone B or C, Cynognathus AZ, South Africa: SAM-PK-K5266; from Lady Frere, subzone B, Cynognathus AZ, South Africa: BP/1/1195, 3754, 3756–3758, BSP 1934 VIII 14, 15, 16, 17 (holotype of G. grossarthi), 18 (holotype of G. broomi), 19 (holotype of G. haughtoni), 20, 505, BSP 1936 II 8 (holotype of Sysphinctostoma smithi), MB R1004, NHMUK PV R2576–7 (holotype of G. polyphagus), R2578, UMCZ T.434, T.436 (holotype of D. laticeps), T.438, T.441, T.445, T.454, T.826, T.971; from Rouxville, subzone A, Cynognathus AZ, South Africa: BP/1/4529, 4647 (Bethel/Slootkraal localities for the two last), 4669 (Gladde Grond 530 locality), 4677 (Betjieskraal 36 locality); from Norwood farm, subzone C, Cynognathus AZ, South Africa: BP/1/5541; from Avilion farm, subzone C, Cynognathus AZ, South Africa: BP/1/5542; from unknown localities in South Africa: NHMUK PV R3767; SAM-PK-K5223 (Cape Province), K5716, K8971, K9969; from the Luangwa Basin, Ntawere Formation, Zambia: BP/1/3639 (holotype of D. rhodesiensis); from Etjo Mountain, Omingonde Formation, Namibia: GSN R321, R327, R335, RK3; from San Rafael, Río Seco de la Quebrada Formation, Argentina: MHNSR–Pv 357; from Beardmore Glacier region, Fremouw Formation, Antarctica: possibly AMNH FR 24421.

Occurrence: Wonderboom, Burghersdorp, Aliwal North, Steynsburg, Winnaars Baken, Queenstown and Hofmeyr, Joe Gqabi District, Walter Sisulu Municipality, Eastern Cape Province, South Africa; Avilion and Norwood farms, Chris Hani District, Enoch Mgijima Municipality, Eastern Cape Province, South Africa; Lady Frere, Chris Hani District; Rouxville, Xhariep District, Mohokare Municipality; Free State, South Africa; Drysdall and Kitching’s locality 16, north of Sitwe, northern Luangwa Basin, Zambia (Peecook et al., 2018); Etjo Mountain, Otjozondjupa Region, Namibia; Puesto Viejo farm house, 40 km southwest of San Rafael, Mendoza Province, Argentina; possibly Gordon Valley, Beardmore Glacier region, Transantarctic Mountains, Antarctica.

Horizon: subzones B–C of the Cynognathus AZ, Burgersdorp Formation, Karoo Basin; lower Ntawere Formation; lower fauna of the upper Omingonde Formation; Río Seco de la Quebrada Formation, upper unit of the Puesto Viejo Group; possibly upper Fremouw Formation.

Age: early to late Anisian, Middle Triassic; n.b, Valencio, Mendía & Vilas (1975) provided a 40K/40Ar dating of 232 ± 10 Ma for the middle section of the Río Seco de la Quebrada Formation of Argentina, placing this unit into the Upper Triassic (Carnian; Fig. 1B). Likewise, Ottone et al. (2014) obtained a 38U/206Pb age of 235.8 ± 2.0 Ma for the underlying Quebrada de los Fósiles Formation (Fig. 1B), providing additional support to the fact that the Río Seco de la Quebrada Formation is early Carnian. This unit, which has yielded remains of the basal cynognathians Cynognathus (Abdala, 1996, 1999) and Diademodon (Martinelli, De La Fuente & Abdala, 2009), would thus be at least 10 million years younger than the putative age usually attributed to the African Cynognathus AZ (Martinelli et al., 2017; Peecook et al., 2018; Gaetano, Mocke & Abdala, 2018, and references therein). Ottone et al. (2014) suggested two scenarios to explain their results: (i) the Cynognathus AZ and other fossiliferous units from African basins (i.e., Omingonde Formation of Namibia, Manda Beds of Tanzania, Ntawere Formation of Zambia) are wrongly attributed to the Anisian and should be instead assigned to the Carnian (Fig. 1B, in green); (ii) an Anisian age of the Cynognathus AZ and other contemporaneous biostratigraphic units from Southern Africa is correct and the stratigraphic duration of Cynognathus and Diademodon is much longer than expected, ranging from the Anisian (Africa) to the Carnian (South America).

Dental formula: i4/3 : c1/1 : pc7-16/7-13 (cpc1-6 : gpc2-7 : tpc0-2 : spc1-4).

Dental morphology: The dental morphology, postcanine microstructure, dental replacement pattern, and postcanine occlusion of D. tetragonus are fairly well-known (Seeley, 1894, 1895, 1908; Watson, 1911, 1913; Broili & Schröder, 1935; Brink, 1955, 1963, 1977; Crompton, 1955, 1963, 1972; Fourie, 1963; Ziegler, 1969; Hopson, 1971; Osborn, 1974; Grine, 1977). Few of these studies provide, however, detailed information and illustrations on the dental morphology and a thorough description of the anatomy of the incisors, canines, and postcanines is here provided.

Incisors

Little information on the upper and lower incisors is available in the Diademodon specimens examined first hand. Incisors are preserved in the skulls of BP/1/3756 and BP/1/4669, the crania of AM 3753, BP/1/2522 and BSP 1934 VIII 14, and the mandible of MB R1004. The incisors of the latter specimen are, however, the only ones in natural position and visible in lingual view. Only the labial side of the teeth is visible in the other specimens, except for BSP 1934 VIII 14, in which three loose upper incisors are preserved and show their labial or lingual sides. As most trirachodontids, there are four upper and three lower incisors in Diademodon (Brink, 1955). The upper incisors of BSP 1934 VIII 14 appear to be salinon-shaped in cross-section at mid-crown and both mesial and distal carinae are denticulated. The distal carina faces distally whereas the mesial carina is linguomesially displaced and almost reaches the cervix (Figs. 4A and 4C). The denticles are minute, well-delimited, apicobasally elongated to subquadrangular, and their external margin, made of a thin layer of enamel, is weakly convex (Figs. 4B and 4D). There are eight and six denticles per millimeter on the mesial and distal carinae at mid-crown in BSP 1934 VIII 14, respectively. This suggests that the distal denticles of the incisors are larger than the mesial ones (DSDI >1.2) in Diademodon. Concave surfaces adjacent to the mesial carina are present on the lingual surface and marginal to the distal carina on the labial side of the crown (Fig. 4A). The lower incisors of MB R1004 are too badly preserved. They are mildly procumbent and no constriction appears to be present between root and crown. Their morphology does not seem to depart from that of the upper incisors.

Figure 4 Dentition of Diademodon tetragonus.

(A) Isolated upper incisor (first left incisor?) of BSP 1934 VIII 14, with (B) close up on the distal denticles, in labial view; (C) Isolated upper incisor (first or second right incisor?) of BSP 1934 VIII 14, with (D) close up on the mesial denticles, in lingual view; (E) Isolated canine of SAM-PK-571b, with (F) close up on the distal denticles, in lingual? view; (G) First to third right upper conical postcanines of MB R1004 in labial view; (H) Fifth right upper conical postcanine of BSP 1934 VIII 14 in apicolabial view; (I and J) Third and fourth left lower conical postcanines of AM 458 in (I), labial and (J) lingual views; (K) Second to fourth left lower conical postcanines of SAM-PK-5877 in labial view; (L and M) Isolated upper gomphodont postcanine of SAM-PK-571 in (L), apical and (M), mesial views; (N and O) Second and third right upper gomphodont postcanines of SAM-PK-571 in (N) apical and (O) distal view; (P) Fourth right upper gomphodont postcanine of BSP 1934 VIII 14 in apical view. Abbreviations: cc, central cusp; cic, cingular cuspules; dac, distal accessory cusp; dar, distal accessory ridge; dca, distal carina; dcc, distal cingular cuspule; dci, distal cingulum; dde, distal denticle; lac, labial cusp; ladc, labiodistal accessory cusp; lic, lingual cusp; limc, linguomesial accessory cusp; lri, longitudinal ridge; mar, mesial accessory ridge; mca, mesial carina; mcc, mesial cingular cuspule; mci, mesial cingulum; mde, mesial denticle; tc, transverse crest.

Canines

Only the two partially complete isolated canines of the holotypic specimen SAM-PK-571b can be examined in all views. The upper canines of AM 3753, BP/1/3756, BP/1/4669, and BSP 1934 VIII 14 and 15 are relatively well-preserved but BSP 1934 VIII 15 is the only specimen with in situ upper canines that can be seen in both labial and lingual views. Likewise, only the lower canines of BSP 1934 VIII 505 can be observed in both labial and lingual views, yet they are poorly preserved and incomplete. The isolated canines of the holotype bear two poorly defined longitudinal ridges on both labial and lingual sides (Fig. 4E). These ridges extend along the whole crown and delimit narrow labial and lingual depressions mesiodistally that extend also onto the root. Although the canines of several specimens of Diademodon (e.g., BSP 1934 VIII 14, 15 and 505; BP/1/4669) do not bear any ridges, three to four faint longitudinal ridges can be seen on the canines of AM 3753 and BP/1/3756. Interestingly, SAM-PK-571b, AM 3753, and BP/1/3756 with ridged canines belong to small individuals whereas large size specimens of Diademodon have canines devoid of longitudinal ridges. This strongly suggests that the presence of ridges varies throughout ontogeny, as observed in the canines of some traversodontids (A. Martinelli, April 2018, personal communication). The mesial and distal carinae of the upper canines are centrally positioned on the crown and the cross-sectional outline at mid-crown is lenticular in AM 3753. Both carinae are also denticulated and the mesial carina extends well-apical to the cervix in SAM-PK-571b. The denticles are well-preserved in this specimen and clearly show the peculiar condition of changing sporadically in size along the carinae (Fig. 4F). As in the incisors, the denticles of the canines are apicobasally elongated in shape and their external margin is weakly symmetrically convex. Seven and 7.5 denticles per millimeter are present at mid-crown in the canine of the holotype (CH ∼ 10 mm), whereas three and 3.5 to four denticles per millimeter can be counted at mid-crown in the canines of the larger specimens BSP 1934 VIII 14 (CH ∼ 14 mm) and BP/1/4669 (CH ∼ 19 mm), respectively. Due to preservation, it is unknown whether the lower canines of BSP 1934 VIII 505 were denticulated.

Conical postcanines

The upper and lower conical postcanines vary from one in the newly born/juvenile specimen BSP 1936 II 8 (Broili & Schröder, 1936; Hopson, 1971; Brink, 1977) to six in the large-sized and most probably adult specimen NHMUK PV R3308 (Hopson, 1971). As described by Crompton (1955), the upper conical postcanines are straight, apically pointed, and slightly labiolingually compressed or subcircular. The mesial and distal carinae of these teeth are also denticulated in NHMUK PV R3308 (Crompton, 1955) and denticles appear to be present at the base of the distal carina in BSP 1934 VIII 14 (Fig. 4H). The upper conical teeth are, however, unserrated in BP/1/3639 and possibly in MB R1004 (Fig. 4G). In the latter specimen, the upper conical postcanines are also mesiodistally constricted at the cervix but this may result from taphonomical deformation (Fig. 4G). The upper conical postcanines of NHMUK PV R3308 clearly increase in width distally, and the mesial four (possibly also the unpreserved fifth) crowns are monocuspid (i.e., tooth strictly conical in shape, with a single centrally positioned cusp; Crompton, 1955). The sixth postcanine, considered to be a conical type by Hopson (1971), departs from this morphology. Although subcircular in outline, the tooth shows an important lingual projection and includes a high labial cusp and a low lingual cingulum. The latter is comprised of three or more cingular cuspules linguomesially and an additional cingular cuspule behind the labial cusp distally (Crompton, 1955).

As in the upper jaw, the lower conical postcanines increase in width distally, being weakly labiolingually compressed in the first two or three teeth and subcircular in the following crowns. No information on the first two conical teeth could be extracted from the specimens. These lower postcanines are believed to be monocuspid like those from the cranium, and do not show any cingulum. The third and fourth lower conical postcanines preserved in AM 458 are proportionally wider than the upper conical postcanines of MB R1004 and BSP 1934 VIII 14 from the same alveoli. In AM 458, the mesial carina of conical postcanines faces linguomesially whereas the distal carina is positioned distally (Figs. 4I and 4J). Both mesial and distal carinae are clearly denticulated in AM 458. The distal denticles are large and apically inclined whereas the mesial denticles are low and show a widely convex external margin. Cingular cuspules are visible on the linguodistal surface, at the base of the third and fourth conical postcanines of AM 458 (Fig. 4J). A distal accessory cusp is also present at mid-crown height in the lower conical teeth of SAM-PK-5877 (Fig. 4K).

Upper gomphodont postcanines

According to Brink (1977), the number of upper gomphodont postcanines varies between two (BSP 1936 II 8; Broili & Schröder, 1936; Hopson, 1971) to seven (BP/1/2522; Brink, 1963). The best-preserved upper gomphodont postcanines are from the holotypic specimen SAM-PK-571a, which includes in situ teeth within the maxillae (Figs. 4N and 4O) as well as two well-preserved isolated postcanines from the more distal portion of the jaw (Figs. 4L and 4M). With a CBR ranging from one to 1.7, most of the upper gomphodont postcanines are wider than the lower gomphodont postcanines. Gomphodont postcanines of Diademodon are characterized by the presence of several accessory ridges and bumps on the mesial and distal surfaces of the transverse crest. The mesial and distal accessory ridges vary in length, orientation, and extension along the tooth row so that each gomphodont postcanine has a unique morphology along the jaw. All upper postcanines, however, bear a large labial cusp longer and slightly higher than the lingual cusp (Figs. 4L and 4N). A smaller and lower yet well-demarked labiodistal accessory cusp always follows the labial cusp. The mesial and distal crests of the labial cusp appear to be weakly denticulated, with the largest denticles being present in the basal portion of the cusp. The lingual cusp is always adjacent to a linguomesially accessory cusp, and the latter is as long and tall as the lingual cusp. In the fourth right gomphodont postcanine of BSP 1934 VIII 14, the linguomesial accessory cusp is in fact much longer than the lingual cusp (Fig. 4P) and is followed by a second shorter linguomesial accessory cusp. The presence of accessory ridges and bumps on the transverse crest makes it difficult to delimit the central cusp in SAM-PK-571a, which seems to be centrally positioned on the crest (Figs. 4L and 4M). Both mesial and distal margins of the upper gomphodont postcanines are delimited by one or several accessory cusps forming a cingulum (Fig. 4L). The size, position, and height of these cingular cuspules vary in each upper postcanine along the tooth row (see below). The isolated upper postcanines of the holotype bear a single mesial cingular cuspule (Fig. 4M) and perhaps some other minor cingular cuspules, whereas the distal cingulum is made of four to five cuspules increasing in height and width lingually (not clearly visible in Fig. 4L, see 3D model). The distal basin is longer than but as deep as the mesial basin. The root is more than twice as long as the crown and bears a deep, wide and centrally positioned depression on its distal surface.

The upper gomphodont postcanines from the anterior half of the maxilla (Figs. 4N and 4O) follow the same pattern of ridges and cusps seen in the isolated upper postcanines, that is, they have two main labial and lingual cusps adjacent to two prominent labiodistal and linguomesial cingular cuspules, respectively. The labial and lingual cusps have, however, the same height. The labiodistal accessory cusp is significantly lower and shorter than the labial cusp, whereas the linguomesial accessory cusp is longer and as high or higher than the lingual cusp. The transverse crest of these postcanines bears accessory ridges and bumps on the mesial surface only, the distal surface being smooth (Fig. 4N). In one of the two in situ upper postcanines, the mesial surface shows an accessory ridge running perpendicular to the transverse crest and parallel to a second, poorly marked ridge, as well as a single accessory bump on its linguomesial portion. The central cusp of the transverse crest is long, low and appears to bear some poorly defined serrations on its labial and lingual crests. Some portions of the mesial and distal cingula and labiomesial, labiodistal, linguomesial, and linguodistal ridges also show a beaded appearance and are finely serrated in the three best-preserved upper postcanines. The distal basin of the distalmost preserved postcanine is narrow, well defined and bounded by the distal cingulum. This basin is, however, shallow and poorly defined in the preceding tooth (Fig. 4N) and absent in the mesialmost preserved postcanine. The mesial basin of the two first postcanines has also been worn out, a feature shared with Titanogomphodon. Similar to the isolated upper postcanines, the mesial margins of the in situ postcanines have a single and weakly lingually deflected cingular cuspule. Likewise, the distal cingulum is formed by four cingular cuspules that increase in width and height lingually. Two distal cingular cuspules are seen on the linguodistal margin of the preceding tooth (Fig. 4N). A higher number of cingular cuspules can, however, been counted on the cingula of other Diademodon specimens, with five, possibly six, cingular cuspules in the mesial cingulum of the best-preserved right upper postcanine of NHMUK PV R3303, and five to six cingular cuspules in the distal cingulum of the largest left and right upper postcanines of NHMUK PV R3765. Although the width and height of the mesial cingular cuspules also increase lingually in NHMUK PV R3303, the cingular cups of the distal carina decrease in size either toward the center of the cingulum or sporadically along the cingulum in NHMUK PV R3765.

Lower gomphodont postcanines

The number of lower gomphodont postcanines appears to vary between three (juvenile specimen BSP 1936 II 8; Broili & Schröder, 1936; Hopson, 1971) to five (large-sized/adult specimens BSP 1934 VIII 505 and SAM-PK-177), or even possibly more. Two lower gomphodont postcanines from the posterior portion of an incomplete left mandible of the holotype SAM-PK-571a are also among the best-preserved lower postcanines known (Figs. 5A–5E). The best-preserved lower postcanine, which is the most distal one, is longer than wide and the lingual cusp is as wide but slightly shorter and lower than the labial cusp (Figs. 5A and 5C). It is difficult to know whether accessory cusps were present. Based on the size of the cusps, the linguomesial and labiodistal accessory cusps appear to be present and may not correspond to cingular cuspules. These two cusps are lower and shorter than the lingual and labial cusps (Figs. 5A and 5B). It is unknown whether the lingualmost cingular cuspule of the mesial cingulum is homologous to the linguomesial accessory cusp (Figs. 5A and 5B). The mesial and distal accessory ridges are prominent, higher than the low and poorly delimited transverse crest and diagonally oriented from it. Three accessory ridges can be counted on the mesial surface of the transverse crest and one on the distal surface (Fig. 5A). A second much shorter distal ridge extending linguodistally is present at the level of the distal basin. Three protuberant and well-delimited mesial cingular cuspules composed the mesial cingulum whereas the distal cingulum encompasses four smaller and poorly delimited cingular cuspules (Figs. 5A and 5C). As observed in some upper postcanines, the mesial and distal edges of the labial cusp are denticulated (Fig. 5E). The second more mesial lower gomphodont postcanine is more sub-circular in outline and the mesial and distal accessory ridges on the transverse crest are not so prominent. We assume the presence of a labiodistal and linguomesial accessory cusps in this tooth, but the mesialmost and distalmost cingular cuspules are as wide and as well-delimited as these two putative accessory cusps. Both labial and lingual cusps are incomplete and it is unknown whether they had the same width and height.

Figure 5 Dentition of Diademodon tetragonus.

(A–C and E) Lower gomphodont postcanine from the posterior portion of the mandible of SAM-PK-571b in (A) apical, (B) labial, (C) distal views, and (E) close up on the labial cusp of the distal postcanine; (D) distalmost right lower gomphodont postcanine of SAM-PK-5877 in apical view; (F) Sixth? gomphodont, first and second transitional, and first sectorial postcanines from the left mandible of SAM-PK-K177 in apical view; (G) First transitional and first sectorial postcanines of the right mandible of MB R1004 in apical view; (H) Transitional, first and second left upper sectorial postcanines of MB R1004 in labial view (photo credit: Christian Kammerer); (I–K) Second right upper sectorial postcanine of BP/1/4529, with close up on (J) the distal main cusp, and (K) the mesial denticles, in labial view. Abbreviations: cc, central cusp; dac, distal accessory cusp; dar, distal accessory ridge; dcc, distal cingular cuspule; dci, distal cingulum; dde, distal denticle; de, denticle; dmc, distal main cusp; lac, labial cusp; ladc, labiodistal accessory cusp; lic, lingual cusp; licc, lingual cingular cuspule; limc, linguomesial accessory cusp; lir, lingual ridge; mar, mesial accessory ridge; mc, main cusp; mcc, mesial cingular cuspule; mci, mesial cingulum; mde, mesial denticle; mmc, mesial main cusp.

The lower gomphodont postcanines of the large majority of other more mature specimens of Diademodon preserving the lower dentition (e.g., BMNH R3588; BSP 1934 VIII 16, 505; MB R1004; GSN R321) are worn out and only the distalmost mandibular postcanines of AM 458, AM 3753, and SAM-PK-5877 are complete enough to provide some information on their morphology. In all specimens, the labial cusp is the largest, and is significantly longer, wider, and higher than the lingual cusp. The transverse crest connecting the two cusps forms a narrow ridge bearing a low and poorly defined central cusp. One or two large pointed mesial cingular cuspules are visible in the three specimens, mesial to the lingual cusp (Fig. 5D). Based on their size, these cingular cuspules may correspond to the linguomesial accessory cusps of more derived gomphodonts. A wide and high centrally positioned mesial cingular cuspules is well-visible in SAM-PK-5877 (Fig. 5D) but worn out in AM 458 and AM 3753. The central cingular cuspules are wider than the mesial cuspules directly mesial to the lingual cusp but narrower than the preceding one on the linguomesial margin of the lower postcanine. The distal cingulum, well-preserved in the left and right distalmost lower postcanines of SAM-PK-5877, bears four to six well-delimited cingular cuspules (Fig. 5D). These distal cingular cuspules extend on the labiodistal ridge of the labial cusp in the left postcanine where they increase in length mesially. As in the holotype, several accessory ridges of variable length and orientation (i.e., perpendicular, diagonally oriented or parallel to the transverse crest) are present on the mesial and distal surfaces of the transverse crest in SAM-PK-5877, AM 458 and AM 3753. In SAM-PK-5877, one of the mesial accessory ridges reaches the largest mesial cingular cuspules (Fig. 5D).

Transitional postcanines

The transitional upper and lower postcanines, which are relatively well preserved in MB R1004, SAM-PK-K177, and AM 3753, have an intermediate morphology between the distalmost gomphodont and the sectorial postcanines. They are subcircular, weakly labiolingually elongated, drop-shaped or oval in outline. These teeth show a large recurved cusp on the labial portion of the crown homologous to the labial cusp and the main cusp of gomphodont and sectorial postcanines, respectively. A lingual cingulum formed by three to five well-delimited and strongly protruding cingular cuspules are also visible (Figs. 5F and 5G). Labiodistally oriented ridges connecting the main cusp to one or several of these cingular cuspules can be seen in some specimens (Fig. 5F). One or two distal accessory cusps are also present distal to the main cusp, with the one directly mesial to the main cusp being likely homologous to the labiodistal accessory cusp of gomphodont postcanines of Diademodon and possibly other trirachodontids and traversodontids. Two transitional postcanines can be observed in the lower jaw of SAM-PK-K177 (Fig. 5F). Although incomplete, the main cusp of those two teeth appears to be linguomesially oriented and subcircular in outline. The transitional upper postcanines (the intermediate gomphodont of Osborn, 1974) were lost in some specimens of Diademodon such as BSP 1934 VIII 14 in which they probably were replaced by gomphodont postcanines, following the replacement model proposed by Hopson (1971) and Osborn (1974).

Sectorial postcanines

The upper and lower sectorial postcanines always bear a recurved main cusp followed by a distal main cusp, or, more commonly, one distal main cusp and a smaller distal accessory cusp (Figs. 5H and 5I). The distal accessory cusp of the sectorial teeth is either of the same length or slightly shorter than the distal main cusp. Some teeth also bear a minute mesial main cusp on the basal third of the mesial margin of the main cusp (Fig. 5H). The mesial and distal carinae of the main cusp are usually denticulated along the whole crown height. The specimen BP/1/4529 shows the unique feature of having both mesial and distal denticles divided into two parts, that is, a large sub-denticle apically and a smaller one basally (Figs. 5J and 5K). The distal main and distal accessory cusps are also denticulated in this specimen (Fig. 5J). A lingual cingulum composed of six to seven cuspules is clearly present in the mesial lower sectorial teeth of AM 3753, MB R1004, and SAM-PK-K177. This cingulum is, however, absent in the upper sectorial postcanines of BSP 1934 VIII 14 and in the distalmost upper and lower sectorial teeth of MB R1004.

Crown enamel surface texture and microstructure

Histological studies of the postcanine of D. tetragonus were done by Grine (1977, 1978), Osborn & Hillman (1979) and Sander (1997b). The enamel microstructure of this taxon was revealed to be prismless and columnar (the synapsid columnar enamel SCE of Sander, 1997b), and to include incremental lines and enamel tubules. The enamel surface texture (sensu Hendrickx, Mateus & Araújo, 2015) of incisors, canines and postcanines is braided and apicobasally oriented.

Titanogomphodon crassusKeyser, 1973

Holotype: GSN R323, an incomplete cranium missing the mesial tip of the rostrum.

Referred dental material: None.

Occurrence: Northern slope of Etjo Mountain, Otjozondjupa Region, Namibia.

Horizon: Etjo Beds, upper levels of the upper Omingonde Formation.

Age: late Anisian-Ladinian?, Middle Triassic. The upper fauna from the upper Omingonde Formation is correlated to that from the Dinodontosaurus AZ of Brazil and the Chañares Formation of Argentina (Abdala & Smith, 2009; Abdala et al., 2013; Martinelli et al., 2017). Given that the Chañares Formation was recently dated to the lower Carnian (Marsicano et al., 2016), the upper fauna from the Omingonde Formation might also be from the lowermost part of the Upper Triassic.

Dental formula: i?/? : c?/? : pc>7/? (cpc?: gpc>3: tpc1 : spc3).

Dental morphology: Keyser (1973) described the dentition of Titanogomphodon in relative details given the poor preservation of the teeth. The holotypic specimen GSN R323 preserves the distal portion of the three distalmost gomphodont postcanines as well as a transitional postcanine and the first and third sectorial teeth (Fig. 6).

Figure 6 Dentition of Titanogomphodon crassus.

(A–C and E) Right upper gomphodont, transitional and sectorial postcanines of GSN R322, with close up on (B) the penultimate gomphodont, (C) mesial accessory ridge of the distalmost gomphodont, and (E) distalmost sectorial postcanine in apical view; (D–F) left mesialmost preserved upper gomphodont postcanine of GSN R322 in (D) apical and (F) mesial views. Abbreviations: lac, labial cusp; lamc, labiomesial accessory cusp; lic, lingual cusp; mar, medial accessory ridge; mcc, mesial cingular cusp; spc, sectorial postcanine; tc, transverse crest; tpc, transitional postcanine.

Upper gomphodont postcanines

As noted by Keyser (1973), the transverse crest of the gomphodont postcanine, which connects the labial and lingual cusps (both unpreserved in all gomphodont postcanines of GSN R323), is slightly mesially displaced (Fig. 6B). As seen in Diademodon upper postcanines, a faint mesial accessory ridge can be observed in the distalmost gomphodont tooth (Figs. 5A and 5C). Unlike Keyser’s (1973) description of the upper gomphodont postcanines, no labiodistal accessory cusp or distinct cuspules forming a distal cingulum can be observed in the preserved teeth (Fig. 6A). Nonetheless, the damaged right mesialmost gomphodont postcanine appears to show a labiomesial accessory cusp, mesial to the unpreserved labial cusp as well as a mesial cingular cuspule (Figs. 6D and 6F). Many portions of the enamel and dentine surface of this tooth are, however, missing and the presence of these two accessory cusps/cingular cuspules needs to be confirmed. Similar to some upper postcanines of Diademodon that have been worn out, and unlike trirachodontid postcanines, the mesial and distal basins are absent in all preserved postcanines of Titanogomphodon.

Transitional and sectorial postcanines

According to Keyser (1973), Titanogomphodon is diagnosed by the diagonal orientation of the sectorial postcanines, of which the long axis is linguomesially oriented. Such orientation of the sectorial teeth indeed contrasts with the mesiodistally oriented upper sectorial postcanines of other gomphodont cynodonts such as Diademodon, Langbergia, Trirachodon, Andescynodon, and Boreogomphodon, in which the long axis is parallel to the labial margin of the upper jaw. As in Diademodon and unlike trirachodontids, a transitional postcanine is present between the gomphodont and sectorial teeth. Little information can be extracted from the transitional and sectorial postcanines due to wear and the incompleteness of the crowns. No main and accessory cusps, or cingular cuspules are preserved in the sectorial postcanines and only a concavity centrally positioned on the lingual surface of the crown and root is present in the two sectorial teeth (Figs. 6A and 6E).

Trirachodontidae Romer, 1967

Langbergia modiseiAbdala, Neveling & Welman, 2006

Holotype: NMQR 3255 (from Langberg), an incomplete cranium.

Referred dental material: from Goedgedacht: NMQR 3251; from Langberg: NMQR 3256, BP/1/5362; from Rexford Store: NMQR 3268, BP/1/5363; from Eerste Geluk: NMQR 3280, 3281; from Bosrand: BP/1/5400, 5401, 5404; from Driefontein: BP/1/5666; from Moerbeidal: CGP/1/33; from Palmiet Fontein: CGP/1/120; from Kaaimansgat: SAM-PK-11481.

Occurrence: (from Abdala, Neveling & Welman (2006), Table 1) Langberg 556 and Driefontein, Paul Roux, Thabo Mofutsanyane District, Free State Province, South Africa; Rexford Store 433, Bethlehem, Thabo Mofutsanyane District; Eerste Geluk 131, Kestell, Thabo Mofutsanyane District; Goedgedacht 15, Marquard, Thabo Mofutsanyana District; Bosrand 12, Senekal, Thabo Mofutsanyane District; Moerbeidal 648, Kaaimansgat 146 and Hugo’s Kop 620, Rouxville, Xhariep District, Free State Province, South Africa; Palmiet Fontein 94, Tarkastad, Chris Hani District, Eastern Cape Province, South Africa.

Horizon: subzone A of the Cynognathus Assemblage Zone (AZ), Burgersdorp Formation, Beaufort Group, Karoo Supergroup.

Age: late Olenekian, Early Triassic.

Dental formula: i4/3 : c1/1 : pc7-9/8 (cpc1-2: gpc3-7?: spc2-4).

Dental morphology: The dentition of Langbergia modisei has been well-described by Abdala, Neveling & Welman (2006). Here, we provide additional information based on firsthand examination of the best-preserved specimens.

Incisors

The mesial carina of the incisors does not seem to be denticulated, whereas the distal carina is denticulated and bears around 30 denticles per five mm (contra Abdala, Neveling & Welman, 2006; Figs. 7A–7C). The denticles of the first right lower incisor of NMQR 3251 have a symmetrically convex external margin and are poorly delimited. They are apicobasally elongated and change sporadically in size along the carina (Figs. 7A and 7B). The largest denticles are at mid-crown and the denticles do not extend up to the crown apex and the root (Figs. 7A and 7B). The distal denticles of the fourth upper incisor of NMQR 3281 and SAM-PK-11481 share the same morphology, with the largest denticles at mid-crown and the basalmost and apicalmost denticles extending far above the root (Fig. 7C) and below the crown apex, respectively. A concave surface adjacent to the distal carina is also present on the labial surface of the third? right incisor in SAM-PK-11481. Based on the first and best-preserved lower incisor of NMQR 3280, the mesial carina faces mesially whereas the distal carina is distally, almost linguodistally positioned, giving a lenticular to D-shaped cross-sectional outline at mid-crown. The distal profile of the upper incisors of this specimen is sigmoid in labial view, with the basal one-third and the apical two-thirds of the crown being convex and concave, respectively. The incisors are distally recurved and some of them are weakly mesiodistally constricted at the cervix.

Figure 7 Dentition of Langbergia modisei.

(A and B) First left lower incisor of NMQR 3251, with (B) close up on the distal carina in labial view; (C) close up on the distal carina of the fourth left upper incisor of NMQR 3281 in labial view; (D and F–G) right lower canine of NMQR 3251, with close up on the (F) distal and (G) mesial carinae, in labial view; (E) cross-sectional outline of the right upper canine of SAM-PK-11481 in apical view (labial and mesial sides to the top and the right, respectively); (H) right upper canine of BP/1/5362 in labial view; (I) centromesial portion of the erupting left upper canine of NMQR 3268 in labial view; (J) fourth and fifth left upper gomphodont postcanines of NMQR 3251 in apical view; (K) third left upper postcanine of NMQR 3255 in apical view; (L) close up on the fourth left upper postcanine of NMQR 3251 in mesial view (crown upside down); (M) third and fourth right lower gomphodont postcanine of NMQR 3251 in apical view; (N) first left lower gomphodont postcanine of NMQR 3251 in apicodistal view; (O) first and second right upper sectorial postcanine of NMQR 3251 in labial view; (P) first right upper sectorial postcanine of BP/1/5362 in labial view; (Q) second left upper sectorial postcanine of NMQR 3255 in linguomesial view. Abbreviations: cc, central cusp; dac, distal accessory cusp; db, distal basin; de, denticle; dmc, distal main cusp; lac, labial cusp; lamc, labiomesial accessory cusp; lic, lingual cusp; licc, lingual cingular cuspule; lidc, linguodistal accessory cusp; limc, linguomesial accessory cusp; lri, longitudinal ridge; mac, mesial accessory cusp; mb, mesial basin; mc, main cusp; mcc, mesial cingular cuspule; mci, mesial cingulum; mmc, mesial main cusp. Elongation axes of the crown and the central cusp in green and red, respectively.

Canines

Both upper and lower canines of L. modisei have denticulated carinae (BP/1/5362; NMQR 3251, 3268) with a denticle density ranging from 20 to 32 denticles per five mm. In NMQR 3251 and 3268, the mesial and distal denticles of the canines do not extend to the cervix basally, but do reach the crown apex in NMQR 3251. Similar to the incisors, the mesial, and distal denticles change sporadically in size along the carinae and have a symmetrically to asymmetrically convex external margin (Figs. 7F and 7G). The denticles of the canines tend to be better delimited and mesiodistally longer than those of the incisors so that denticles are apicobasally elongated and sometimes subquadrangular in shape. Despite the fact of changing randomly in size, the basal and apicalmost denticles are smaller than those of the mid-crown on both mesial and distal carinae. Faint longitudinal ridges delimiting mesiodistally short, flat facets can be observed on the labial surface of the canines of BP/1/5362 (Fig. 7H) and SAM-PK-11481. In the latter specimen, the mid-crown cross-sectional outline of the upper canine is roughly salinon-shaped (sensu Hendrickx, Mateus & Araújo, 2015), with a wide convex labial margin showing a weakly concave labiodistal surface, and a biconcave lingual margin, of which the mesial concavity is longer than the distal one (Fig. 7E). A concave surface is also present on the mesial and distal surfaces of the erupting right and left upper canines of NMQR 3268, respectively. The cross-sectional outline of the canine of NMQR 3251 is, however, parabolic at mid-crown. This indicates that there is variation in cross-sectional outline in the canines. No transverse undulations could be observed on the upper and lower canines of the specimens examined. The enamel surface texture of NMQR 3268 is clearly braided and curves basally close to the carinae (Fig. 7I).

Gomphodont postcanines

As noted by Abdala, Neveling & Welman (2006), the first upper and lower postcanines differ in their morphology. The first upper postcanine of NMQR 3251 is significantly smaller than the more distal ones, has a sub-circular cross-sectional outline and only bears a single main cusp, a morphology that is similar to the conical postcanines of Diademodon. In NMQR 3255, the first upper postcanine has an oval cross-sectional outline and is only slightly smaller than the other gomphodont postcanines. Given the incomplete preservation of the first upper postcanines on both sides, it is unknown whether they were monocuspid as well. The morphology of the first lower postcanine appears to be more complex. It includes a main centrally positioned and a slightly labially displaced cusp (Fig. 7N), which is homologous to the central cusp of more distal gomphodont postcanines based on its size, position, and morphology. This main cusp is bordered by a small apically pointed mesial accessory cusp and a low lingual cingulum formed by at least three minute cingular cuspules (Fig. 7N), a morphology reminiscent to that of sectorial postcanines seen in diademodontids and trirachodontids. Because the distal portion of this tooth is missing, it is unknown whether an accessory cusp was present distal to the main cusp.

Unlike other trirachodontids, the upper and lower gomphodont postcanines of Langbergia roughly share the same morphology, that is, they are elliptical in cross-section, slightly labiolingually elongated and their long axis is slightly to strongly linguomesially oriented (Abdala, Neveling & Welman, 2006). This diagnosis applies to the specimens with upper and lower postcanines visible in apical view (i.e., NMQR 3251 and NMQR 3255), with the exception of NMQR 3268. Indeed, although the upper and lower gomphodont postcanines of NMQR 3251 and NMQR 3255 share a crown base ratio (CBR) varying between 1.3 and 1.45, the upper gomphodont postcanines of NMQR 3268 are more labiolingually elongated, with a CBR around 2 in the widest teeth. Likewise, the long axis of the upper postcanines of NMQR 3268 is perpendicular to the long axis of the cranium. It is unknown whether these dental differences are ontogenetic, intra- or interspecific, and NMQR 3268 is here tentatively kept within the species L. modisei based on the absence of a maxillary platform, a labiolingually narrow and prominent central cusp, and the stratigraphic distribution of the specimen (i.e., Cynognathus subzone A).

Both upper and lower gomphodont postcanines of Langbergia increase in size up to the penultimate tooth, which is slightly larger than the last gomphodont postcanine. The long axis of the upper gomphodont postcanines is roughly parallel in all teeth in NMQR 3251, 3255, and 3268, with very little variation along the tooth row. In the lower postcanines, however, it changes in orientation, with the long axis being strongly mesially oriented in the first two postcanines, then almost parallel to the long axis of the mandibular tooth row in the last gomphodont tooth. All gomphodont postcanines bear a high, wide, centrally positioned central cusp bordered labially and lingually by the labial and lingual cusps, respectively (Figs. 7J–7M). The central cusp always appears to be higher and wider than both labial and lingual cusps. It is separated from these two cusps by a deep and narrow valley extending basally at a certain distance above the mesial and distal basins (Fig. 7L). The labial, central, and lingual cusps that form the transverse crest are not perfectly aligned on the same plane of elongation of the postcanine. The long axis of the central cusp either follows that of the transverse crest or is diagonally oriented to it (Figs. 7J, 7K and 7M). Likewise, the labial cusp of the upper postcanines is significantly deflected mesially from the diagonally oriented axis of the transverse cusp whereas the lingual cusp of the lower postcanines is weakly deflected distally in some teeth (Figs. 7J, 7K and 7M). The lingual and central cusps are, however, aligned in the same plane of elongation of the postcanine while the labial cusp remains mesially displaced from that plane. A cingulum formed by three to four large and sometimes strongly protruding cingular cuspules delimits both the mesial and distal margin of the upper and lower gomphodont postcanines in NMQR 3251 and NMQR 3255. The labiomesial and linguodistal cusps are the largest accessory cusps of the cingula in the upper gomphodont postcanines whereas the labiomesial, linguomesial, and linguodistal are the largest in the lower postcanines. Although incomplete and badly preserved, the upper postcanines of NMQR 3268 do not seem to bear these large accessory cusps on the cingula. The mesial and distal cingula instead comprise a large number of small cingular cuspules, as seen in Cricodon and Trirachodon. Due to the incompleteness of the upper postcanines of NMQR 3268, it is unknown whether the cingular cuspules extend along the whole width of the crown or are restricted to a certain part of the cingulum. A similar morphology is seen in the mesial cingulum of the last right upper gomphodont of SAM-PK-11481, in which the cingular cuspules also form low and poorly delimited convexities. The whole morphology of this gomphodont postcanine, which is strongly diagonally oriented and whose labiomesial portion is the only one not hidden by the matrix, in fact departs from that seen in the gomphodont teeth of NMQR 3251, 3255, and 3268. The labial cusp of the crown in SAM-PK-11481 appears to form a minute projection separated from the central cusp by a much wider and higher cusp. It is, however, possible that the wider cusp directly adjacent to the central cusp is in fact the labial cusp, and that the small cusp, which is connected to the mesial cingulum by a low ridge, corresponds to a cingular cuspule mesial to the labial cusp. The cingular cuspules, however, increase in width toward the center of the crown at least on the distal cingulum in this specimen. The mesial and distal basins, delimited by the transverse crest and the cingulum, are deep and mesiodistally short in Langbergia (Figs. 7J, 7K and 7M). As observed for the canines, the enamel texture of the gomphodont postcanines is braided and apicobasally oriented.

Sectorial postcanines

The sectorial postcanines always include a main recurved cusp followed by one distal main cusp and, in some teeth, a distal accessory cusp distally, and one or no mesial main cusp (Figs. 7O and 7P). The mesial margin of the main cusp is denticulated on the apical third of the crown in BP/1/5362 (Fig. 7P). The denticles are poorly defined, low, apicobasally elongated projections on the carina. As in the denticles of the incisors and canines, they sporadically change in size and their external margin is symmetrically to asymmetrically convex. The distal main cusp is the longest whereas the distal accessory cusp is significantly smaller and sometimes appears as a minute projection. The mesial main cusp is often large, projects apically and strongly protrudes from the basal third of the crown (Fig. 7O). The distal main cusp also projects distally in some specimens. As observed by Abdala, Neveling & Welman (2006), a lingual cingulum composed of five cingular cuspules is present in the second upper sectorial postcanine of NMQR 3255 (Fig. 7Q). The mesialmost cingular cuspule is long and followed distally by two short cusps, a much longer one, and then a medium sized one. A sigmoid and basally inclined cingulum formed by several minute cingular cuspules is also visible on the lingual surface of the third lower sectorial postcanine of NMQR 3251. In the latter specimen, the last upper sectorial crown is significantly shorter and slightly lower than the preceding one. It is a tricuspid postcanine in which the main cusp, distal main cusp, and distal accessory cusp projects basodistally and decrease in height and length distally (Fig. 7O). The distal main cusp is a subtriangular prominence at mid-crown height whereas the distal accessory cusp is a minute projection in the basalmost part of the crown (Fig. 7O).

Cricodon metabolusCrompton, 1955

Holotype: UMZC T905 (from Njalila, Manda Beds, Tanzania), incomplete upper and lower jaws with several loose teeth.

Referred dental material: from Avilion farm, subzone C, Cynognathus AZ, South Africa: BP/1/5540, 5835, 6159; from Norwood farm, subzone C, Cynognathus AZ, South Africa: BP/1/6102; from Sitwe, Ntawere Formation, Zambia: NHCC LB28; from Ruhuhu Basin, Manda Beds, Tanzania: NMT RB227, NHMUK PV R36800; from Aliwal North, subzone B, Cynognathus AZ, South Africa: NHMUK PV R3722, SAM-PK-6212, 5881a and b.

Occurrence: Stockley’s bone locality B11, Njalila, Tanzania; Outcrop Z40 (Z51 according to Sidor & Hopson, 2018), Ndatira River, Litumba Ndyosi District, Ruhuhu Basin, Tanzania (Smith et al., 2018); Outcrops Z29, Z81, near the village of Lutimba Kuhamba, Litumba Kuhamba District, Tanzania (Sidor & Hopson, 2018; Smith et al., 2018); Outcrop L23, five km west of Sitwe, Eastern Province, Zambia; Avilion and Norwood farms, Bamboeshoek Valley, Sterkstroom, Chris Hani District, Enoch Mgijima Municipality, Eastern Cape Province, South Africa; Aliwal North, Joe Gqabi District, Walter Sisulu Municipality, Eastern Cape Province, South Africa.

Horizon: middle to upper Lifua Member, Manda Beds; subzones B and C of the Cynognathus AZ, Burgersdorp Formation, Beaufort Group, Karoo Supergroup; upper Ntawere Formation.

Age: early to late Anisian, Middle Triassic.

Dental formula: i4/3 : c1/1 : pc9-10/11 (cpc0 : gpc7-9 : spc0-2).

Dental morphology: The dentition of the holotypic specimen of Cricodon metabolus was described by Crompton (1955) and additional information was given by Abdala, Hancox & Neveling (2005) and Sidor & Hopson (2018) based on referred specimens. Although Sidor & Hopson (2018) recently comprehensively described the well-preserved dentition of an immature Cricodon metabolus (NMT RB227) from Zambia, a detailed examination of the holotype and referred material deposited in South Africa and England enable us to provide additional information on the dental morphology of this taxon. BP/1/5540, 5835, 6102, 6159, NHMUK PV R3722, R36800, SAM-PK-6212, 5881a (represented by a badly crushed snout and the anterior portion of a mandible) and 5881b (preserving the left maxillary tooth row with five upper gomphodont postcanines) are here tentatively referred to Cricodon metabolus based on the large size, the low denticle density of the incisors and canines, the drop-shaped outline of the distalmost upper postcanines, the step-like disposition of distalmost upper gomphodont postcanines, and the long axis of the lower postcanines perpendicular to the mandibular tooth row. We, however, note important differences in the cranial, mandibular, and dental anatomy of the holotype and referred specimens NMT RB227, BP/1/6102, SAM-PK-5881 (e.g., length of the diastema between the canine and the first upper/lower postcanine, morphology of the first upper postcanines and last gomphodont postcanines, lateral extension of the platform marginal to the dental series). If these differences could be ontogenetic, we do not exclude the possibility that these referred specimens represent one or several new trirachodontid species. Conversely, BP/1/5538 from the subzone C Cynognathus AZ of the Avilion farm (Sterkstroom District) is confidently excluded from this taxon. This specimen, represented by the posterior portion of the snout and orbits with the three last gomphodont and the first sectorial postcanines preserved, differ from Cricodon by: (i) longer and more ovoid upper distal gomphodont postcanines (CBR of 1.5 vs. CBR of 1.85–2.5 typical of Cricodon); (ii) no step-like disposition of the antepenultimate and penultimate upper gomphodont postcanines; (iii) upper postcanines with a large labiomesial accessory cusp and no distal cingulum (contra Abdala, Hancox & Neveling, 2005); (iv) sectorial upper postcanines with a large strongly pointed and strongly apically recurved main cusp and two distal accessory cusps. BP/1/5538 may also belong to a new gomphodont taxon.

Incisors

The mesial and distal carinae of the partially erupted first left lower incisor of the holotype UMZC T905 are clearly denticulated and include apicobasally elongated to subquadrangular and weakly apically inclined denticles with a symmetrically convex external margin (Figs. 8A and 8B). Both mesial and distal denticles decrease in size apically and reach the crown apex, although the tip of the apex remains unserrated. There are around 1.75 denticles per mm (denticle density of ∼9/5 mm) in the basalmost erupted part of the mesial and distal carinae, and 3.5 denticles per mm at the apex. The upper incisors of BP/1/6102 and BP/1/5540 share the same denticle density, with 7–10 denticles per five mm in the distal carina (Figs. 8C and 8D). This contrasts with the much higher density of denticles (>20/5 mm at mid-crown) on the incisors of other trirachodontids. The mesial carina and denticles of BP/1/6102 are not visible and are most likely covered by sediment. Only the apical denticles of the first left upper incisor can be seen on the mesial carina of BP/1/5540. These mesial denticles are apicobasally elongated, have a flat external margin and extend one mm below the crow apex. The denticles of the distal carina of BP/1/6102 and BP/1/5540 also extend from the cervix to the base of the crown apex (Fig. 8C). In BP/1/6102, the distal denticles decrease in size apically and basally from the mid-crown height, and are apicobasally elongated at mid-crown and in the apical part of the crown and almost subquadrangular in the basal third of the carina (Fig. 8D). On the other hand, the distal denticles of BP/1/5540 are apically inclined and subquadrangular on most of the carina. The denticles of the third replacing upper incisor of NHCC LB28 are more prominent distally than mesially, and the distal denticles also increase in size from the apex toward the base of the crown in both upper and lower incisors (Sidor & Hopson, 2018). The external margin of the denticles in BP/1/5540 is symmetrically to asymmetrically convex and weakly parabolic. The interdenticular space is apicobasally wide in the basal denticles and narrow in the mid-crown and apical denticles in BP/1/6102 whereas it is wide in most of the denticles in BP/1/5540. The distal denticles of the incisors change sporadically in size along the carina in Cricodon. The mesial carina faces mesially whereas the distal carina is oriented linguodistally in the first two and distally in the last two upper incisors. The incisors are recurved, with a strongly convex mesial profile and a straight to slightly concave distal profile. Although the incisors from all BP specimens referred to Cricodon do not seem to be constricted at the cervix, a constriction appears to be present in the lower incisors of NHCC LB28, giving a spatulate outline to the crowns (Sidor & Hopson, 2018). No information can be provided on the cross-sectional outline of the incisors, yet the labial surface is convex and the lingual flat in NHCC LB28 (Sidor & Hopson, 2018). The root of the incisors of this specimen are long, taper apically and have an oval cross-sectional outline that is narrower mesiodistally than labiolingually (Sidor & Hopson, 2018).

Figure 8 Dentition of Cricodon formosus.

Dentition of Cricodon metabolus. (A and B) First left lower incisor of UMCZ T905 in (A) labial and (B) labiodistal views; (C and D) fourth left upper incisor of BP/1/6102, with (D) close up on the distal denticles, in labial view; (E and F) left lower canine of SAM-PK-5881a, in (E) lingual view; with (F) cross-sectional outline at mid-crow in apical view; (G) centromesial and (H) centrodistal denticles of the right upper canine of BP/1/6102 in labial view; (I) centrodistal denticles of the right upper canine of BP/1/5540 in labial view; (J) eight? and nine? right upper gomphodont postcanines of UMCZ T905 in apical view; (K) eight and nine right lower gomphodont postcanines of UMCZ T905 in apical view; (L) first left lower gomphodont postcanine of SAM-PK-5881a in apical view; (M) last right upper gomphodont postcanine of SAM-PK-5881b in distal view; (N) last left upper gomphodont postcanine of UMCZ T905; (O) close up on the denticles on the labiodistal ridge of an isolated left upper postcanine of UMCZ T905 in labiodistal view; (P and Q) ninth right upper sectorial postcanine of UMCZ T905 in (P) labial and (Q) apical views. Abbreviations: cac, central accessory cusp; cc, central cusp; cos, concave surface; db, distal basin; dcc, distal cingular cuspule; dci, distal cingulum; dde, distal denticle; de, denticle; dmc, distal main cusp; ent, enamel surface texture; lac, labial cusp; ladc, labiodistal accessory cusp; lic, lingual cusp; lidc, linguodistal accessory cusp; limc, linguomesial accessory cusp; lri, longitudinal ridge; mb, mesial basin; mc, main cusp; mcc, mesial cingular cuspule; mci, mesial cingulum; mde, mesial denticle; mmc, mesial accessory cusp; tc, transverse crest; tun, transverse undulation. The red lines highlight the angular labiodistal margin of the lower postcanines.

Canines

The morphology of the canines of Cricodon is based on the upper and lower canines of NHCC LB28, the upper canines of BP/1/6102, BP/1/5540 and NHMUK PV R3722, and the lower canines of SAM-PK-5881a, as the canines are not preserved in the holotype (Crompton, 1955). As noted by Abdala, Hancox & Neveling (2005) and Sidor & Hopson (2018), both carinae of the lower canines are denticulated (Fig. 8E). The denticles are slightly bigger than those of the incisors, with six to seven denticles per five mm on the distal carinae at mid-crown in BP/1/6102 and BP/1/5540 (Figs. 8G–8I), and 9–11 in NHCC LB28 (Sidor & Hopson, 2018). The denticle density of the smaller specimens NHMUK PV R3722 and SAM-PK-5881a vary between 12.5 and 14 denticles per five mm. The distal denticles are larger than the mesial ones in BP/1/6102, BP/1/5540, and NHCC LB28, with a denticle density of 8.6 in the mesial carina and 6.8 in the distal carina in BP/1/6102 (Figs. 8G and 8H), 8 and 6.6 denticles/five mm in BP/1/5540, and 9–11 and 15–19/5 mm in NHCC LB28 (Sidor & Hopson, 2018). Although Sidor & Hopson (2018) consider a difference in mesial and distal denticle density as a diagnostic feature in Cricodon, we note, however, no size discrepancy between the mesial and distal denticles in the upper and lower canines of NHMUK PV R3722 and SAM-PK-5881a. The denticles are apicobasally elongated in the mesial and distal carinae of NHMUK PV R3722 and SAM-PK-5881a, and apicobasally elongated to subquadrangular in the distal carina of BP/1/6102 and BP/1/5540. Similar to the incisors, there is a sporadic variation in denticle size along the carina, with a minute denticle followed by a much larger one in some portion of the carina. The interdenticular space is particularly wide in the mid-crown denticles of the distal carina in BP/1/5540 (Fig. 8I). The cross-sectional outline is lenticular at mid-crown in the upper canines of BP/1/6102 and NHCC LB28 (Sidor & Hopson, 2018, figure 2C), and weakly salinon-shaped in the lower canine of SAM-PK-5881a (Fig. 8F) due to the presence of concave surfaces adjacent to both carinae (Fig. 8E). In the latter specimen, several transverse undulations are present on the lingual surface of the lower canine at mid-crown (Fig. 8E). Short transverse undulations (“crenulations” sensu Sidor & Hopson, 2018) are also present in the replacement upper canines of NHCC LB28 (Sidor & Hopson, 2018). Unlike the postcanines, the surface texture of the incisors and canines appears to be irregular and non-oriented.

Upper gomphodont postcanines

The morphology of the first to the fifth upper postcanines relies on specimens NHCC LB28 and SAM-PK-5881a. In NHCC LB28, the first two postcanines are sectorial, a condition only seen in the putative juvenile specimens of Trirachodon BP/1/4534 and 4535 among Gomphodontia (Sidor & Hopson, 2018; C. Hendrickx, 2018, personal observation). Both sectorial postcanines are particularly small compared to the more distal ones, and diagonally oriented from the long axis of the upper postcanine tooth row. They include a centrally positioned main cusp and a mesial and distal main cusps (Sidor & Hopson, 2018). The mesial main cusp occupies one fourth of the main cusp length and projects at three-fourths of the crown (Sidor & Hopson, 2018). The distal main cusp is larger, longer (i.e., one half of the main cusp length) and better separated from the main cusp (Sidor & Hopson, 2018). This distal main cusp also lies more basal than the mesial main cusp, around one half of the crown height. A small cingular cuspule is present on the lingual side of the second postcanine, basal to the mesial valley separating the main cusp from the mesial main cusp. The sectorial postcanines are swollen just beneath the cusps and decrease in mesiodistal length toward the root (Sidor & Hopson, 2018).

The upper postcanines 3–10 of NHCC LB28 are gomphodont, with their long axis perpendicular to that of the tooth row. The postcanines increase in mesiodistal length, labiolingual width and elongation up to the seventh tooth, then decrease in size and elongation more distally (Sidor & Hopson, 2018). The third and fourth upper postcanines are slightly smaller and shorter than those from the ninth and 10th alveoli (Sidor & Hopson, 2018, table 1). Likewise, postcanines eight to 10 are significantly smaller than the three preceding postcanines, and are labially deflected from them, so that their labiolingual width are in line with the labial two-thirds of postcanines five to seven. In apical view, all upper gomphodont postcanines are roughly pear-shaped, with the labial third of the crown being significantly mesiodistally shorter than the lingual two-thirds. Postcanines eight to 10, in fact, display a mesiodistal constriction labially, at one-third of the crown (Sidor & Hopson, 2018). Each upper gomphodont postcanine includes a labiolingually oriented and centrally positioned transverse crest comprising a labial, lingual, and central cusps, all separated by shallow valleys (Sidor & Hopson, 2018). Minute central accessory cusps are also present on the transverse crest, on each side of the central cusp, as seen in the eighth unworn postcanine. The central cusp is labiolingually elongated and lingually deflected, whereas the labial and lingual cusps are mesiodistally long and have a lenticular cross-sectional outline, with their pointed extremity directed toward the central cusp (Sidor & Hopson, 2018). The labial cusp, which is slightly longer than the lingual cusp, is the tallest of the three cusps, whereas the lingual and central cusps share the same height (Sidor & Hopson, 2018). The upper gomphodont postcanines are ringed by mesial and distal cingula, which enclose shallow and short mesial and distal basins, respectively. Both cingula extend on the lingual two-thirds of the crown, the labial third being devoid of cingular cuspules. The mesial cingulum lies more basal than the distal one and joins the base of the labiomesial ridge of the labial cusp in postcanines 3–5 (Sidor & Hopson, 2018). There are around six cuspules on each mesial and distal cingula in postcanines 5–7 (eight on the mesial cingulum of the fifth postcanine), with the largest cingular cuspules being in the lingual portion of the crown, adjacent to the lingual cusp. Postcanines nine and 10, however, have a lower number of cingular cusps, with five and three cuspules on the distal cingulum, respectively, and three and two in the preserved portion of the mesial cingulum, respectively (Sidor & Hopson, 2018). Similar to the more mesial postcanines, the cingular cuspules of postcanines nine and 10 are restricted to the lingual portion of the cingula, with the most lingual cuspules being the largest. As seen in the holotype, the enamel surface texture of the upper gomphodont postcanines is apicobasally oriented and braided (Sidor & Hopson, 2018).

The first upper postcanine of SAM-PK-5881a is incomplete, worn and its labial part is hidden by matrix, but clearly shows a gomphodont morphology. No mesial and distal basins can be seen but they might have been worn out. The more distal upper postcanines, which are relatively worn, share the same morphology and only their size and labiolingual elongation increase distally. The mesial margin of the second and third upper postcanines is slightly concave and follows the convexity of the distal margin of the preceding crown.

All isolated and in situ upper gomphodont postcanines of the holotype UMCZ T905, which belong to the distal half of the maxilla, are labiolingually expanded and ovate in outline, with a CBR of 2.5, 2.3, 2, and 1.85 for the sixth, seventh, eighth, and ninth left postcanines, respectively. These morphology contrasts with those of the eighth to 10th upper postcanines of NHCC LB28, in which CBR does not exceed 1.5 (Sidor & Hopson, 2018, table 1). The long axis of postcanines six to nine of the holotype is roughly perpendicular to the long axis of the tooth row, and only the last gomphodont tooth is labiomesially oriented. A striking feature of the upper tooth row of Cricodon is the step-like disposition of the antepenultimate and penultimate gomphodont postcanines, so that each of these two teeth is at the level of the base-crown or root of the preceding postcanine. This feature is also present in NHCC LB28 (Sidor & Hopson, 2018, figure 3A), SAM-PK-5881b, SAM-PK-6212, and BP/1/6102. The upper gomphodont teeth are slightly mesiodistally constricted at the labial one-third of the crown, giving an asymmetrical outline of the postcanine in apical view, with the labial one-third of the tooth being shorter than the lingual one (Fig. 8J). A similar constriction can be seen in the ninth upper gomphodont postcanine. The constriction is, however, slightly more pronounced on the distal surface of the eighth postcanine and particularly well-developed on the distal margin of the sixth and seventh teeth. A wide concavity occupying the labial third of the crown can be seen on the distal surface of these two postcanines. The upper gomphodont postcanines include a low and centrally positioned transverse crest, which is straight (Fig. 8J), sigmoid or weakly parabolic in outline, with the concavity directed distally, in apical view. The transverse crest is made of a wide and lingually displaced central cusp bordered by two prominent labial and lingual cusps. No preserved upper postcanines bear fully complete labial, central, and lingual cusps but the labial cusp appears to be as high as the central cusp, and as high as or slightly higher than the lingual cusp. As described by Crompton (1955), the labial and lingual ridge of the central cusp are denticulated, with the widest denticles being close to the apex (Fig. 8N). The denticles of the central cusp are wide, with a weakly convex external margin. Denticles are also clearly visible on the mesial, distal, and central ridges of the labial and lingual cusps, being particularly numerous and well-developed on the labiomesial and labiodistal ridges of the labial cusp (Fig. 8O). Unlike the central cusp, the denticles of the mesial, distal, and central ridges of the labial and lingual cusps diminish in size apically. Similar to the denticles of incisors and canines, they are apicobasally elongated and change sporadically in size. Their external margin is, however, flattened or weakly convex (Fig. 8O). The denticles on the labiomesial and labiodistal ridges either extend along the whole cusp up to its apex, or are restricted to the basal half of the cusp.

The mesial and distal cingula of the upper postcanines bear a large number (>7) of small and well-delimited cingular cuspules along most, if not all, the width of the mesial and distal rims of the crown (Figs. 8J and 8N). Unlike more mesial teeth, the distal cingulum of the ninth left upper postcanine does not extend to the labial cusp. The same can be said of the mesial cingulum of a loose left upper postcanine. The cingular cuspules sporadically vary in size along the crown, yet the widest cuspules are found in the central and/or lingual parts of the cingulum. We counted 10 cingular cuspules on the mesial cingulum and seven distal in the ninth left upper gomphodont postcanine, and seven cuspules in the preserved labial half of the distal cingulum of the eighth left upper postcanine. The preserved portion of the mesial and distal cingula of the upper gomphodont postcanine sampled by Hendrickx, Abdala & Choiniere (2016) includes nine and 11 cingular cuspules, respectively. Unlike to the ninth left upper postcanine, the widest cingular cuspules of this isolated tooth are seen in the labial portion of the distal cingulum and in the centrolabial and lingual parts of the mesial cingulum (Fig. 8N). The mesial cingulum is parabolic or weakly sigmoid and slopes basolabially in mesial view. The distal cingulum, on the other hand, is symmetrically parabolic or biconcave. In two loose upper postcanines, the denticulated labiodistal ridge of the labial cusp does not connect to the distal cingulum as the lingual extremity of the latter extends on the labiodistal surface of the labial cusp, so that the lingualmost distal cingular cuspules are situated apical to the labiodistal ridge.

The upper postcanines of the holotype and two specimens with the same specimen number (SAM-PK-5881) largely follow the same morphology, yet a few differences can be noted. Although the CBR of the largest upper postcanine (i.e., third right gomphodont postcanine, CBR of 2.3) of SAM-PK-5881a falls within the range of values obtained in the holotype, the upper postcanines of SAM-PK-5881b are more ovoid, with a CBR beyond 1.7. In addition, the long axis of the last upper postcanine is not diagonally oriented in SAM-PK-5881b. A deep and narrow concavity is also present in the central portion of the distal cingulum of the two distalmost upper postcanines in this specimen, so that the cingulum of these crowns is conspicuously biconvex (Fig. 8M). The upper postcanine roots of SAM-PK-5881b, well visible in mesial and labial views, are apicobasally long, three times the height of the crown in at least the first gomphodont postcanine, and weakly taper apically. The cervical line separating the crown from the root is roughly horizontal and particularly neat in this specimen due to the light brown-orange coloring the crowns, which contrasts with the white color of the roots.

Lower gomphodont postcanines

As described by Crompton (1955), the upper and lower gomphodont postcanines of the holotype roughly share the same morphology, but their CBR is significantly lower in the lower teeth, varying from 1.2 to 1.6 (Fig. 8K). Similar to the upper postcanines, the first lower postcanine, preserved in SAM-PK-5881a, is weakly elongated labiolingually and includes a transverse crest made of labial, central and lingual cusps and ringed by the mesial and distal cingula (Fig. 8L). The central cusp of the lower postcanine is, however, long and wide at its base and surrounded by merged mesial and distal basins, forming a subcircular groove. Three cuspules are preserved on the lingual part of the mesial cingulum and four on the labial portion of the distal one, and both cingula probably bore between six and eight cuspules. A mesiodistal constriction is present at the cervix in the first postcanine, which is also constricted at the same level on its labial surface. The morphology of the second to fifth lower postcanines of SAM-PK-5881a follows that of the more well-preserved distal teeth of the holotype. The size and CBR of these teeth increase up to the fifth lower postcanines (CBR of 1.6), then decrease more distally. The transverse crest is positioned at mid-length of the crown and perpendicular to the long axis of the mandible in all lower postcanines of SAM-PK-5881a. Only the fourth left lower gomphodont tooth of this specimen preserves the distal cingulum. The latter is formed by five cingular cuspules, three labiolingually wide in the labial two-thirds and two small cuspules adjacent to the unpreserved lingual cusp.

A comparative analysis of the upper and lower postcanines of the holotype provides major differences between them. Similar to the postcanines of the cranium, the lower gomphodont postcanines include a transverse crest composed of a labial, central, and lingual cusps ringed by the mesial and distal cingula. Yet, unlike the upper postcanines, large cingular cuspules appear to be present linguomesially and labio- and linguodistally in the lower gomphodont teeth (Fig. 8K). These cuspules are most likely homologous to the putative labiodistal, linguomesial, and linguodistal accessory cusps of Langbergia. Similar to the upper postcanines, the transverse crest is diagonally oriented from the long axis of the postcanine tooth row in the last lower gomphodont tooth, and perpendicular to this axis in more mesial postcanines, as seen in the eighth right (Fig. 8K) and what are assumed to be the fifth and sixth left lower postcanines (n.b., Crompton (1955) interpreted these two teeth as corresponding to the seventh and eighth lower postcanines, yet based on their size and the height of the preserved portion of the dentary, they are considered here as the fifth and sixth). Unlike the last upper gomphodont postcanine, the transverse crest of the ninth lower postcanine is only mildly diagonally oriented (Fig. 8K). The central cusp forms a centrally positioned dome on the transverse crest and, as the upper postcanines, its labial and lingual ridges are denticulated. It is unknown whether the central, mesial, and distal ridges of the labial and lingual cusps also bore denticles in the eighth and ninth right lower gomphodont postcanines, but the mesial and distal ridges of the labial cusp of the fifth? left lower postcanine is clearly serrated. The labial, central, and lingual cusps all share the same height in the eighth right lower postcanine, the only one having these cusps complete. Unlike the upper postcanines, the labiodistal surface is angular in apical view in the eighth and ninth right lower postcanines, the best preserved of the mandible, with the labiodistal margin connecting to the labial surface of the crown at a slightly obtuse angle (Fig. 8K). The mesial and distal margins of the fifth and sixth left and right lower postcanines are, however, gently convex, with no angular corner. There are 10 mesial and seven distal cingular cuspules on the cingula in the ninth right lower postcanine, the only lower crown preserving the majority of these elements. Both cingula extend along the whole width of the postcanine, and the smallest cingular cuspules are found in the central portions of the cingulum. The mesial and distal cingula are badly preserved in the fifth? and sixth? left lower gomphodont teeth but the labialmost cingular cusps are particularly large on both cingula. In the fifth? left postcanine, the labiomesial cingular cuspule is significantly larger than the labiodistal one, and strongly protrudes apically, as seen in Langbergia. The distal cingulum is almost complete in this tooth and preserves six cingular cusps increasing in width and height lingually. Both upper and lower postcanines of the holotype show a braided and apicobasally oriented enamel surface texture best visible on the mesial and distal surfaces of the central cusp (Fig. 8N).

The preserved lower gomphodont postcanines of the referred specimens NHMUK PV R36800 and NHCC LB28, on the distal portion of a mandible, show some differences with those of the holotype. As in the latter, there is a decrease in size and elongation in the distalmost postcanines of the mandible in NHMUK PV R36800, yet the last two lower gomphodont postcanines are particularly small and subcircular in outline in NHMUK PV R36800. As seen in the last lower gomphodont tooth of the holotype, the long axis of the antepenultimate lower gomphodont postcanine is slightly diagonally oriented from the labial margin of the mandible. The long axis of the two last lower gomphodont postcanines is, however, perpendicular to the long axis of the mandibular tooth row. The transverse crest of the preserved lower postcanines comprises three high subtriangular cusps, of which the central cusp is the widest, even in the subcircular postcanines. The labial, lingual, and central cusps appear to share the same height. Unlike the holotype, the labiodistal margin of the lower gomphodont postcanines of NHMUK PV R36800 is not angular but smoothly convex, whereas the linguodistal surface is concave in the third and second last gomphodont teeth. Six distal cingular cuspules decreasing in size toward the central part of the cingulum can be counted in the antepenultimate gomphodont postcanine. Likewise, there are at least seven mesial cingular cuspules in the last gomphodont postcanine, and their size diminishes toward the mid-width of the crown. As in the holotype, the enamel surface texture, well-visible on the surface of the central cusp of NHMUK PV R36800 is braided.

Unlike the lower gomphodont postcanines of the holotype, the eighth right lower tooth, the only lower gomphodont postcanine preserved in NHCC LB28, is subrectangular in outline in apical view, with a nearly straight lingual surface and a biconvex distal margin (Sidor & Hopson, 2018). As seen in the ninth right lower postcanine of the holotype, the crown shows a weakly diagonally oriented transverse crest made of labial, lingual, and central cusps, as well as a notch in the labiodistal corner of the crown, behind the labial cusp. The central cusp is centrally positioned on the crown and 1.5 times wider than the subequal width of the labial and lingual cusps. On the transverse crest, the labial is the highest cusp and the central the lowest (Sidor & Hopson, 2018). The mesial and distal basins surround the transverse crest and are bounded by the cingula, respectively. The mesial cingulum is restricted to the labial two-thirds of the crown and comprises seven cingular cuspules whose width changes sporadically along the cingulum. From labial to lingual, there are three small cuspules, two large and elongate ones, and two minutes cuspules on the mesial cingulum (Sidor & Hopson, 2018). The distal cingulum includes a larger number of cuspules (i.e., eight or nine), with the widest cuspules located in the labial portion of the cingulum distal to the labial cusp, as seen in the holotype. The mesial basin extends from the labiomesial surface of the lingual cusp to the labialmost mesial cingular cuspules, whereas the distal basin extends from the distal side of the lingualmost distal cingular cuspule to the notch on the labiodistal surface of the crown (Sidor & Hopson, 2018).

Sectorial postcanines

An isolated sectorial postcanine (Figs. 8P and 8Q) belonging to the holotype was illustrated and briefly described by Hendrickx, Abdala & Choiniere (2016). This tooth most likely represents the shearing-type crown underlying the ninth right upper postcanine described by Crompton (1955). The postcanine consists of a mesiodistally long main cusp adjacent to a poorly developed mesial main cusp, and a well-developed subtriangular and apically pointed distal main cusp. Given this morphology, we assume that this is the distalmost tooth of the sectorial tooth row, probably the 10th right upper postcanine, as suggested by Crompton (1955). The mesial and distal carinae are denticulated along the apical portion of the main cusp and the serrations extend across the apex of the cusp (Figs. 8P and 8Q). The denticles are low, apicobasally elongated and show a symmetrically convex external margin. There is no trace of a cingulum on the labial or lingual margins of the crown.

The lower sectorial postcanines partially visible in the left and right 11th alveoli of NHCC LB28 and comprehensively describe by Sidor & Hopson (2018), provide for the first time information on the sectorial postcanines from the mandibular region of Cricodon. Unlike the upper sectorial postcanine of the holotype, the lower sectorial tooth is significantly longer, wider and lower. It is diagonally oriented at an angle of 45° from the rest of the jaw and includes a large subtriangular main cusp, two smaller distal and mesial main cusps, and a distal accessory cusp (Sidor & Hopson, 2018). The mesial main cusp is a low but long convexity lying much lower than the distal main cusp. It is separated from the main cusp by a low valley and is slightly lingually deflected from the main and distal main cusps. The latter occupies the distal fourth of the crown and forms a low subtriangular prominence protruding vertically. It is followed by the distal accessory cusp, an even lower convexity on the distal margin of the distal main cusp. Unlike the preserved upper sectorial postcanine of the holotype, the lower sectorial tooth of NHCC LB28 includes a prominent lingual cingulum enclosing a shallow and labiolingually narrow lingual basin (Sidor & Hopson, 2018). The lingual cingulum comprises five cingular cuspules, that is, a small mesialmost one followed distally by three mesiodistally long cuspules, which increase in length distally (Sidor & Hopson, 2018, figure 5F). A fifth cingular cuspule is present at the distal extremity of the cingulum but too incomplete to provide information on its size. The cingulum extends from the mesial main cusp to the distal accessory cusp, as revealed by the presence of a cingular cuspule distal to the distal accessory cusp in the left postcanine (Sidor & Hopson, 2018). Two lower sectorial postcanines are also present in NHMUK PV R36800 but they are poorly preserved. The first sectorial postcanine is being replaced by a mesiodistally elongated gomphodont? postcanine with a narrow and apically pointed lingual cusp and a much longer and higher central cusp. Both lower sectorial postcanines include a large recurved main cusp bounded mesially and distally by small mesial and distal main cusps. Although the mesial main cusp of the first sectorial tooth appears to form a relatively large convexity, that of the second sectorial postcanine corresponds to a small protuberance at one-third of the crown height, slightly more basal than the distal main cusp. The latter is unpreserved in the first sectorial postcanine and incomplete in the second one. Based on the preserved portion of the cusp, the distal main cusp is much longer and more prominent than the mesial main cusp in this tooth. No lingual cingulum or denticulated carinae appear to be visible in these two lower sectorial postcanines but their presence cannot be ruled out.

Crown microstructure

Hendrickx, Abdala & Choiniere (2016) recently explored the enamel and dentin microstructure in two isolated gomphodont and sectorial postcanines of the holotypic specimen of Cricodon metabolus. This study revealed that the enamel microstructure of this taxon is prismless and composed of discontinuous columnar divergence units (SCE). Abundant tubules and around 19 irregularly spaced striae of Retzius were also observed in the enamel layer, which is 11.5 times thicker in gomphodont postcanines (176.5 μm in average in horizontal section) compared to sectorial teeth (15.4 μm in average in horizontal section). The dentin layer of the Cricodon postcanine includes a large amount of tubules and approximately 100 incremental growth lines of von Ebner were counted in the gomphodont tooth.

Trirachodon berryiSeeley, 1894

Holotype: NHMUK PV R3579 (from Lady Frere, subzone B, Cynognathus AZ, South Africa), the mesial portion of the cranium with poorly preserved teeth.

Referred dental material: from Burghersdorp, subzone B, Cynognathus AZ, South Africa: AM 434 (Cragievar locality), 461 (holotype of Trirachodon kannemeyeri), BP/1/3775 (Cragievar locality), NHMUK PV R2807, SAM-PK-987; from Aliwal North, subzone B, Cynognathus AZ, South Africa: NHMUK PV R3306, R3307, R3350, R3721 (holotype of Trirachodon browni), SAM-PK-873 (holotype of Trirachodon minor), 5880; from Winnaars Baken, subzone B, Cynognathus AZ, South Africa: BP/1/3511, 4258, 5050, SAM-PK-12168 (= K5821), K170, K171, K7888; NMQR 1399; from Lady Frere, subzone B, Cynognathus AZ, South Africa: BSP 1934 VIII 21, 22, 23; from Rouxville, subzone A, Cynognathus AZ, South Africa: BP/1/4658, 4661 (Leewspuit locality), CGP/1/33 (Moerbeidal locality), SAM-PK-K10157, K10161, K10176, K10207, K10411; from Kestell, subzone A, Cynognathus AZ, South Africa: NMQR 3279; from Bergville, subzone A, Cynognathus AZ, South Africa: CGP/1/79 (= CGP JNN 2000-7-2A); from unknown localities in South Africa: SAM-PK-K142, K4801, K4803, NMQR 122; from Etjo Mountain, Omingonde Formation, Namibia: GSN R327.

Occurrence: Lady Frere, Chris Hani District, Emalahleni Municipality, South Africa; Burghersdorp, Aliwal North, and Winnaars Baken, Joe Gqabi District, Walter Sisulu Municipality, South Africa; Lemoenfontein 44 and Moerbeidal, Rouxville, Xhariep District, Free State Province, South Africa; Bergville, Thabo Mofutsanyane district, Maluti-a-Phofung Municipality, Free State Province, South Africa; western buttress of Etjo Mountain, Otjozondjupa Region, Namibia.

Horizon: subzone A and B of the Cynognathus AZ, Burgersdorp Formation, Beaufort Group, Karoo Supergroup; lower? assemblage of the upper Omingonde Formation.

Age: late Olenekian–early Anisian, Middle Triassic.

Dental formula: i4/3 : c1/1 : pc6-12/8-9 (cpc1 : gpc4-10 : spc0-2).

Dental morphology: The dentition of Trirachodon has received particularly little attention compared to other gomphodonts, and information on the dental anatomy of this taxon mostly relies on the rather limited descriptions and illustrations of Seeley (1894) and Broili & Schröder (1935). Firsthand examination of Trirachodon specimens with the best-preserved dental material now allows for a comprehensive description of the dentition of this taxon. Dental differences in the specimens referred to T. berryi and T. kannemeyeri, considered to be separated species by several authors (Sidor & Hopson, 2018), are also considered in the discussion.

Incisors

The incisors, well-preserved in the specimens CGP/1/79 (= CGP JNN 2000-7-2A), SAM-PK-K171, SAM-PK-12168, and BP/1/4658, have denticulated mesial and distal carinae at least in the first upper and lower incisors of SAM-PK-12168 (Figs. 9A–9C) and SAM-PK-K171, respectively. Mesial denticles are also visible in the third upper incisor of BP/1/4658 and an isolated upper? incisor of CGP/1/79. No denticles could be seen in the lower incisors of BP/1/4658 and 4661 and some upper and lower incisors of CGP/1/79. It is unknown whether the absence of denticles in some upper and lower incisors is a genuine condition of Trirachodon. The presence of unserrated canines in some specimens would, however, suggests so. The mesial denticles of SAM-PK-12168 are well-separated, apicobasally elongated and almost subquadrangular at the base of the carina (Fig. 9B), whereas those of CGP/1/79 are poorly defined, particularly low and apicobasally subrectangular all along the carina. As other diademodontids and trirachodontids, they sporadically change in size and their external margin is weakly symmetrically to asymmetrically convex. Unlike Cricodon, which features much larger denticles, there are six and 10 mesial denticles per mm in the Trirachodon specimens SAM-PK-12168 and BP/1/4658, respectively. The largest denticles are situated apically, and given the fact that the apex and basal portions of the crown are missing, it is unknown whether the denticles reached the apex and cervix. The distal denticles of the first and second incisors of SAM-PK-12168 are incipiently developed and form poorly delimited convexities changing sporadically in size along the carina (Fig. 9C). These denticles appear to extend well-above the cervix and get flared at mid-crown. Some upper and lower incisors are weakly mesiodistally constricted at the cervix, conferring a folidont (i.e., leaf-shaped) morphology to the crown. The cross-sectional outline of the upper and lower mesial incisors is D- to salinon-shaped, with the mesial carina positioned mesially and the distal carina facing linguodistally (Figs. 9D and 9E). The incisors from the distal half of the jaw appear to have a more semi-circular cross-sectional outline, with the distal carina facing distally. Similar to Langbergia, a concave surface adjacent to the distal margin of the crown is visible on the labial surface of the second? upper incisor of SAM-PK-12168.

Figure 9 Dentition of Trirachodon berryi.

(A–C) First? left upper incisor, with close up on (B) mesial and (C) distal denticles of SAM-PK-12168 in labial view; (D) first left and right upper and (E) first right lower incisors of BP/1/4658 in apical views; (F and G) right upper canine of BP/1/4661 with (G) close up on the centrodistal part of the crown, in labial view; (H) distal denticles of the right lower canine of BP/1/4658 in labial view; (I) fifth and sixth right upper gomphodont postcanines of SAM-PK-K171 (“T. kannemeyeri” morphotype) in apical view; (J) 10th and 11th left (reversed) upper gomphodont postcanines of BSP 1934 VIII 21 (T. berryi morphotype) in apical view; (K) penultimate right upper gomphodont postcanine of SAM-PK-K4801 (“T. kannemeyeri” morphotype) in apical view; (M) penultimate left upper gomphodont postcanine of NHMUK PV R3307 (“T. kannemeyeri” morphotype) in apical view; (N) 10th left upper gomphodont postcanine of BSP 1934 VIII 21 (T. berryi morphotype) in apical view; (O) partially erupted twelfth (and last) upper gomphodont postcanine of BSP 1934 VIII 21 in distal view. Abbreviations: cc, central cusp; db, distal basin; dcc, distal cingular cuspule; dci, distal cingulum; ent, enamel surface texture; lac, labial cusp; ladc, labiodistal accessory cusp; lamc, labiomesial accessory cusp; lic, lingual cusp; lidc, linguodistal accessory cusp; limc, linguomesial accessory cusp; lri, longitudinal ridge; mb, mesial basin; mcc, mesial cingular cuspule; mci, mesial cingulum; tun, transverse undulation.

Canines

The canines of the specimens of Trirachodon CGP/1/79, SAM-PK-K171, 12168, K10157 and BP/1/4658, 4661 and 3511 have both mesial and distal carinae denticulated. Unserrated canines are, however, present in the specimens BSP 1934 VIII 21 and 22 as well as SAM-PK-K7888. The denticles are minute, sometimes barely discernable, and always apicobasally elongated (Fig. 9H). They sporadically change in size along the carina and have symmetrically to asymmetrically convex external margins. In some specimens, the denticles are incipiently developed or restricted to a small portion of the crown (e.g., BP/1/3511, SAM-PK-12168). The denticles of the canines are larger than those of the incisors, with mesial and distal denticle density ranging from three to nine denticles per mm in the canines. In BP/1/4658, the mesial carina is lingually deflected whereas the distal carina is centrally positioned on the distal margin of the crown. An apicobasally long concave surface adjacent to the distal carina is clearly present on the labial and sometimes lingual sides of the canine in BSP 1934 VIII 22, CGP/1/79, SAM-PK-K171, 12168, K10157, and BP/1/4658, 4661. A similar concave surface adjacent to both carinae on the lingual side can be seen in the lower canine of SAM-PK-K171. Transverse undulations and several faint longitudinal ridges can be observed on the labial surface of the right canine of BP/1/4661 (Figs. 9F and 9G). Flat or mesiodistally convex surfaces meeting at acute corners are instead visible on the labial surface of one or both canines of SAM-PK-12168 and CGP/1/79. The cross-sectional outline of the canines is elliptical, lenticular, or salinon-shaped at mid-crown, indicating some important variation in cross-sectional outline in Trirachodon. The enamel surface texture of BP/1/4661 and CGP/1/79 is clearly braided and apicobasally oriented, curving basally adjacent to the carinae (Fig. 9G). The texture of the enamel appears to be smooth in the canines of the other specimens.

Upper gomphodont postcanines

The morphology of the upper postcanines is based on the specimens BSP 1934 VIII 21, SAM-PK-K171, SAM-PK-K4801, NHMUK PV R3307, and BP/1/4658. As in Cricodon and Sinognathus, the upper postcanines are labiolingually expanded and ovate in outline. They increase in size and elongation distally up to the antepenultimate or penultimate gomphodont tooth. The first upper postcanine is the smallest and, with a CBR ranging from 1.1 to 1.6, the least labiolingually elongated of the upper gomphodont tooth row. This tooth is missing, badly preserved or in occlusion and obscured by the lower postcanines in all specimens examined first hand but CT-scan data of AM 461 provides some information on its morphology. The tooth is subconical, that is, the crown is simple, slightly labiolingually elongated, labiolingually constricted at the cervix and bears a centrally positioned central cusp but no labial and lingual cusps nor mesial and distal cingula.

The CBR of the more distal upper postcanines varies from 1.4 to 2.4 and the outline of the crown in apical view only slightly changes along the tooth row. In BSP 1934 VIII 21 and BP/1/4658, the postcanines from the mesial half of the upper postcanines are reniform (sensu Hendrickx, Mateus & Araújo, 2015), with a weakly to strongly concave mesial margin. This outline results from the presence of a distal “shouldering” between each tooth, which are in close contact in these two specimens. In SAM-PK-K171 and SAM-PK-K4801, the reniform outline is restricted to the third and fourth upper postcanines, yet the mesial margin is only weakly concave in the fourth tooth. Although the distal margin of the upper postcanines from the mesial half of the Trirachodon crown is typically convex, this distal margin is sigmoid in BSP 1934 VIII 21 due to the presence of a narrow concavity on the lingual third of the distal surface of the crown. This dental feature is, however, restricted to this specimen. The more distal upper postcanines of Trirachodon are roughly ovate in outline, but a weak mesiodistal constriction occurs at the labial third of the crown in SAM-PK-K171, SAM-PK-K4801, and NHMUK PV R3307 (Figs. 9I and 9K–9M). This constriction, created by the presence of a concavity on the labial third/half of the distal margin of the postcanine, leads to an asymmetrical crown-shape, in which the labial third is shorter than the lingual two-thirds of the postcanine (Figs. 9I and 9K–9M). Such a concavity is particularly pronounced in the last upper gomphodont postcanines. It is, however, absent in the postcanines of BSP 1934 VIII 21 and BP/1/4658 so that the upper postcanines are subsymmetrical in shape in these two specimens (Fig. 9J).

As other trirachodontids, the upper postcanines of Trirachodon consist of a mesial and distal cingula extending on the mesial and distal rims of the crown as well as a labiolingually elongated transverse crest. The latter is made of a labial, central and lingual cusps and bounded mesially and distally by the wide mesial and distal basins, respectively (Figs. 9I–9K). The transverse crest is centrally positioned on the crown in the postcanines from the mesial half of the upper tooth row in BP/1/4658, SAM-PK-K171, and NHMUK PV R3307, and distally deflected in BSP 1934 VIII 21. Unlike Cricodon, the transverse crest runs diagonally on the crown in the distalmost gomphodont postcanines. The diagonal orientation of the transverse crest, particularly noticeable in NHMUK PV R3307, SAM-PK-K171, and SAM-PK-K4801, results from the distal deflection of the labial cusp, which points apicodistally in these specimens (Figs. 9I and 9K–9M). The central cusp is closer to the lingual cusp in most specimens, and centrally positioned on the transverse crest in SAM-PK-K4801. The central cusp is typically apicolingually inclined in distal view, so that the valley bordering the central cusp on the transverse crest is deeper between the central and labial cusps. The latter consists of a three-faced pyramidal structure, in which the edges form the labiomesial, labiodistal, and labiocentral ridges. The lingual cusp shares the same shape, with a labiodistally shorter linguocentral ridge. Similar to Cricodon, the labiomesial and labiodistal ridges of the labial cusp are denticulated in at least one postcanine of SAM-PK-K171 and SAM-PK-K4801. The linguomesial, linguodistal, centrolabial, and centrolingual ridges do not, however, appear to have serrations. The mesial and distal cingula typically bear a low number (i.e., <6) of cingular cuspules restricted to a certain portion of the cingulum or along the whole width of the cingulum (Figs. 9M and 9N). We counted three, four, and five cingular cuspules on the mesial cingulum of the best-preserved postcanines of NHMUK PV R3307, SAM-PK-K171, and BSP 1934 VIII 21, respectively, and three to four, four, five, and six to seven distal cingular cuspules in SAM-PK-K171, BSP 1934 VIII 21, SAM-PK-K4801, and NHMUK PV R3307, respectively. The size of the cingular cuspules varies sporadically along the cingulum and does not follow any pattern along the tooth row. We, however, note that the widest cingular cuspules are often located in the middle or lingual part of the crown. When restricted to a certain portion of the postcanine, the cingular cuspules variously occupy the central, lingual or labial part of the crown. Large cingular cuspules have been observed in the labiomesial, labiodistal, linguomesial, and linguodistal sides of the fully complete postcanines. Given the fact that none of these large cingular cuspules are present in all postcanines of the tooth-row, they may not be homologous to the accessory cusps of some gomphodonts. The presence of labiomesial, labiodistal, linguomesial, and linguodistal accessory cusps in the upper gomphodont postcanines Trirachodon cannot, however, be ruled out.

The morphology of the distalmost upper gomphodont postcanine of the holotype and referred specimens NHMUK PV R3579 and BSP 1934 VIII 21, respectively, which represent the specimens of Trirachodon with the highest number of gomphodont postcanines (i.e., 11 and 12, respectively), departs from that of the more mesial upper postcanines. In NHMUK PV R3579, the last postcanine is half the size of the preceding teeth, elliptical, long and short (CBR of 1.3) and displays a centrally positioned transverse crest ringed by the mesial and distal basins and cingula. The transverse crest includes a long and wide labial cusp, a lingually displaced central cusp and a narrow lingual cusp. The labial cusp, of which the base is preserved, was the largest and likely the highest of all three, while the central and lingual cusps appear to have the same height. The mesial cingulum seems to encompass four cingular cuspules, in which the widest, which may correspond to the labiomesial accessory cusp, is directly mesial to the labial cusp. Two particularly wide distal cingular cusps adjacent to the labial and lingual cusps may also represent the labio- and linguodistal accessory cusps of other gomphodont upper postcanines. Unlike NHMUK PV R3579, the last upper postcanine of BSP 1934 VIII 21 appears to be strongly labiolingually elongated and includes a long, high and lingually deflected central cusp and a much shorter, apically pointed labial cusp (Fig. 9O). Unlike NHMUK PV R3579 and more mesial gomphodont postcanines, this tooth does not have a lingual cusp nor a distal basin and cingulum. Given that the mesial portion of the tooth is still embedded in matrix, it is unknown whether the mesial basin and cingulum were present or not. The root is twice as long as the crown at mid-width of the tooth, tapers apically and has a basolingually inclined lingual margin and a sub-vertical labial border (Fig. 9O). The basolingual portions of the crown and root of BSP 1934 VIII 21 also show an extensive parallelogram-shaped wear facet on its distal surface.

Lower gomphodont postcanines

Well-preserved lower gomphodont postcanines are present in SAM-PK-K171, SAM-PK-K4801, and BP/1/4658. The first postcanine, only seen in BP/1/4658 (Fig. 10D), is significantly smaller than the more distal postcanines, as witnessed by the particularly small size of the alveoli in SAM-PK-K171. Its morphology does not differ from that of more distal gomphodont postcanines (Fig. 10D). Similar to Langbergia and unlike Cricodon, the long axis of the lower gomphodont postcanines is linguomesially oriented and never perpendicular to the mandibular tooth row (Figs. 10A–10C). The linguomesial inclination of the first and last gomphodont teeth is more pronounced than the others in BP/1/4658 and SAM-PK-K171. The lower postcanines decrease in size from the middle of the mandibular tooth row in BP/1/4658, so that the last gomphodont tooth is as small as the first postcanine. As in Langbergia and unlike Cricodon, such decrease in size occurs from the penultimate lower postcanine in SAM-PK-K171 and SAM-PK-K4801. The lower gomphodont postcanines share the same morphology than that of upper postcanines, yet they are symmetrical in the transverse and sagittal axes and narrower, with a CBR varying from 1.3 to 1.6 (Figs. 10A–10C). The central cusp is centrally positioned on the crown and occupies most of the transverse crest width. As in other trirachodontids, this cusp is bordered by the labial and lingual cusps, and the three cusps all share the same height. The mesial and distal basins are deep and both surround the dome-like central cusp mesially and distally, respectively (Figs. 10A–10C). The mesial and distal cingula are symmetrically concave and parabolic in shape in mesial and distal views. They include a variable number of cingular cuspules extending between the labial and lingual cusps. There are two to three and four cingular cuspules on the mesial cingulum in the best-preserved lower postcanines of SAM-PK-K171 (Figs. 10A and 10E) and SAM-PK-K4801 (Figs. 10B and 10C), respectively, and four distal cingular cuspules in both specimens. Six to seven cingular cuspules appear to be present on the distal cingulum of the fifth right and left postcanines of BP/1/4658, respectively, while more than seven distal cuspules were present on the sixth left lower tooth. Unlike Cricodon, the widest cuspules typically occur in the center or labial portions of the crown. The last lower gomphodont postcanine of SAM-PK-K4801 (Fig. 10C) has the peculiarity of having a particularly high transverse crest, higher than that of more mesial postcanines. The central cusp strongly protrudes apically and is higher than both the labial and lingual cusps. The distal cingulum comprises six well-separated cingular cuspules whereas the mesial cingulum includes three cuspules in its labial half (Fig. 10C). The last lower postcanine of SAM-PK-K171 and BP/1/4658 shares the same morphology with the other lower postcanines. The enamel texture of Trirachodon’s upper and lower postcanines, clearly visible on the mesial and distal surfaces of the central cusp, is braided and apicobasally oriented.

Figure 10 Dentition of Trirachodon berryi.

(A) Fifth and sixth right lower gomphodont postcanines of SAM-PK-K171 in apical view; (B) penultimate and (C) ultimate (and basolingually rotated) right lower gomphodont postcanines of SAM-PK-K4801 in apical view; (D) first left lower gomphodont postcanine of BP/1/4658 in apical view; (E) fifth (bottom) and sixth (top) right lower gomphodont postcanines of SAM-PK-K171 in apicomesial view; (F) first (left) and second (right) right upper sectorial postcanines of SAM-PK-K4801 in labial view; (G and H) first right lower sectorial postcanine of SAM-PK-K4801 in (G) labial and (H) lingual views; (I and J) first and second (erupted and replacing tooth) right lower sectorial postcanines of BP/1/4658 in (I) labial and (J) apical views. Abbreviations: cc, central cusp; dac, distal accessory cusp; db, distal basin; dcc, distal cingular cuspule; dci, distal cingulum; dmc, distal main cusp; lac, labial cusp; lacc, labial cingular cuspule; laci, labial cingulum; lic, lingual cusp; licc, lingual cingular cuspules; lici, lingual cingulum; mb, mesial basin; mc, main cusp; mca, mesial carina; mcc, mesial cingular cuspule; mci, mesial cingulum; mmc, mesial main cusp; pespc, partially erupted sectorial postcanine.

Sectorial postcanines

One or two upper and lower sectorial teeth are borne by the specimens SAM-PK-K4801, SAM-PK-K171, SAM-PK-12168, BP/1/4658, BP/1/4661, BP/1/3511, and CGP/1/79. The first right upper sectorial tooth of SAM-PK-K4801 is simpler and shorter than the second one (Fig. 10F). It includes a main distally positioned and recurved cusp followed mesially by a smaller mesial main cusp. This mesial main cusp is symmetrically convex and projects from the mesial surface at one-third of the crown. The single or distalmost upper sectorial postcanine shares the same morphology, that is, it comprises a main cusp, a mesial main cusp at one half of the crown and two distal cusps, the distal main and distal accessory cusps, both occupying the distal third of the crown (Fig. 10F). The main cusp typically occupies the two-fifths to three-fifths of the crown length. It consists of a subtriangular, apicodistally pointed or strongly recurved (Fig. 10F) projection, and its external margin is strongly parabolic and asymmetrically convex. The mesial main cusp can be minute and forms a tiny bump on the mesial surface of the main cusp, or a large, apically pointed projection similar in size or slightly longer than the distalmost accessory cusp. The distal main and distal accessory cusps are straight, decrease in size distally, and either point apically or apicodistally. The distal main cusp either forms a low, weakly pointed and mesiodistally long convexity or a high and mesiodistally short projection. No cingular cuspule were observed on the labial and lingual margin of the upper sectorial postcanines of Trirachodon. Variation in the morphology of the last sectorial appears to occur between the left and right sides of the cranium, as seen in SAM-PK-K4801. In this specimen, the mesial main cusp of the last upper sectorial postcanine consists of a poorly-developed bump on the right side (Fig. 10F) and a large and strongly protruding projection on the left side. Likewise, the distal main cusp of this sectorial postcanine is mesiodistally short and high on the right side (Fig. 10F) and long and low on the left side.

The morphology of the single or first lower sectorial postcanine is significantly different from that of the upper sectorial teeth. The crown is particularly wide, especially in BP/1/4658 (CBR of 0.6–0.65). The crown consists of a large and high apically pointed main cusp bordered by two small apically projected mesial and distal main cusps (Fig. 10I). The distal main cusp is larger than the mesial one in SAM-PK-K171 and BP/1/4658. The three cusps are bounded lingually by a mesiodistally concave cingulum made of minute cingular cuspules (Fig. 10G). This lingual cingulum either extends along the whole length of the crown or is restricted to the mesial half of the postcanine, as seen in SAM-PK-K171. A labial cingulum is also present on the lower sectorial postcanine and occupies the whole length, as seen in the right lower postcanine of BP/1/4658 (Fig. 10I), or a small portion of the distal surface of the crown only, as in SAM-PK-K4801 (Fig. 10H).

The second lower postcanine, only well-preserved on the left jaw of BP/1/4658 (Figs. 10I and 10J), is narrower, with an CBL of 0.54. The tooth includes a main cusp occupying slightly more than the mesial half of the crown, as well as a distal main cusp and a distal accessory cusp. No mesial main cusps appear to be present in the second left and right lower sectorial postcanines. The distal main cusp points apically and is not recurved (Figs. 10I and 10J). The main cusp, of which only the base is visible, was likely significantly higher than the distal main cusp on the left side of the jaw. The distal main cusp of the second right sectorial postcanine is, however, much larger and probably only slightly lower than the main cusp. The distal main cusp consists of a symmetrically parabolic convexity while the distal accessory cusp is a minute and pointed projection (Figs. 10I and 10J). A lingual cingulum bearing several minute cuspules extends along the whole length of the crown (Fig. 10J). The second right lower sectorial postcanine of BP/1/4658 is significantly shorter than the preceding one (65% smaller) and is being replaced by another sectorial postcanine (Figs. 10I and 10J). This tooth erupts distally from the second sectorial postcanine, and its morphology is reminiscent to that of the upper sectorial postcanines in that it has an apically recurved main cusp. No mesial or distal main cusps appear to be present on this erupting postcanine but they might be hidden by the matrix. The main cusp has an unserrated mesial carina twisting lingually and no cingulum is present at the base of the lingual surface of the crown (Fig. 10J).

Dental morphotypes in Trirachodon berryi

As noticed by previous authors such as Sidor & Hopson (2018), two dental morphotypes of Trirachodon can be observed. AM 461, SAM-PK-K171, SAM-PK-12168, NHMUK PV R3307 (included in “Cricodon kannemeyeri” by Sidor & Hopson (2018), see Discussion below), as well as CGP/1/79, SAM-PK-K4801, BP/1/4661, and BP/1/3511, share a combination of dental features not seen in other specimens of Trirachodon, that is, denticulated canines, less than 10 upper postcanines comprising one or two upper and lower sectorial teeth, transversely asymmetrical distal upper gomphodont postcanines, in which the lingual half of the crown is longer than the labial one, a diagonally oriented transverse crest on the crown, and a slightly apicodistally pointed labial cusp. These specimens indeed differ from NHMUK PV R3579, BSP 1934 VIII 21–23, and SAM-PK-K7888 (classified among Trirachodon berryi by Sidor & Hopson (2018), see below), characterized by the following dental features: unserrated canines, more than nine upper postcanines, no sectorial teeth, transversely symmetrical distal upper postcanines (i.e., labial and lingual halves of the crown of the same length, apically pointed labial cusp, and non-diagonally oriented transverse crest on the crown) and last upper gomphodont postcanine being significantly smaller than the preceding tooth.

It should, however, be noted that the specimen NHMUK PV R3307 does not have a sectorial postcanine on the right portion of the maxilla while a particularly low number of upper postcanines (i.e., seven) is present in this specimen. The absence of a diagonally oriented transverse crest in the upper postcanines of BSP 1934 VIII 21 could also be explained by the fact that the labial cusp of some gomphodont postcanines is worn out. All specimens classified in the morphotype of Trirachodon berryi do not preserve incisors and only two of them (i.e., BSP 1934 VIII 21, SAM-PK-K7888) bear canines. The latter are, however, incomplete and/or badly preserved and the absence of denticles could be explained by the fact that they have been worn out from the carinae due to the advanced age of the individuals (see Discussion). We also note that BP/1/4658 appears to represent a transitional form between the dental morphotypes of T. berryi and “T. kannemeyeri,” as this specimen includes finely denticulated canines, subsymmetrical and non-mesiodistally constricted upper gomphodont postcanines with a non-diagonally oriented transverse crest and a complete distal cingulum (present at least in the fifth right upper postcanine), penultimate and last upper gomphodont postcanines with a CBR >2, seven upper postcanines, as well as one upper sectorial postcanines and two lower tricuspid sectorial postcanines bearing a lingual cingulum. We propose that the dental differences between these two morphotypes are ontogenetic and not interspecific, as some authors suggest, an hypothesis developed in the “Discussion” section.

Sinognathus gracilisYoung, 1959

Holotype: IVPP V2339, an almost complete skull.

Occurrence: Peipanching Shihpi, Wuhsiang-Yüshe area, Shanxi Province, China.

Horizon: upper Ermaying Formation.

Age: late Anisian, Middle Triassic.

Dental formula: i4/2 : c1/1 : pc6/7 (cpc? : gpc5 : spc?).

Dental morphology: The dentition of Sinognathus gracilis, briefly described by Young (1959) due to the lack of preparation of the skull, has received a relatively good description from Sun (1988) following a re-preparation of the postcanines teeth. Firsthand examination of the holotype provides a few additional details on the dental material of this taxon.

Incisors and canines

The incisors, poorly preserved in IVPP V2339, do not show a constriction at the level of the cervix (Fig. 11A). Minute denticles can be seen at the base and apex of the distal carina of the fourth left upper incisor (Figs. 11B and 11C). We counted eight denticles per mm on this tooth. No other incisors appear to bear serrations. The cross-section through the root base of the first left upper incisor is subcircular (Fig. 11D). The root base of the left and right upper canines, the crown base of the right lower canine as well as the root, crown base, and a small portion of the mesial part of the left lower canine are preserved (Figs. 11E and 11F). None of these portions of the canines show serrations (Fig. 11F) and it is unknown whether denticles are unpreserved or absent. The cross-sectional outline through the root base of the left upper canine is roughly lenticular (Fig. 11G). The left lower canine is mostly projected apically, whereas the right canine appears to be weakly procumbent. The root base of the left lower canine is slightly longer than the crown base. The root is two-thirds the height of the crown and tapers basally (Fig. 11E).

Figure 11 Dentition of Sinognathus gracilis (IVPP V2339).

(A) Second left upper incisor in labial view; (B) fourth left upper incisor, with (C) close up on basodistal denticles, in labial views; (D) basal root cross-sectional outline of the first left upper incisor in apical view; (E) left and (F) right lower canines in labial view, with (G) close up on the cross-sectional outline through the root base of the left upper canine; (H) first to sixth upper and lower left gomphodont postcanines (sagittal cross-section through mid-crown), with (I) close up on the fourth upper gomphodont postcanine, in labial view; (J) right second to sixth upper (bottom) and second to seventh lower (top) gomphodont postcanines in linguodistal view (and apicodistal and basodistal views for the upper and lower gomphodont postcanines, respectively); (K and L) third right upper gomphodont postcanine in (K) apicodistal and (L) apicolingual views; (M) third right lower gomphodont postcanine in labial view; (N) fourth left lower gomphodont postcanine in labial view; (O and P) sixth (and last) upper gomphodont postcanine in (O) linguodistal and (P) apicolingual views. Abbreviations: cc, central cusp; dci, distal cingulum; de, denticle; lic, lingual cusp; limc, linguomesial accessory cusp; mci, mesial cingulum; ro, root; tc, transverse crest; I–VI, first to sixth gomphodont postcanines.

Gomphodont postcanines

All postcanines are in occlusion. Information on their morphology can, however, be extracted from the right upper postcanines, of which the lingual portion has been prepared and is visible in medioventral view (Figs. 11J–11L), and from a mid-crown sagittal section through the upper and lower postcanines of the left jaw side of the skull (Figs. 11H, 11I and 11N). Some parts of the labial portion of the right lower postcanines can also be seen in labial view (Fig. 11M). As described by Sun (1988), the upper postcanines are labiolingually elongated and consist of a labiolingually oriented and slightly distally positioned transverse crest. The latter is apically high and comprises labial, central, and lingual cusps of the same height. A small valley separates the central and lingual cusps in the best-prepared upper postcanine (Figs. 11J–11L). The central cusp is wide, the widest of the three cusps, and centrally located on the transverse crest. The mesial and distal ridges, visible in the sagittal cross-section through the fourth left upper postcanine, bound the mesial and distal margins of the gomphodont postcanines (Sun, 1988). These ridges are not visible in the lingual portion of the exposed upper postcanines, yet poorly preserved cingular cuspules appear to be present on what would then be the distal cingulum of the third upper postcanine (Fig. 11K). A linguomesial accessory cusp may have been present in the upper gomphodont postcanine (Fig. 11K). No other accessory cusps are visible in the upper postcanines. The mesial basin is longer than the distal one, and both basins are horizontal and shallow.

Few details on the lower gomphodont postcanines of the holotype can be observed. They are labiolingually elongated and bear an apically high transverse crest bounded by the mesial and distal basins. Both basins are mesiodistally concave, yet the distal basin is horizontally positioned and shallow whereas the mesial basin is basally inclined and shorter. The transverse crest is slightly mesially deflected and includes labial, central, and lingual cusps roughly sharing the same height. A faint mesial ridge is present at the rim of the lower postcanine. Remnants of mesial cuspules appear to be present on this ridge (Fig. 11M), but future CT-data of better preserved postcanines will corroborate this information. Based on the sagittal cross-section through the left lower postcanines, no distal ridge or cingulum appears to be present.

The sixth right upper postcanine was interpreted as a sectorial by Sun (1988). The crown is, however, incomplete, with the labial and central portion missing, and only an horizontal cross-section through the root can be seen in ventromedial view (Fig. 11O). A closer look at the cross-sectional outline reveals that the sixth upper tooth was a gomphodont and not a sectorial postcanine. This is corroborated by the fact that the preserved portion of this crown shows a labiolingually oriented transverse crest in the middle of the crown (Figs. 11O and 11P), whereas no sectorial postcanines bear such a ridge in gomphodont cynodonts. In addition, sectorial teeth/alveoli are typically labially positioned from the long axis of the tooth row in all gomphodont cynodonts (with the possible exception of Beishanodon; see discussion below), at the level of the labial half of the gomphodont postcanines. The preserved portion of the sixth upper postcanine is, however, at the same level as the lingual half of the preceding tooth (Fig. 11P). Based on firsthand examination of the specimen IVPP V2339, nothing supports the presence of sectorial postcanines in Sinognathus.

Beishanodon youngiGao et al., 2010

Holotype: PKUP V3007, an incomplete cranium.

Occurrence: Beishan locality, Quarry-3, in Beishan Hills, northern Gansu Province, China.

Horizon: Lower Triassic dark shales, Beishan fossil beds, Hongyanjing Formation.

Age: late Olenekian, Middle Triassic.

Dental formula: i4/? : c1/? : pc8/? (cpc2 : gpc7 : spc?).

Dental morphology: The dentition of this taxon, which is only known from six upper gomphodont postcanines, has been well-described by Gao et al. (2010). Because we have not had the opportunity to examine the holotypic specimen of Beishanodon, we rely on Gao et al.’s (2010) text and figures to describe the dentition of this taxon.

Incisors and canines

There are four upper incisors separated from the canine by a diastema whose length corresponds to that of two incisor alveoli. As noted by Gao et al. (2010), the outline of the incisor alveoli suggest that the root and crown base of the upper incisors were particularly labiolingually expanded and slightly mesiodistally shorter, especially in the first two incisors. The canine alveoli is positioned lateroposteriorly from the paracanine fossa, and its long axis is diagonally and labiomesially oriented from that of the cranium. A short diastema of the length of the first postcanine lies between the canine and postcanines.

Upper postcanines

The postcanine tooth row is straight, extends posteriorly to the level of the anterior margin of the orbit, and its long axis is directed toward the center of the subtemporal fenestra (Gao et al., 2010). Based on the size of the alveoli, the teeth increased in width and length up to the fifth postcanine, and the two last alveoli seem to have roughly the same size. Only the four first right and the fifth and sixth left upper postcanines are preserved. The first postcanine is the smallest of the tooth row and corresponds to a conical crown with a subcircular cross-sectional outline. A large cusp lies directly labial to the apex of this tooth (Gao et al., 2010). The second to fourth postcanines are gomphodont in shape and elliptical in outline, with their long axis perpendicular to the long axis of the tooth row but linguodistally oriented from that of the skull. The CBR of the preserved teeth varies between 1.25 and 2. The gomphodont postcanine includes a labiolingually oriented, weakly distally deflected, and labiolingually bowed transverse crest, with the apex of the concavity directed distally in apical view. The transverse crest is made of labial, central, and lingual cusps, with the central cusp being located in the labial half of the crown, close to the labial cusp (contra Gao et al., 2010). A distal cingulum comprising up to 10 cingular cuspules can be seen at the rim of the fourth right postcanine (Gao et al., 2010). The gomphodont postcanines do not seem to include any accessory cusps, and only the labiomesial and labiodistal ridges are clearly visible on the labial cusp in these teeth. It is unknown whether one or several upper sectorial postcanines were present in the distalmost portion of the tooth row, but the shape of the last right alveolus, which is longer than wide, would suggest that there was a sectorial postcanine in that position.

Discussion

The dentition of diademodontid and trirachodontid gomphodonts, which has one of the most complex occlusal pattern among non-mammaliaform cynodonts (Crompton, 1972), shows a large variety of dental features previously unknown or undescribed. The following homologous dental characters were shown to be present and relatively common in the dentition of non-traversodontid gomphodonts: (i) concave surfaces adjacent to the carinae in the incisors and canines; (ii) transverse undulations, longitudinal ridges and a sporadic variation in denticles size along the carinae in the canines; (iii) denticulated carinae in the incisors, canines, and sectorial postcanines; (iv) denticulated labial cusps, mesial, and distal cingular cuspules, and a braided enamel surface texture in the gomphodont postcanines; and (v) lingual and labial cingula in the sectorial postcanines. In diademodontids, we revealed the presence of denticles changing sporadically in size along the carinae in the canines, lingual cingular cuspules in the conical postcanines, denticulated labial cusp in the gomphodont postcanines and bi denticulated denticles in the sectorial postcanines in Diademodon, and a labiomesial accessory cusp as well as mesial cingular cuspules, in the upper gomphodont postcanines of Titanogomphodon. The presence of these dental characters in Titanogomphodon, however, needs confirmation with better-preserved specimens. In trirachodontids, we described for the first time denticulated carinae in the incisors and sectorial postcanines as well as longitudinal ridges in the canines of Langbergia, concave surfaces adjacent to the carinae in the canines and a biconvex distal cingulum in the upper postcanines of Cricodon, transverse undulations and longitudinal ridges in the canines as well as labial and lingual cingular cuspules in the lower sectorial postcanines of Trirachodon, and denticulated incisors and no sectorial postcanines in Sinognathus.

The affinities of the Chinese gomphodonts

According to Gao et al. (2010), the placement of Beishanodon within Trirachodontidae is supported by an ovoid-elliptical outline of the upper postcanines and a long axis of the postcanine tooth row directed toward the center of the subtemporal fenestra. The upper postcanines of Beishanodon also share with trirachodontids a low transverse crest made of a labial, central, and lingual cusps combined with a distal cingulum comprising a large number of cingular cuspules. The dentition of this taxon is nonetheless unusual for a trirachodontid in many aspects: the upper postcanine are particularly labiolingually narrow in relation to the skull length (the distance between the anteriormost point of the snout and the posteriormost point of the occipital condyles) and only Diademodon shares a low upper postcanine CBW/skull length ratio (Fig. 12). The first conical postcanine has a large labial cusp (Gao et al., 2010), a feature absent in other gomphodont taxa. In addition, the transverse crest of the postcanines three and four is located near the distal border of the tooth, presenting an occlusal basin that resembles a traversodontid postcanine. According to figure 6A from Gao et al. (2010), the central cusp of the second right postcanine is also strongly labiodistally positioned, a feature absent in all other trirachodontids. Finally, the sectorial postcanine(s), if present, would be weakly lingually deflected from the long axis of the tooth row (Gao et al., 2010, figure 5 and 6a) whereas all gomphodont cynodonts have their sectorial postcanine labial to this axis.

Figure 12 Comparison of skull length and widest upper gomphodont postcanine labiolingual width in gomphodont cynodonts, with linear regression trendlines for gomphodont taxa (in black) and Diademodon (in yellow).

Diademodon I, II, and III refer to specimens BSP 1934 VIII 14, MB R1004, and BSP 1934 VIII 19, respectively, Andescynodon I and II to specimens PVL 3836 and PVL 3894-1, Pascualgnathus I and II to specimens PVL 4416 and MLP 65-VI-18-1, and Massetognathus I, II, III, and IV to specimens BP/1/4245, PVL 4729, PVL 4727, and PVL 4613, respectively. Abbreviations: CBW, crown base width.

Although the latest cladistic analyses on gomphodont cynodonts (Liu & Abdala, 2014; Sidor & Hopson, 2018) recovered Beishanodon and Sinognathus within the clade Trirachodontidae + Traversodontidae, these two taxa were recently suggested to be probainognathians by Sidor & Hopson (2018) and basal probainognathians closely related to Aleodon brachyrhamphus by Hopson (2014) and Hopson & Sidor (2015). Nonetheless, the dentition of Aleodon, the only non-tritylodontid probainognathian with labiolingually elongated and ovoid postcanines (Crompton, 1955; Abdala & Giannini, 2002; Martinelli et al., 2017), strongly differs from that of Beishanodon and Sinognathus in the following features: (i) upper postcanine tooth row curved, with a long axis of the posterior portion directed toward the lateral part of the subtemporal fossa; (ii) postcanines strongly protruding from the alveoli; (iii) second and, in some cases, third upper postcanines subcircular and significantly smaller than the following teeth; (iv) no transverse crest, central cusps or mesial, and distal cingula in the postcanines; (v) labial cusp significantly larger (i.e., longer, wider, and higher) than the lingual “cusp” (when present) in the upper postcanines; and (vi) postcanines with a wide lingual platform extending from the sectorial/labial portion of the crown. Sidor & Hopson (2018) note that the two Chinese cynodonts also share probainognathian cranial characters such as the presence of a widely bowed zygomatic arch and the absence of a parietal foramen and a zygomatic/suborbital process of the jugal, three cranial features absent in basal cynognathians (i.e., Cynognathus, diademodontids, trirachodontids, and basal traversodontids) according to these authors. The zygomatic arch is, however, poorly anteroposteriorly convex in Sinognathus (IVPP V2339), showing the same convexity than some trirachodontid specimens (e.g., NMQR 3255, CGP/1/79). In fact, the temporal fenestra of Sinognathus is subrectangular in dorsal view, as in Langbergia and Trirachodon and unlike Lumkuia, Chiniquodon, and Aleodon. If a parietal foramen is indeed present in all African trirachodontids (including the Trirachodon specimen AM 461; Benoit, 2018, personal communication), it is, however, absent in the basal traversodontids Pascualgnathus (MLP 65-VI-18-1; PVL 4416) and Andescynodon (PVL 3836, 3840; Liu & Powell, 2009). Likewise, a suborbital process of the jugal is absent in the basal traversodontid Scalenodon (Abdala, Neveling & Welman, 2006) and more derived forms such as Boreogomphodon and Massetognathus (Abdala, Neveling & Welman, 2006; Sues & Hopson, 2010). For those reasons, we consider both Beishanodon and Sinognathus to be confidently classified among gomphodont cynognathians. Sidor & Hopson’s (2018) phylogeny recovered Sinognathus and Beishanodon among basal traversodontids, close to Pascualgnathus. Nevertheless, Sinognathus lacks the shearing planes between the outer surface of the main cusp of the lower and the inner surfaces of the main cusps of the upper postcanines characteristic of Traversodontidae. Likewise, the transverse crest of Sinognathus is centrally positioned on the crown, forms a low ridge and lacks the deep labial valley present in traversodontids. Consequently, this Chinese gomphodont is here confidently classified as a trirachodontid, pending a phylogenetic analysis to support this hypothesis, which will be conducted in a forthcoming contribution. Finally, given the presence of a single conical postcanine, six upper gomphodont postcanines comprising a distal cingulum and a low transverse crest made of a labial, central, and lingual cusps, a straight postcanine tooth row directed toward the centromedial portion of the temporal fenestra, and possibly one or two sectorial postcanines, we tentatively refer Beishanodon to Trirachodontidae. If correct, this reveals that diademodontids and some trirachodontids convergently evolved large skulls compared to the upper postcanine width, which remained particularly small in mature individuals.

Dental differences between Trirachodon berryi and “Trirachodon kannemeyeri”

Two species of Trirachodon, T. berryi, and “T. kannemeyeri,” were distinguished by Seeley (1894) based on the snout expansion and the number and labiolingual elongation of the maxillary teeth. Although these two taxa were synonymized by Hopson & Kitching (1972) (along with Trirachodon minor, Trirachodontoides berryi, and Inusitatodon smithi) and Abdala, Neveling & Welman (2006), they were considered as separate species by Kitching (1977) and more recently by Hopson & Sidor (2015) and Sidor & Hopson (2018). According to the two latter authors, T. berryi includes the specimens NHMUK PV R3579, BSP 1934 VIII 21–23 and SAM-PK-K7888, whereas the specimens AM 461, SAM-PK-K171, SAM-PK-12168 (= SAM-PK-K5821), and NHMUK PV R3307 belong to “T. kannemeyeri.” SAM-PK-11481 is also included in the latter taxon by Sidor & Hopson (2018), yet, as stated by Abdala, Neveling & Welman (2006), SAM-PK-11481 is here referred to Langbergia based on the absence of a maxillary lateral platform, the curvature of the ventral margin of the dentary, and the presence of a high central cusp separated from labial cusp by a narrow valley and strongly protruding cingular cuspules. Future CT-scanning and reconstruction of the occluded dentition of this specimen will corroborate referral to this taxon.

Sidor & Hopson (2018) list the following apomorphic characters in the dentition of Trirachodon berryi: (i) the upper gomphodont postcanines are slightly less than twice as wide as long (i.e., CBR <2) and are transversely symmetrical in apical view; (ii) the mesial and distal cingula are usually complete and typically bear small cingular cuspules; (iii) the sectorial upper postcanines are absent; and (iv) the last upper gomphodont postcanine is transversely expanded, and about two-thirds the transverse diameter of the preceding tooth, with reduced labial and lingual cusps. Sidor & Hopson (2018) also suggest that the dentition of “T. kannemeyeri” and Cricodon are closer than that of Trirachodon berryi, so that these authors propose the new combination Cricodon kannemeyeri. According to Sidor & Hopson (2018), shared synapomorphies between Cricodon metabolus and “Cricodon kannemeyeri” are: (i) transversely expanded upper and lower gomphodont postcanines with one or two sectorial teeth at the rear of the tooth row; (ii) distal upper gomphodont postcanines more than twice as wide as long (i.e., CBR >2) and transversely asymmetrical in apical view, with the lingual half of the crown usually being mesiodistally longer than the labial half of the crown; (iii) cingular cuspules varying in size, with some being particularly large; (iv) mesial and, less frequently, distal cingula often incomplete, and sometimes even absent; (v) distalmost one or two postcanines sectorial and formed by three main cusps oriented mesiodistally and often with a narrow lingual cingulum. We, however, note that a single character (i.e., the presence of one or two sectorial postcanines; character 59) in Sidor & Hopson’s (2018) datamatrix unites Cricodon metabolus and “Cricodon kannemeyeri” and differentiate them from Trirachodon berryi.

Although Sidor & Hopson (2018) note that the mesial and distal cingula are usually complete and bear small cingular cuspules in T. berryi, the distal cingulum is complete and includes a large number of cingular cuspules in NHMUK PV R3307 and SAM-PK-K4801. Likewise, important variations in size in the mesial and distal cingular cuspules, which are as large as other Trirachodon specimens, can be noticed in BSP 1934 VIII 21. Seeley (1894) and Sidor & Hopson (2018) also observe difference in the elongation of the upper gomphodont postcanines between these species, yet the CBR of the penultimate upper gomphodont postcanine of BSP 1934 VIII 21 is 2.3 whereas the crown of SAM-PK-K171 from the same position has a CBR of 2.

As stated before, we agree with Sidor & Hopson (2018) that two morphotypes can be observed among the specimens referred to Trirachodon berryi and “T. kannemeyeri” but we interpret them as ontogenetic variations. “C. kannemeyeri” sensu Sidor & Hopson (2018) can be considered as an younger ontogenetic stage of T. berryi. In adults of this sequence: (i) the last sectorial postcanines were replaced by transversely symmetrical gomphodont postcanines; and (ii) two to three upper gomphodont postcanines, with the last one bearing a large central cusp, and reduced labial cusp and lingual cusps (Fig. 9O), were added in the distal portion of the tooth row. Such dental replacement and increase in the number of gomphodont teeth throughout ontogeny are common in gomphodont cynodonts and were observed in the closely related trirachodontid Cricodon metabolus (Crompton, 1955) as well as the basal traversodontids Scalenodon angustifrons (Crompton, 1955) and Andescynodon mendozensis (Goñi & Goin, 1988).

Given that both morphotypes of T. berryi and “C. kannemeyeri” are represented by large-sized specimens with a skull of more than 10 cm (see Abdala, Neveling & Welman (2006), Table 4; C. Hendrickx, 2018, personal observation), this suggests that the largest specimens referred to “C. kannemeyeri” were subadult/non-fully grown individuals in which: (i) the maximum size was reached; and (ii) the sectorial postcanines remained to be replaced by gomphodont postcanines latter in their life. Specimens classified as T. berryi would consequently represent fully grown, most likely old, individuals in which all sectorial postcanines were replaced by gomphodont postcanines. This hypothesis is supported by size and wear in the postcanines. All specimens belonging to T. berryi have skulls of more than nine centimeters (see Abdala, Neveling & Welman (2006), Table 4) and show extensive wear in the upper and lower gomphodont postcanines (e.g., NHM R3579, BSP 1934 VIII 21–23; SAM-PK-K7888; Sidor & Hopson, 2018, Fig. 10C). On the other hand, wear is barely present or less extensive on the upper and lower gomphodont postcanines of specimens belonging to “C. kannemeyeri” (e.g., SAM-PK-K171, NHMUK PV R3307), which includes individuals whose skull length varies from 5.2 to 10 cm (see Abdala, Neveling & Welman (2006), Table 4; C. Hendrickx, 2018, personal observation). An exception to this is BP/1/4658 which bears seven worn out upper and lower postcanines, and one upper and two lower sectorial teeth. As stated before, BP/1/4658 appears to represent a transitional form between the dental morphotypes of T. berryi and “C. kannemeyeri” and likely belongs to a subadult individual. Consequently, all specimens previously referred to the species “C. kannemeyeri” are here interpreted as younger individuals of T. berryi.

Conclusions

The teeth of diademodontids and trirachodontids are morphologically complex and show particularly important positional, ontogenetic, and intraspecific variations. Even the most comprehensive descriptions of the dentition of these gomphodont cynodonts have inadvertently omitted dental features that might provide taxonomic and phylogenetic information. The main differences between the dentition of diademodontids and trirachodontids include: (i) the presence of a at least one transitional postcanine between gomphodont and sectorial teeth in the upper and lower tooth row; (ii) one or several accessory ridges on the mesial and/or distal surfaces of the transverse crest; and (iii) the absence of mesial and distal basins in some upper gomphodont postcanines in diademodontids; and (i) the presence of labial and lingual cusps of the same height in the lower postcanines; (ii) the absence of shearing planes between the outer surface of the main cusp of the lower and the inner surfaces of the main cusps of the upper postcanines in trirachodontids. Based on the morphology of the dentition, Trirachodon berryi and T. kannemeyeri more likely correspond to two morphotypes of the same species, with T. kannemeyeri representing younger individuals of T. berryi. The Chinese cynodonts Sinognathus and Beishanodon are here reassigned to the Gomphodontia based on dental characters, with Sinognathus being referred to Trirachodontidae, whereas Beishanodon is tentatively ascribed to this clade. This study reveals the importance of describing the dentition of gomphodonts comprehensively and providing as much information as possible on the morphological variation of the complete dentition as well as crown ornamentations and surface texture, with the goal of enhancing the identification of isolated gomphodont teeth and clarifying the phylogenetic position of gomphodont taxa, an objective that we wish to tackle in subsequent contributions.

Supplemental Information

Supplemental Information 1 Appendix 1.

Links to each of the 3d-models of non-traversodontid gomphodont postcanines downloadable on MorphoBrowser.

Click here for additional data file.

The cynodont material was examined in several institutions of South Africa, Namibia, Germany, the UK, Argentina, and China and access to the material was possible thanks to Johann Neveling (CGP), Helke Mocke (GSN), Mathew Lowe (UCMZ), Sandra Chapman (NHMUK), Gertrud Rößner and Femke Holwerda (BSPG), Zheng Fang (IVPP), Elize Butler and Jennifer Botha-Brink (NMQR), Zaituna Erasmus (SAM), Bernard Zipfel, and Sifelani Jirah (ESI). We are particularly thankful to Helke Mocke (GSN) for giving us access to unpublished material. We also thank Christian Kammerer (NCSM) and Christian Sidor (University of Washington) for taking for us and sharing photos of the dentition of Diademodon and Cricodon, Christian Sidor for sharing an unpublished manuscript describing new material of Cricodon from Zambia, and Mike Day (NHM) and Johann Neveling (CGP) for providing information on the locality and horizon on some specimens. We would like to express our utmost thanks to José Eduardo Camargo Martínez and Sergey Meleshin for allowing us to use their artwork for the black silhouettes of cynodonts. Our thanks finally go to Agustín Martinelli, Christian Sidor, and Kirstin Brink for their helpful comments and suggestions, which considerably improved the quality of this manuscript. C. H. dedicates this work to O. Mateus.

Institutional Abbreviations

AM Albany Museum, Grahamstown, South Africa

AMNH American Museum of Natural History, New York, New York, USA

BP Evolutionary Studies Institute (formerly Bernard Price Institute for Palaeontological Research), University of the Witwatersrand, Johannesburg, South Africa

BSP Bayerische Staatssammlung für Paläontologie und Geologie, Munich, Germany

CGP Council for Geosciences, Pretoria, South Africa

GSN Geological Survey of Namibia, Windhoek, Namibia

IVPP Institute for Vertebrate Paleontology and Paleoanthropology, Beijing, China

MB Museum für Naturkunde der Humboldt Universität, Berlin, Germany

MHNSR–Pv Museo de Historia Natural de San Rafael, Mendoza, Argentina

NHCC National Heritage Conservation Commission, Lusaka, Zambia

NHMUK PV Natural History Museum, London, UK

NMQR National Museum, Bloemfontein, South Africa

NMT National Museum of Tanzania, Dar es Salaam, Tanzania

PKUP Peking University Paleontological Collections, Beijing

SAM-PK Iziko, the South African Museum, Cape Town, South Africa

UMZC University Museum of Zoology, Cambridge, UK

Additional Information and Declarations

Competing Interests

Author Contributions

Data Availability

The authors declare that they have no competing interests.

Christophe Hendrickx conceived and designed the experiments, performed the experiments, analyzed the data, contributed reagents/materials/analysis tools, prepared figures and/or tables, authored or reviewed drafts of the paper, approved the final draft.

Fernando Abdala conceived and designed the experiments, authored or reviewed drafts of the paper, approved the final draft.

Jonah N. Choiniere conceived and designed the experiments, authored or reviewed drafts of the paper, approved the final draft.

The following information was supplied regarding data availability:

The 20 3D-models of gomphodont postcanines are available at the MorphoBrowser database (links are in Appendix 1).

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
