# Peer review of "A proposed terminology for the dentition of gomphodont cynodonts and dental morphology in Diademodontidae and Trirachodontidae"

_PeerJ, doi:10.7717/peerj.6752_

## Round 0.1 · original submission · Major Revisions

Dear authors,

I apologize for the delay in my decision, but due to the size of the manuscript and delay in getting some of the reviews in a time when the teaching semester started and my commitment to properly compare my assessment with that of the reviewers, I could not make my decision earlier. You comprehensibly describe and revise the terminology in gomphodont cynodonts. These observations are new and have important implications. It therefore would like to see it published. However, the manuscript is currently difficult to follow in some parts and several crucial points need to be addressed before publication:

Justification for new terminology of dental anatomy: you focus on the description of dental morphology of non-traversodontid gomphodontians, but also introduce a new terminology for dentition in gomphodont cynodonts. However, you need to state more clearly why a new terminology is necessary. Furthermore, as you don´t analyse the newly described characters cladistically/phylogenetically, it a bit difficult to infer autapomorphies confidently.

more adequate to change the title and rewrite the article I have no issue with using previously published phylogenetic hypotheses and mapping dental morphology on them, but if you did this it should at least be more clearly stated – suspect autapomorphies could in such a context only be suggested. A last important point to make this new terminology useful would be that terms used in older literature are also mentioned (see comment by reviewer 1). In this context, it might also be appropriate to modify the title to: Dental morphology in in Diademodontidae and Trirachodontidae: implications for the terminology of the dentition of gomphodont cynodonts or something along those lines.

Phylogenetic analysis: If you identified autapomorphies in the course of this work, a more complete systematic palaeontology section would be necessary. Furthermore, a phylogenetic analysis would also be recommended (see comments by reviewers 1 and 3) unless you just want to map dental morphology on previously suggested phylogenies.

Description of Dental Morphology : The assignment of specimens to particular species should be better justified. The manuscript is already quite long, but a proper synonymy list and diagnosis of taxa or at least a statement which diagnosis you follow is necessary (see particularly comments of reviewer 1)

Terminology: your manuscript is quite heavy on terminology as it is part of its aim to revise and homogenize terminology for disparate teeth in gomphodonts . Most of these are properly defined under proposed dental terminology section, although it think the manuscript would benefit from placing these parameters in a table rather than as plain text. It seems that parameters are written in capitals and normal abbreviations aren´t, but it would help to state this more clearly. Most importantly, you use terms (labiolingually expanding, cervix, ) without defining them in detail when they are used the first time (sometimes already in abstract) which hard to follow for non-experts. I have not issue with using such terms, but it would be good to define them as soon as they are used and restrict such terms in the abstract (or at least mention a synonym or description for them) if these are of key importance, so everybody is able to follow your manuscript. Reviewers 1 and 2 lso pointed out that some of the used term are used in the wrong context or imprecise way (denticle versus serration: see comment by reviewer 2)


Extra figure or table: you properly define the abbreviation in the “proposed dental terminology” but it remains difficult to track and compare between different species. I feel it would help to add a schematically figure showing and comparing the dental formulae of all taxa and the interpretations of the relationship and changes through ontogeny. In my opinion, this would make the manuscript easier to follow particularly for non-cynodont experts and aid your goal in providing an easy to follow terminology for dental morphology. For this purpose I suggest to schematically represent all teeth for each taxon in a row – also those that are potentially missing and their potential “homology” or relationship with other teeth. You would have a much rows as taxa and as much teeth as are represented or could be represent in particular taxa. Simple schematic presentations of the teeth in top or side view would both work. You could potentially also use topographic colour profiles (see above; such an example can be found in Plyusnin I, Evans AR, Karme A, Gionis A, Jernvall J (2008) Automated 3D Phenotype Analysis Using Data Mining. PLOS ONE 3(3): e1742.https://doi.org/10.1371/journal.pone.0001742). A table comparing all formulae which potentially also list CBR would also potentially work.

Parameters: It would be good to place numerical values like CBR in a table, which would make it easier to follow. Potentially also combine it with the figure showing the formulae. A more common structure more in line with systematic paleontology sections in general would be useful (see also comments by reviewer 1)

Three-dimensional data: please make sure that the photogrammetry and tomographic data are available for everyone (not just reviewers) during final publication of the paper. You also refer to CT-scan data from Trirachodon kannemeyeri. It is however necessary to discuss the scanning specifications as well as it would be appropriate for scientific reproducibility (see Davies et al. 2017 for standards in the field). The same would also be useful for the photogrammetry. I understand your concern with potentially making raw data available where you still want to work on, but most of these platforms also allow putting an embargo on data until publication or longer. Furthermore, for your purposes you would only need to make available the relevant data on the dentitition. Although not a necessity – topographic colour profiles might also help to better represent the 3D structure of the teeth. I have seen submission doing this with Paraview, but there are probably many other options to produce such profiles using 3D models. Davies, T. G., Rahman, I. A., Lautenschlager, S., Cunningham, J. A., Asher, R. J., Barrett, P. M., ... & Braga, J. (2017). Open data and digital morphology. Proc. R. Soc. B, 284(1852), 20170194.


Please address all comments made by the reviewers and myself (also annotated pdfs) in addition to the most crucial points summarized above.


Line 20: I guess you mean “might represent” instead of “represents” as you did not assesses this quantitatively or cladistically.

Line 26: I guess it should be made clear on what the recognition of autapomorphic dental feature is based on as you did not do any new analyses. I feel it would be interesting to see the results of novel analysis. However, it is also works to explore dental features in the context of previous phylogenies as you do here, but you would at least need to state this approach more clearly from the beginning.

Line 36: I have nothing against the use of Oceania. However, as it is written in your manuscript you can get the impression that Oceania is a continent, although it rather refers to a region contains parts of at least 3 continents including Australia, Zealandia and Eurasia. It might be better to spell out which continents you are particularly referring too or find an alternative way to write what you want to say using Oceania.

Line 53: It is ok to assign personal observations to particular co-authors, but please also do it in the same; if you want to abbreviate please do it always (C.H.; F.A.; etc.).

Line 55: it might be a tricky here to say that it represent the primitive condition if there is debate about it or no extensive morphological analyses have been performed. So maybe consider to write “likely represents” or “is considered” as you are relying on previous work.

Line 72; Do you mean resistant compared with other parts of the skeleton? Please specify to what type of resistance you are referring.

Line 95; I agree with placing the institutional abbreviations after the introduction as you already start using them in the next paragraph.

Line 126: It would be appropriate to provide the scanning specifications – potentially with a new subheading for clarity and making the raw data available for reproducibility

Line 139-140; it should be “Sidor & Hopson (2018)”; please make sure that all references are properly formatted

Line 148: I feel it would be more appropriate to place “Proposed dental terminology” in a table – like a terminology list; two tables/subheadings would be appropriate – one for “quantitative” parameters (CBL to DSDI) and one dental terminology abbreviations (ac to tun).

Line 389: you properly define the abbreviation in the “proposed dental terminology” but it remains difficult to track and compare between different species. I feel it would help to add a schematically figure showing and comparing the dental formulae of all taxa and the interpretations of the relationship and changes through ontogeny. In my opinion, this would make the manuscript easier to follow particularly for non-cynodont experts and aid your goal in providing an easy to follow terminology for dental morphology. For this purpose I suggest to schematically represent all teeth for each taxon in a row – also those that are potentially missing and their potential “homology” or relationship with other teeth. You would have a much rows as taxa and as much teeth as are represented or could be represent in particular taxa. Simple schematic presentations of the teeth in top or side view would both work. You could potentially also use topographic colour profiles (see above; such an example can be found in Plyusnin I, Evans AR, Karme A, Gionis A, Jernvall J (2008) Automated 3D Phenotype Analysis Using Data Mining. PLOS ONE 3(3): e1742.https://doi.org/10.1371/journal.pone.0001742)

Line 398: please define salinon-shaped

Line 401: I guess you mean to say that denticles are minute – not minutes

Line 1347: I would feel placing the information contained in lines 1348 to 1388 would also make it easier to follow.

Line 1441-1443: These couple of lines are really odd. You mention the CT-scanning in the methods, but do not properly describe the scanning details or procedure. Also here you kind of say that you could scan or maybe even did, but don´t discuss the full details. You need to decide what you want to do – or you describe at the least the information for dentition from this scan in detail or you don´t mention it at all. You cannot have both – at least not as it is currently written. If you want to scan it in the feature – just say that you suspect that when scanning could corroborate it. As you do refer to information from this scan – there is no way around providing at least the basic information (scan settings, raw data as tiff files or reconstruction of dentition as stl files).

Figures 3-10: The depiction of 3D-reconstructions in grayscale is ok, but I feel it would make it easier to understand and get your points across if you would provide actual 3D images or topographic colour profile. It is a bit of a shame you have all this 3-dimensional data, but it is not really used or presented in the fullest.

Figure 11: you plot these values through ontogeny, but it might help to show depiction/sketch of at one or two specimens next to it to fully convey what this shows. How is skull size defined? Does it refer to the length?

·

Basic reporting

Overall Review
As the titles states, this paper has two main goals regarding non-traversodontid gomphodontians: provide a revised terminology for dental anatomy and provide new information about the dental anatomy itself. Below I raise five major comments as well as provide a number of more minor edits. Overall, these suggestions amount to fairly significant modifications that will necessitate a second round of reviews. (Christian Sidor)

Comment 1: The dental morphology of two families of non-traversodontid gomphodontians is the primary topic of the manuscript and the authors should be commended for the amount of information they provide. However, the authors do not provide evidence that a revised set of terms describing gomphodont dental anatomy is needed, despite terminology being front-and-center in the title. The introduction provides ample background on the families discussed, but that is very different from providing background on what has been problematic with the anatomical terms previously used, or what confusion exists with the current descriptions of the dental anatomy concerned. Why is a new set of terms needed? The introduction should be rewritten to focus on the main thrust of the paper and to explain to the reader why this review/revision is needed. Wilson (1999) is a good example of a paper introducing terminology with ample documentation of why it was needed. Personally, I do not feel that a revised terminology is needed, but perhaps I could be convinced if the authors provide context for this component of the paper.

Comment 2: The lack of a formal Systematic Paleontology section is problematic, especially given that this is meant to be a comprehensive review of two families. Although the authors provide most of what would be contained in such a section within the Results section, several important issues remain:
A) The authors do not seem to have a grasp of the use of taxonomic authorities. For example, parentheses-e.g., (Seeley, 1894)-have a very specific connotation within systematic paleontology (e.g., the species was previously assigned to a different genus) and thus they should not be used unless warranted. In other words, using parentheses in the context of systematic paleontology is not the same as referring to an author in a citation. The authors should review the relevant ICZN guidelines and confirm or delete parentheses as appropriate throughout.
B) More importantly, definitions and diagnoses are not provided for most taxa, and without this information it is not clear how the specimens are assigned. For example, what diagnostic features were used to assign specimens to Cricodon metabolus? Likewise, how will a future researcher assign a new fossil to Diademodontidae without an explicit phylogenetic definition or diagnosis? By contrast, the amount of detail provided in the Occurrence and Horizon subheadings seems unnecessary for a paper discussing anatomy (not geography or stratigraphy). The occurrence data is especially not useful since it doesn't link a specimen number to a locality. A list of apomorphies is provided starting at L1348, but identifying apomorphies can only be accomplished in the context of a phylogenetic analysis. If the autapomorphies for each species have been recognized in the course of this research, the authors are well on their way to being able to provide a complete systematic paleontology section in the revision.

Comment 3: This relates to the issues raised in Comment 2, but highlights the effort that should be expected from a significant review-type paper such as this. Sidor and Hopson (2018) offered the most recent treatment of trirachodontid dental anatomy and systematics. They provided a phylogenetic definition of the family, diagnoses for each of the four species, and a cladistic analysis placing trirachodontids among a wide range of cynodonts. Their results show that 'Chinese trirachodontids' are better understood as traversodontids (not trirachodontids). The current authors, who had access to the Sidor and Hopson (2018) manuscript since late 2016, should to either 1) adopt the Sidor and Hopson (2018) results, 2) explain in more detail why they believe it is wrong (including the data needed in Comment 2), or-best of all-3) provide an analysis showing why their taxonomic organization is preferable. Either this is a descriptive paper, in which case systematic conclusions are not warranted, or this is a systematic paper, in which case an analysis needs to be presented.

A related issue is the assignment of specimens to species. This is critical and can't be evaluated without a cladistic analysis and more complete systematic paleontology section. For example, SAM-PK-K5881 is used to describe the dentition of Cricodon metabolus, but it is unclear why this specimen pertains to C. metabolus in the first place. It doesn't seem to fit stratigraphically and the morphology doesn't correspond to Sidor and Hopson's (2018) diagnosis.

The anatomical data on diademodontids (with a definition and diagnosis included) is uncontroversial and worth its own paper. The trirachodontid section needs substantially more work.

Comment 4: The term "apex" has a long history of use in the dental literature (see Fehrenbach and Popowics, 2016 and other textbooks) and corresponds to the opposite direction from how the authors use it here (see also Hendrickx et al. 2015). For a paper attempting to clarify and regularize terminology, this is troubling to say the least. This issue should be addressed in the introduction and a convincing argument made to separate the human dental terms from those used for cynodonts. Wilson (2006) discusses similar issues and I believe Larry Witmer has a paper talking about applying bird terms to dinosaurs (but I can't find the reference).

Comment 5: The dental terms discussed from Lines 172-359 lack references to previous terms used in the literature. In fact, there are several seminal works (e.g., Crompton, 1972) that should be noted and comparisons made (either for individual terms or at the beginning of the section). In addition, the abbreviations provided do not specify uppers or lowers, which previous abbreviations did. All of this is related to Comment 1, which is a request to explain the history of these terms and what confusion needs to be clarified.



Minor edits
L1 - Reverse order of title so anatomy comes first and terminology second? Just an idea, but anatomy forms the bulk of the paper and, as I noted above, I'm not particularly sold on the need for a revised terminology.
L16 - Delete "are"
L23 - Replace "taxon" with "species". Taxon is too vague.
L26 - Replace "dentition morphology" with "dental morphology". This type of grammatical mistake occurs repeatedly should not occur when there is a native English speaker among the coauthors.
L30 - Replace "A" with "Finally, we propose"
L44 - Replace "the" with "their"
L61 - Replace "carnassial" with "sectorial". Carnassial has very specific taxonomic and functional connotations that are not appropriate for a cynodont.
L62 - Replace "taxon" with "species". Taxon is too vague.
L62 - Trirachodontids were recently reviewed in depth by Sidor and Hopson (2018).
L72 - Wynd et al. (2018) provide a recent example of the difficulty in assigning non-cranial material to Triassic cynodonts.
L79 - Replace "most" with "the descriptions of many". Citing "CH pers ob." is unnecessary.
L82 - Traversodontids should be mentioned, at least briefly, somewhere in the introduction. It might also be worth noting the controversial placement of tritylodontids among gomphodontians by several authors (e.g., Hopson and Kitching, 2001; others by Sues).
L83 - "Aims to" should be deleted and the rest of the sentence updated to a more active tense (i.e., "i) proposes a standardized…")
L92 - Update to "data matrix"
L96 - Delete "City"
L130 - Add citation to Sidor and Hopson (2018) at the end of the first sentence, as they previously came to the same conclusion.
L133 - See Comment 4 above regarding use of "apex" and "apical" and any other derivatives (e.g., apicobasal).
L139 - Fix parentheses around citations.
L213 - Why is there commentary on the distribution of this feature, but not others?
L221 - This definition seems to be at odds with Brink and Reisz's (2014) distinction between serrations and denticles and should be resolved.
L251 - "Oval, quadrangular…" are not useful descriptors unless the reader knows what view you are talking about.
L256 - Is the proposed function ("specialized for cutting") necessary?
L257 - Why is a traversodontid feature included here?
L305 - Why is this called longitudinal if apicobasal is your preferred term for this axis?
L309 - Replace "their" with "its"
L311 - Change to "Two or more…"
L327 - This is the only non-dental term on the list. Needed?
L334 - Replace "Blade-like" with "Labiolingually compressed" because the former does not specify a direction to the blade.
L351 - "Intermediated" is not a word.
L357 - This description is unclear. In this case, transverse seems to refer to transverse to the long axis of the tooth, but this is not specified.
L362 - Remove parentheses.
L362 - Add definition subheading and provide phylogenetic definition.
L363 - Check systematics to see if parentheses are necessary or not.
L379 - Add diagnosis subheading and add cladistic diagnosis.
L385 - Diademodon is likely also from the upper Fremouw Formation of Antarctica.
L394 - Replace "dentition morphology" with "dental morphology".
L397 - Are there no differences between upper and lower incisors? If not, this should be stated explicitly.
L401 - Delete "s" from minutes.
L409 - The SAM specimen number is incomplete.
L415 - "well-above" is not clear, since the distinction between upper and lower canine was not made. This should be clarified.
L421 - The BPI specimen number should be BP/1/4669 (I believe).
L428 - Not clear what specimen of Diademodon is being considered in the second half of this sentence.
L429 - Replace "while" with "whereas"
L430 - Clarify why denticles and serrations are both used in this sentence.
L436 - Delete "specimen". (Holotype is a noun, so either use holotypic specimen or holotype)
L439 - Insert "proportionately" before "wider"
L447 - Not clear what "at base of the cusp" refers to. Closest to cusp? Please rewrite for clarity.
L452 - Delete "d" from located.
L485 - The NHMUK specimen number format is incomplete. Use NHMUK PV R3303 (and numerous places elsewhere in the paper)
L492 - The SAM specimen number is incomplete.
L507 - Would not "denticulated" be preferable to "serrated" here?
L525 - Replace "is" with "are"
L553 - Delete "distally" because it is repetitious. Recurved denotes distal (otherwise it would be procurved).
L564 - The line about enamel texture is out of place here. It should go at the beginning of the section.
L571 - Remove parentheses.
L573 - Add diagnosis subheading and cladistic diagnosis.
L585 - Delete "the"
L595 - Add "upper" before "postcanines"
L602 - Not clear if there are one or more transitional postcanines because of lack of agreement between "a" and "postcanines"
L607 - Remove parentheses
L608 - Add definition subheading and provide phylogenetic definition.
L608 - Add diagnosis subheading and add cladistic diagnosis.
L608 - Remove parentheses
L612 - Add diagnosis subheading and add cladistic diagnosis.
L657 - Replace "cross-section" with "cross-sectional" here and elsewhere (e.g., L665) as you are using the term as an adjective.
L671 - The sentence beginning with "It is formed…" is very confusing as written. I can't tell if you are talking about serial homology or to some other taxon. Please rewrite for clarity.
L718 - Delete "distally" because it is repetitious.
L736 - Remove parentheses
L740 - Add diagnosis subheading and cladistic diagnosis.
L744 - Replace "Formation" with "Beds". See Smith et al. (2018).
L750 -Sidor and Hopson (2018) also described additional information about C. metabolus dental structure based on referred material.
L772 - Sidor and Hopson (2018) report denticle counts for NHCC LB28, a juvenile Cricodon metabolus. 6-8 serrtions/ 3mm in lower incisors.
L776 - Rewrite to "much higher density of denticles …"
L792 - Delete "distally" because it is repetitious.
L803 - Sidor and Hopson (2018) report denticle counts for NHCC LB28, a juvenile Cricodon metabolus. 15-19 serrations/ 3mm in mesially and 9/10 serrations/ 3mm in distally. They consider this disparity to be a diagnostic feature of C. metabolus.
L813 - Change to "cross-sectional"
L814 - Salinon is not a useful descriptive term. I had no idea what it meant and a quick web search did not help.
L820 - Sidor and Hopson (2018) describe the occurrence of a sectorial first upper postcanine in a juvenile C. metabolus, suggesting that mention of ontogenetic changes in dental shape is warranted.
L835 - The step-like position of sequential postcanine teeth could be a feature of tooth replacement. See Sidor and Hopson (2018:fig. 3).
L882 - A brief explanation of why K5881a and K5881b are necessary would be useful. And at L893, the a/b is missing - why?
L905 - The sentence beginning with "If" doesn't make sense.
L959 - Sidor and Hopson (2018) show a lingual cingulum on the distalmost lower postcanine, which contradicts what is written.
L962 - Delete "specimen"
L963 - Rewrite to past tense - "That study revealed that …"
L970 - Check Sidor and Hopson (2018) for parentheses
L971 - Update NHMUK abbreviation to include PV as necessary throughout the paper.
L976 - Sidor and Hopson (2018) show that 12168 is the original specimen number and informed the SAM about this. Check with Zaituna.
L991 - Cite references for "other authors"
L1003 - Replace "counts" with "features"
L1034 - Salinon?
L1054 - The 'shouldering' described is not really the same as what you see in traversodontids.
L1057 - The sentence beginning with "If" doesn't make sense.
L1116 - Change to "postcanine"
L1169 - "two and three fifth" does not make sense.
L1170 - Delete "distally" because it is repetitious.
L1176 - Not clear how a projection can be "high and short"
L1126 - Delete "the taxon"
L1239 - Replace "finally" with "finely"
L1247 - Address parentheses issue noted in Comment 2.
L1312 - Address parentheses issue noted in Comment 2.
L1344 - Change to "postcanines"
L1346 - Apomorphies can only be recognized with after phylogenetic analysis, so I'm unclear how the authors have determined this.
L1358 - This character description is unclear.
L1360 - Sidor and Hopson (2018) provide a more robust diagnosis for the family.
L1389 - Sidor and Hopson (2018) addressed the affinities of Beishanodon with a complete analysis of craniodental characters (not just dental).
L1411 - Change citation of Sidor and Hopson (2018) to Hopson and Sidor (2015). The results presented in the 2018 analysis showed the Chinese species as basal traversodontids.
L1421 - Cite Sidor and Hopson (2018), as their analysis showed the Chinese species to be basal traversodontids.
L1429 - See comment 3.
L1444 - Repeating the characters listed by Sidor and Hopson (2018) isn't very useful. This space would be better used to point out issues of homology among the character states, ontogenetic changes in the dentition, or inaccurate character codings.
L1473 - The observed differences between the two taxa can be explained as the result of there being two morphospecies (viz. Sidor and Hopson, 2018) or one species with ontogenetic variation. If the authors truly believe the latter, then some ontogenetic analysis needs to be presented.
L1495 - Why is "K171" used twice?
L1505 - Dental complexity includes species variation, ontogenetic variation, and variation along the tooth row. It might be worth calling out all three at some point.


Literature cited in this review
Brink, K. S., and R. R. Reisz. 2014. Hidden dental diversity in the oldest terrestrial apex predator Dimetrodon. Nature Communications 5:3269 doi: 10.1038/ncoms4269.
Crompton, A. W. 1972. Postcanine occlusion in cynodonts and tritylodontids. Bulletin of the British Museum (Natural History) Geology 21(2):30-71.
Fehrenbach, M. J. and T. Popowics. 2016. Illustrated dental embryology, histology, and anatomy, 4th ed. Elsevier.
Hopson, J. A., and J. W. Kitching. 2001. A probainognathian cynodont from South Africa and the phylogeny of nonmammalian cynodonts. Bulletin of the Museum of Comparative Zoology 156(1):3-35.
Sidor, C. A., and J. A. Hopson. 2018. Cricodon metabolus (Cynodontia: Gomphodontia) from the Triassic Ntawere Formation of northeastern Zambia: patterns of tooth replacement and a systematic review of Trirachodontidae. Pp. 39-64. In C. A. Sidor, and S. J. Nesbitt, eds. Vertebrate and Climatic Evolution in the Triassic Rift Basins of Tanzania and Zambia, Society of Vertebrate Paleontology Memoir 17. Journal of Vertebrate Paleontology 37 (6, supplement).
Wilson, J. A. 1999. A nomenclature for vertebral laminae in sauropods and other saurischian dinosaurs. Journal of Vertebrate Paleontology 19(4):639-653.
Wilson, J. A. 2006. Anatomical nomenclature of fossil vertebrates: standardized terms of 'Lingua Franca'? Journal of Vertebrate Paleontology 26:511-518.

Experimental design

No comment

Validity of the findings

See comments in Basic Reporting.

Additional comments

No comment

·

Basic reporting

There were a few inaccuracies with the English, which are addressed in the attached PDF. The references, article structure, figures, and data are good.

Experimental design

This paper does a very good, thorough job of describing the teeth of gomphodont cyndodonts, and provides many new characters that will be useful for future taxonomic identifications and phylogenetic studies. The identification of a ontogenetic series in Triarchodon is particularly interesting.

Validity of the findings

The data are robust and the conclusions are well stated.

I only have one main concern, which is the lack of differentiation between a serration and a denticle.
As described (Lines 338–346), a serration can be composed of enamel or enamel and dentine. This is very confusing throughout the manuscript when ‘serrations’ are being described, as it is unknown if true denticles (dentine core with an enamel cap) are present or not. It is very important that these differences are noted, since an enamel serration and a denticle are developmentally very different, and a lack of distinction between the two features can hamper future phylogenetic studies and interpretations of the evolution of these characters. It is especially confusing in an example like Figure 2J where serration and denticle are labeled as the same thing.
Based on developmental and genetic data, the development of a denticle, cusp, or cuspule is a very different process than the development of an enamel feature. Also, there are developmental differences between reptiles and mammals (See: Polly, P.D. 2015. Gene networks, occlusal clocks, and functional patches: new understanding of pattern and process in the evolution of the dentition. Odontology 103(2): 117-125. Look at references referring to development of tooth shape (12, 16–20) for more information on how this happens). As gomphodonts are closely related to mammals, it is possible their tooth development is more similar to a mammal than a reptile, but this is still unknown. Therefore, the careful documentation of when a denticle, cusp, or cuspule appears vs. an enamel feature is very important for deciphering the evolution of this dentition in your future phylogenetic analyses (as stated in the last sentence of the conclusions!), especially when considering the homologies of these features and differences between incisors, canines, and postcanines (is it only a denticle if it is on a carina? What is the difference between a serration and a series of cuspules? (lines 343-346) Is a cuspule developmentally similar to a denticle?). Additionally, the presence of true denticle or an enamel serration could impact your interpretations of diet, since a denticle is likely to withstand wear for a longer time than an enamel feature.

Here are some examples that should be clarified, whether by changing the definition of a serration or by using denticle instead of serration. The entire manuscript should be re-examined to ensure that the terms denticle and serration are being used correctly. The term ‘denticulated’ should be used when denticles are present.
Lines 430-431: “The distal denticles are large and apically inclined whereas the mesial serrations are low and show a widely convex external margin.”

I read this as the distal carina has true denticles, while the mesial carina may or may not have true denticles. If the word serration is replaced by denticles, the description would be much more clear.
Lines 628-632: “The largest denticles are at mid-crown whereas the basalmost and apicalmost serrations do not reach the root and the crown apex (Figure 6A, B). The distal denticles of the fourth upper incisor of NMQR 3281 and SAM-PK-K11481 share the same morphology, with the largest denticles at mid-crown and the basalmost and apicalmost serrations extending far above the root (Figure 6C) and below the crown apex, 
respectively.”
Using denticles would clarify that there are differences in the occurrence of true denticles along the mesial and distal carinae.
Lines 856-860: “Unlike the central cusp, the denticles of the mesial, distal 
and central ridges of the labial and lingual cusps diminish in size apically. Similar to the serrations of incisors and canines, they are apicobasally elongated and change sporadically in 
size. Their external margin is, however, flattened or weakly convex (Figure 7O). The serrations on the labiomesial and labiodistal ridges either extend along the whole cusp up to its apex, or are 
restricted to the basal half of the cusp.”
Once again, it is unknown if the serrations on the labiomesial and labiodistal ridges are true denticles or not.
Lines 949-951: “The mesial and distal carinae are serrated along the apical portion of the main cusp and the serrations extend across the apex of the cusp (Figure 7P-Q). The denticles are low, apicobasally elongated and show a symmetrically convex external margin.” 

Lines 1006-1009: “The distal denticles of the first and 
second incisors of SAM-PK-K5821 are incipiently developed and form poorly delimited convexities changing sporadically in size along the carina (Figure 8C). These serrations appear to extend well-above the cervix and get flared at mid-crown.” 

Lines 1259-1261: “Minute denticles can be seen at the 
base and apex of the distal carina of the fourth left upper incisor (Figure 10B, C). We counted 
eight denticles per mm on this tooth. No other incisor appears to bear serrations.” In this example, are the minute serrations true denticles?

·

Basic reporting

It is a good contribution to improve the knowledge of gomphodont dentition, including detailed description all non-traversodontid gomphodonts. Overall, the manuscript is well-structured and its scientific content is satisfactory. The amount of additional data is amazing and will be useful for future contributions. The authors stated in the text that this contribution is part of a series of (at least) three papers. Because of this, some parts of the ms seem to be partially addressed what turn difficult, in my view, some inferences, as for example, homologies among postcanine structures. In the attache file I include several comments that should be considered by the authors prior to publication. I think that after a moderate review the MS should be accepted for publication.

Experimental design

The description are well-performed with well-figured specimens. I think that the figures can be improved by adding a picture/drawing of the whole dentition, as for example, the skull in ventral view and lower jaws (or jaw) in dorsal view to see all the dental elements in position (of course, in the taxa that it is possible). I think that this will greatly improve the provided information. Also, those diagrams will be important to observe changes along a dental series and also to interpret possible homologies among dental structures. Regarding dental homologies, it is the most weak issue in the MS.

Validity of the findings

The authors made an excellent job describing the dentition of all non-traversodontid gomphodonts, with detailed description of not only postcanine teeth, but also incisors and canines. It will be a base for future workers on the subjects. However, i think that an standardized terminology is still difficult to propose if several traversodontids and other basal cynodonts are not considered in the analysis. And not only topology of structures but also tooth replacement, upper/lower tooth occlusion relationships, detailed changes along the dental series and comparison along the different clades should be used for an standardized terminology among cynodonts, in which teeth are extremely important to taxonomy and phylogeny.

Additional comments

The authors can find my comments and suggestions on the annotated manuscript.

---

## Round 0.2 · Minor Revisions

My apologies for getting back to you so late, but as I only recently received the last review and it raised some additional issues i wanted to verify myself. I agree with both reviewers that the latest revisions have made the manuscript easier to follow and that the manuscript is generally in a good state. Your manuscript is as good as accepted as far as i am concerned. There are just some minor additional issues (mainly of a formatting nature) I would like to address before publication.

The main points are:

Capitalization: Be careful with the use of capitals; Upper/Lower or Late/Early should not be written with capitals in several (“inofficial”) stratigraphic terms. For example: Upper Lifua Member or Early or Late Anisian on lines 691-694 (see also comment by reviewer 2; please consistently review this throughout manuscript).

Occurrence/Horizon: I agree with the authors that this information can be relevant to understand which material was investigated, but I concur with Reviewer 2 that it would be easier to reorganize this section or in one your table to include locality data after each referred specimen. It is difficult to follow for someone not familiar with this group (we don’t know which specimen derives from where) and it might also be easier for people familiar with this group to have a list which specimens from these localities you investigated.

In addition to other points raised by the reviewers, please also address the following points:

Line 29: In abstract and in text you use “confidently referred” for Trirachodontidae and Gomphodontia. Note sure if the word confidently is that correct in this sense as you mention throughout the manuscript that further cladistics analysis is necessary. I feel it would be better to drop the word confidently and/or at least add “(seem) to belong to Trirachodontidae and Gomphodontia based on teeth characters, respectively”

Line 106-109: you allude to two upcoming contributions. Personally, I have no issue with mentioning this. However, it might be appropriate to mention them more precisely (citation, see reviewer 2) or just mention that these issues fall outside the scope of this contribution. Personally, I think it is ok to discuss these matters in another contribution as this one already provides sufficient new information and would even get more lengthy when including additional aspects.

Line 227: it is worth considering to start a new line or another visual measure to separate the description of the term from the ones synonymized with them.

Line 229: “cross-sectional outlinerecurved” = this seems out of place (delete?)

Line 691: “middle to upper Lifua Member” ? I did not verify this throughout the manuscript as it am not familiar with all these formations; please check throughout manuscript similar issues with middle and upper

Line 693: I guess you mean rather “early or late Anisian” Otherwise you need to state if these are official defined.

Line 880: space between cingulum (Figure 7N). This is the only one I spotted, but please check entire manuscript as I might have missed other cases.

Line 1332-1334: Another example of “upper” and “late”

Line 1436: “most complex among non-mammaliaform cynodonts”: please add reference for this statement or rephrase if this is an outcome of your study.

Line 1474-1475: “probainognathian”: are you sure about this statement (see comment by reviewer 2)

Line 1592-1594: where they previously/originally assigned to Gomphodontia; then it might be more appropriate to state “remain” or “reassigned to … based on teeth characters”.

Figure 1: it is nice you provide the traditional hypothesis or Ottone´s hypothesis, but it might be appropriate to explain what the difference is between them slightly more extensively in text or caption, particularly for readers not familiar with it. Please take this opportunity to check on more time for potential typos and other formatting issues I might have missed. In some cases, it is hard for me to assess as the manuscript is quite rich in terminology I am not that familiar with.

·

Basic reporting

No comment

Experimental design

No comment

Validity of the findings

No comment

Additional comments

Overall Review
I was asked to review the revised manuscript of Hendrickx, Abdala, and Choiniere for PeerJ. I have read the revised MS (85 pp) and feel that it is a substantial improvement over the initial submission. Although I disagree with some points raised in the rebuttal letter provided by the authors, it is overall fit for publication once the other papers that are part of this series can be assured of publication (i.e., accepted or in press).

The stratigraphic chart in Figure 1 is a welcome addition.

Rebuttal Comment: “as for the term apex, the first reviewer does not seem to be aware of the correct definition of this term” and “It is troubling, to say the least, that this reviewer is unaware of the actual definition of the term apex, which was clearly defined by Hendrickx et al. (2015) in their proposed terminology on the dentition of theropod dinosaurs, a terminology we followed in this MS.”

Here we are going tit-for-tat about this issue. I find it laughable that the rationale that the authors provide is a 2015 paper that they wrote. At minimum, the authors should acknowledge the typical dental textbook use of these terms somewhere early on in the MS.

Comment 2: I agree with the editor’s suggestion that a simple schematic representation of tooth variation across the taxa represented would be a useful synthesis.

Rebuttal Comment: “We believe that data on the occurrence and horizon provided for each taxon is very helpful and we do not intend to remove them.”

As with the first previously submitted version of this manuscript, I disagree with authors that the inclusion of occurrence (=locality) data is necessary or even worthwhile. Horizon data is fine. If I were the editor for this journal, I would ask –how is the inclusion of this information useful to the readers? For example, would a researcher ever look for this information in a paper on tooth anatomy? (No.) I also don’t think including this info is useful for the ‘greater good’ either, because as I noted in my original review, without associating the locality with exactly which specimen comes from it, not much is gained. For example, no one could add a new occurrence to the Paleobiology Database from the information given.

Solution: if the authors reorganized this section to include locality data after each referred specimen, then I have no problem including the information.

Minor Comments:
L238 – Not clear why Cyclogomphodon is in parentheses. If AMNH FR 5519 is the type of Cyclogomphodon platyrhinus, just list it as such and delete the “Diademodon (=” part.

L270 – Regarding the stratigraphic uncertainty noted by Ottone et al., an alternative that should be noted by the authors is that Diademodon might have a longer stratigraphic duration than previously expected (as noted by Peecook et al.. 2018)

A note on geological terms: There is no “Upper Ntawere Formation.” In this case, “upper” is used as an adjective, not as part of a proper noun – and so it shouldn’t be capitalized. Similarly, there is no official “Late Anisian” (but of course there is a Late Cretaceous), so you need to check these case-by-case in the manuscript.

L1474 – It would be worth noting up front that Sidor and Hopson’s (2018) cladistic analysis found Beishanodon and Sinognathus to be traversodonts, but that the authors reiterated their previous suggestions that they might represent probainognathians instead. They way the traversodont result is introduced later in the paragraph is a little confusing and seems to contradict what was written earlier.

·

Basic reporting

no comment

Experimental design

no comment

Validity of the findings

no comment

Additional comments

The authors seem to have addressed all reviewer comments appropriately. A final pass through should be done to catch all English errors and type-os, and then the paper will be ready for publication.

---

## Round 0.3 · accepted · Accept

Thank you for making these final changes.

I only have one final remark. You are correct concerning publications in preparation. I apologize for the confusion, but i just wanted to say to change the reference when published (and cite it appropriately) or cite it as unpublished in text (without listing it in reference list) when not yet accepted for publication. It would therefore be more appropriate to cite those as Hendrickx, Abdala & Choiniere and Hendrickx, Choiniere & Abdala, unpublished in text in your case. As this is the only thing, these minor changes can be done during the proofing process.

#